# XConv: Low-memory stochastic backpropagation for convolutional layers

## Abstract

Training convolutional neural networks at scale demands substantial memory, largely due to storing intermediate activations for backpropagation. Existing approaches—such as checkpointing, invertible architectures, or gradient approximation methods like randomized automatic differentiation—either incur significant computational overhead, impose architectural constraints, or require non-trivial codebase modifications. We propose XConv, a near-drop-in replacement for standard convolutional layers that addresses all three limitations: it preserves standard backpropagation, imposes no architectural constraints, and integrates into existing codebases with minimal changes. XConv exploits the algebraic structure of convolutional layer gradients, storing highly compressed activations and approximating weight gradients via multi-channel randomized trace estimation. We establish convergence guarantees and derive error bounds for the proposed estimator, showing that the variance of the resulting gradient errors is comparable to that of stochastic gradient descent. Empirically, XConv achieves performance comparable to exact gradient methods across classification, generative modeling, super-resolution, inpainting, and segmentation—with gaps that narrow as the number of probing vectors increases—while reducing activation memory—by a factor of two or more when convolutional activations dominate—and remaining computationally competitive with optimized convolution implementations at larger batch sizes. At reduced (half) precision the gradient approximation error falls to the rounding floor, so XConv adds essentially no error beyond that of low-precision arithmetic, a regime typical of finetuning and on-device training.

## 1 Introduction

While transformer-based architectures have achieved remarkable success across numerous domains, their quadratic computational complexity with respect to sequence length poses significant scalability challenges (Vaswani et al., 2017; Tay et al., 2022). This has renewed interest in convolutional architectures (Younesi et al., 2024), with recent works demonstrating that carefully designed convolutional neural networks (CNNs) can match or exceed transformer performance in certain settings (Mao et al., 2021; Liu et al., 2022; Cui et al., 2023b;a; Woo et al., 2023; Wang et al., 2023; Liu et al., 2023; Hou et al., 2024; Ding et al., 2024). As such, convolutional layers continue to form a key component of current neural network designs (Ronneberger et al., 2015; Ding et al., 2022; Liu et al., 2022; Wang et al., 2024; Ma et al., 2024b). However, scaling up CNNs remains challenging due to two primary factors: (i) while the computational demands during forward evaluation are relatively modest, significant computational resources are needed during training (Griewank & Walther, 2000; Ronneberger et al., 2015; Chen et al., 2016); (ii) training requires storing intermediate activations for backpropagation, creating a memory bottleneck that becomes particularly acute when scaling to higher-dimensional data. While several recent works have addressed the computational complexity of CNN training through architectural innovations (Howard et al., 2017; Liu et al., 2022; Chen et al., 2023a;b; Vasu et al., 2023), the memory bottleneck associated with storing activations remains a critical challenge.

Existing methods for addressing this memory bottleneck include checkpointing approaches (Griewank & Walther, 2000; Chen et al., 2016; Beaumont et al., 2019; Shah et al., 2021; Feng & Huang, 2021; He & Yu, 2023; Korthikanti et al., 2023; Beaumont et al., 2024; Hong et al., 2025), which recompute activations

during the backward pass, yielding exact gradients but incurring significant computational overhead and requiring careful integration with automatic differentiation to ensure correct gradient flow; invertible network architectures (Gomez et al., 2017; Haber & Ruthotto, 2017; Jacobsen et al., 2018; Hascoet et al., 2023; Ulidowski, 2023; Orozco et al., 2024; Zhang & Zhang, 2024; Zhao et al., 2024a), which enable activations to be recovered from outputs but impose strict architectural constraints that limit representation power; and approximate-gradient methods, which substitute an inexact gradient for the exact one. Among these, randomized automatic differentiation (RAD) (Oktay et al., 2021) is closest to our setting but intervenes in the computational graph; zeroth-order optimization (Malladi et al., 2023) forgoes backpropagation entirely, estimating gradients from forward evaluations alone, but its estimator variance grows with the number of parameters and it is established only for fine-tuning pretrained models rather than for training from scratch; approximate and memory-sharing backpropagation (Yang et al., 2024b) lowers the activation memory of nonlinearities and normalization layers through custom backward kernels in transformer fine-tuning, leaving convolutional layers unaddressed; and direct feedback alignment (DFA) (Nøkland, 2016; Han & Yoo, 2019; Wang et al., 2019; Nøkland & Eidnes, 2019; Launay et al., 2020; Frenkel et al., 2021; Refinetti et al., 2021; Nakajima et al., 2024; Yang et al., 2024a) bypasses the backward pass by routing error signals through fixed random matrices directly to each layer. Collectively, each family of methods trades one constraint for another: checkpointing sacrifices computation for exact gradients, invertible architectures sacrifice design flexibility, and approximation-based approaches require non-trivial codebase modifications, specialized framework support, or changes to the training pipeline. A method that reduces memory while preserving standard backpropagation, imposing no architectural constraints, and integrating seamlessly into existing codebases remains an open challenge.

Our work is based on the premise that exact computations are often not needed—a viewpoint advocated in the field of randomized linear algebra (Tropp et al., 2019; Martinsson & Tropp, 2020), and more recently in parametric machine learning (Oktay et al., 2021), where it has been argued that spending computational resources on exact gradients is unnecessary when stochastic optimization is used. A similar argument was used earlier in the context of parameter estimation with partial differential equation constraints (Haber et al., 2012; Aravkin et al., 2012; van Leeuwen & Herrmann, 2014). Building on this premise, we propose **XConv**, a memory-efficient training approach for convolutional layers that addresses all three limitations simultaneously. XConv preserves standard backpropagation—unlike DFA or zeroth-order methods—imposes no architectural constraints—unlike invertible networks—and requires no changes to the computational graph or training pipeline—unlike RAD or checkpointing. This is achieved by exploiting the specific algebraic structure of convolutional layer gradients: we store highly compressed versions of layer activations and approximate weight gradients using multi-channel randomized trace estimation, enabling XConv to serve as a near-drop-in replacement for standard 2D and 3D convolutional layers in existing architectures while remaining computationally competitive with optimized convolution implementations at larger batch sizes.

By means of relatively straightforward algebraic manipulations, we write the gradient with respect to a convolution weight in terms of the trace of a matrix formed from the outer product of the convolutional layer input and the backpropagated residual, combined with a shift operation. Next, we approximate this trace with an unbiased randomized trace estimation technique (Hutchinson, 1989; Avron & Toledo, 2011; Roosta-Khorasani & Ascher, 2015; Martinsson & Tropp, 2020; Meyer et al., 2021) for which we prove convergence and derive theoretical error bounds by extending recent theoretical results (Cortinovis & Kressner, 2022). We show that exact convolution gradients are not strictly necessary: randomized gradient estimates yield noise comparable in scale to stochastic optimization noise while reducing memory consumption, with accuracy that improves systematically with the number of probing vectors.

**Practical implications.** Because XConv reduces memory without altering the architecture, the optimization objective, or the backpropagation pipeline, its benefits are most pronounced where activation memory, rather than raw compute, is the binding constraint—i.e., high-resolution and volumetric training, and on-device finetuning or continual adaptation at the edge, where data parallelism does not lower the per-device memory footprint. In these regimes the convolutional activations dominate the memory budget, and XConv replaces each stored activation with a far smaller set of random projections, shrinking its footprint by the ratio of the full activation size to the number of probing vectors—one to two orders of magnitude at typical resolutions and probing budgets. Because XConv preserves standard backpropagation and leaves

the remaining layers untouched, it also composes with complementary memory-saving techniques aimed at the non-convolutional parts of the network, such as low-precision activation storage and operator-specific reductions; XConv removes the convolutional bottleneck while these methods address the rest, so that their savings accumulate. This complementarity is reinforced under reduced precision: at half precision the gradient error introduced by XConv becomes negligible relative to the rounding error already incurred, making the method well suited to the low-precision arithmetic common in finetuning and edge deployment.

**Contributions.**   The main contributions of this work are:

- We propose XConv, a drop-in replacement for standard convolutional layers that approximates gradients via multi-channel randomized trace estimation, enabling memory reduction with minimal implementation overhead.

- We establish convergence guarantees and derive theoretical error bounds for the proposed estimator, extending existing results to non-symmetric matrices.

- We empirically demonstrate that XConv achieves performance comparable to exact gradient methods across classification, generative modeling, super-resolution, inpainting, and segmentation, with accuracy that improves systematically with the number of probing vectors, while meaningfully reducing memory consumption.

**Outline.**   Section 2 provides a high-level overview of XConv and the memory–accuracy tradeoff it introduces. Section 3 reformulates the gradient of convolution layers as a trace and establishes convergence guarantees for its randomized approximation. Section 4 presents the XConv algorithm and describes how it integrates into existing architectures as a drop-in replacement. Section 5 evaluates gradient fidelity, memory savings, and downstream task performance across a range of architectures and applications.

## 2   Overview of XConv

Training high-resolution convolutional networks is increasingly constrained by activation memory during backpropagation. Existing memory-reduction techniques typically achieve savings through recomputation, architectural constraints, or modifications to the training dynamics. For example, activation checkpointing reduces memory usage by discarding activations and recomputing them during the backward pass, while invertible architectures reconstruct activations from subsequent layers. Despite their differences, these approaches reduce memory without exploiting the structure of the convolution gradient itself.

Instead, we seek to reduce the memory footprint of the convolutional layers while preserving the standard network architecture, the optimization objective, and the backpropagation pipeline. In particular, we focus on the memory required to compute the convolutional weight gradients.

The primary memory bottleneck arises from storing activations required for exact convolution-gradient computation. Since the weight gradient depends on interactions between the layer activations $\mathbf{X}$ and the backpropagated residuals $\delta\mathbf{Y}$, standard training must retain the full activation tensor throughout the forward pass. As image resolution and channel dimensionality increase, these activations become a dominant contributor to memory consumption.

Our key observation is that each convolution weight gradient admits a trace-based representation. Rather than viewing the gradient as the result of a convolution operation, we reinterpret it as the trace of a structured matrix involving the activations $\mathbf{X}$ and residuals $\delta\mathbf{Y}$. This perspective is useful because traces admit unbiased randomized estimators whose accuracy can be controlled through a small number of probing vectors.

This reformulation exposes a new memory–accuracy tradeoff. By replacing exact trace computation with randomized trace estimation, XConv stores only compressed projections of the activations during the forward pass. Increasing the probing rank improves gradient fidelity and recovers the exact gradient in the limit, while smaller probing ranks provide greater memory savings. Figure 1 illustrates this reformulation and the resulting XConv computation.

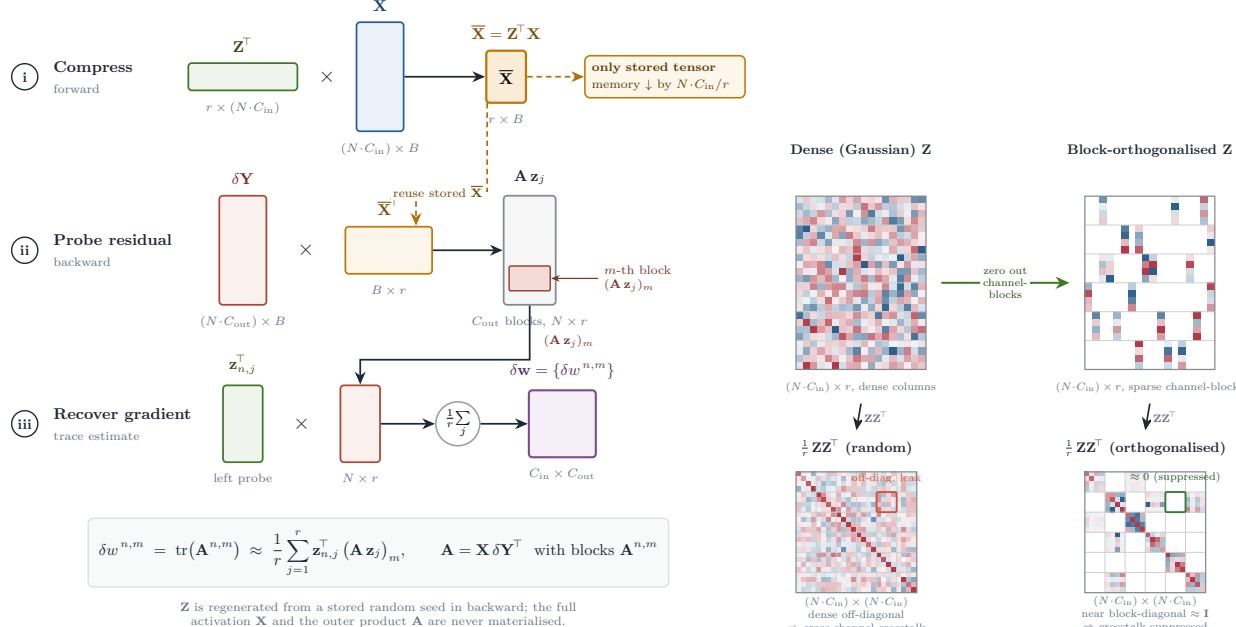

(a) The three-step multi-channel randomized trace estimation.

(b) Effect of block-orthogonalization on the probing matrices.

Figure 1: Overview of XConv. (a) Multi-channel randomized trace estimation: the three steps estimate the trace of a sub-block of the outer product, storing only the compressed activation $\overline{\mathbf{X}} = \mathbf{Z}^\top \mathbf{X}$ rather than the full input. (b) Probing matrices $\mathbf{Z}$ (top) and their Gram matrices $(1/r)\,\mathbf{Z}\mathbf{Z}^\top$ (bottom), for dense Gaussian probing (left) and block-orthogonalized probing (right); zeroing out channel blocks renders the Gram near block-diagonal, suppressing the off-diagonal cross-channel interference between channels.

We next formalize this observation by deriving a trace representation of convolution weight gradients and introducing the corresponding randomized estimator.

## 3 Theory

We start with a single-channel case and prove convergence and error bounds by extending existing results to non-symmetric matrices, and then generalize to multi-channel convolutions.

**Single channel case**  Let us start by writing the action of a single channel convolutional layer as follows

$$\mathbf{Y} = \mathbf{W}\mathbf{X} \in \mathbb{R}^{N \times B}, \quad \text{where} \quad \mathbf{W} = \sum_{i=1}^{n_w} w_i \mathbf{T}_{k(i)}, \tag{1}$$

and $N$, $B$, $n_w$ are the number of pixels, batch size, and number of convolution weights ($K^2$ for a $K$ by $K$ kernel), respectively. For the $i^{\text{th}}$ weight $w_i$, the convolutions themselves correspond to applying a circular shift with offset $k(i)$, denoted by $\mathbf{T}_{k(i)}$, followed by multiplication with the weight. Given this expression for the action of a single-channel convolutional layer, expressions for the gradient with respect to weights can easily be derived by using the chain rule and standard linear algebra manipulations (Petersen & Pedersen, 2008)—i.e., we have

$$
\begin{aligned}
\frac{\partial}{\partial w_i} f(\mathbf{W}\mathbf{X}) &= \text{tr}\left( \left( \frac{\partial f(\mathbf{W}\mathbf{X})}{\partial \mathbf{W}} \right)^\top \frac{\partial \mathbf{W}}{\partial w_i} \right) \\
&= \text{tr}\left( \left( \delta\mathbf{Y}\mathbf{X}^\top \right)^\top \mathbf{T}_{k(i)}^\top \right) = \text{tr}\left( \mathbf{X}\delta\mathbf{Y}^\top \mathbf{T}_{-k(i)} \right), \quad i = 1, \dots, n_w.
\end{aligned}
\tag{2}
$$

This expression for the gradient with respect to the convolution weights corresponds to computing the trace—i.e., the sum of the diagonal elements denoted by $\text{tr}(\mathbf{A}) = \sum_i A_{ii}$, of the outer product between the residual collected in $\delta\mathbf{Y}$ and the layer's input $\mathbf{X}$, after applying the shift. The latter corresponds to a right circular shift along the columns.

Computing estimates for the trace through the action of matrices—i.e., without access to entries of the diagonal, is common practice in the emerging field of randomized linear algebra (Tropp et al., 2019; Martinsson & Tropp, 2020). Going back to the seminal work by Hutchinson (Hutchinson, 1989; Meyer et al., 2021), unbiased matrix-free estimates for the trace of a matrix exist involving probing with random vectors $\mathbf{z}_j$, $j = 1, \ldots, r$, with $r$ the number of probing vectors and $\mathbb{E}(\mathbf{z}_j \mathbf{z}_j^\top) = \mathbf{I}$ with $\mathbf{I}$ the identity matrix. Under this assumption, unbiased randomized trace estimates can be derived from

$$\text{tr}(\mathbf{A}) = \text{tr}\left(\mathbf{A}\mathbb{E}\left[\mathbf{z}\mathbf{z}^\top\right]\right) = \mathbb{E}\left[\mathbf{z}^\top \mathbf{A}\mathbf{z}\right] \approx \frac{1}{r}\sum_{j=1}^{r}\left[\mathbf{z}_j^\top \mathbf{A}\mathbf{z}_j\right]. \tag{3}$$

By combining equation 2 with the above unbiased estimator for the trace, we arrive at the following approximation for the gradient with respect to the convolution weights:

$$\delta w_i \approx \frac{1}{r}\sum_{j=1}^{r}\left(\mathbf{z}_j^\top \mathbf{X}\right)\left(\delta\mathbf{Y}^\top \mathbf{T}_{-k(i)}\mathbf{z}_j\right), \quad i = 1, \ldots, n_w. \tag{4}$$

From this expression the memory savings during the forward pass are obvious since $\overline{\mathbf{X}} = \mathbf{Z}^\top \mathbf{X}$, where $\overline{\mathbf{X}} \in \mathbb{R}^{r \times B}$ with $r \ll N$. However, convergence rate guarantees were only established under the additional assumption that $\mathbf{A}$ is positive semi-definite (PSD, (Kaperick, 2019)). While the outer product $\mathbf{X}\delta\mathbf{Y}^\top \mathbf{T}_{-k(i)}$ we aim to probe here is not necessarily PSD, improving upon recent results by (Cortinovis & Kressner, 2022), we show that the condition of PSD can be relaxed to asymmetric matrices by a symmetrization procedure that does not change the trace. More precisely, we show in the following proposition that the gradient estimator in equation 4 is unbiased and converges to the true gradient as $r \to \infty$ with a rate of about $r^{-1/2}$ (for details of the proof, we refer to the Appendix A.2).

**Proposition 1.** *Let $\mathbf{A} \in \mathbb{R}^{N \times N}$ be a square matrix and let the probing vectors be i.i.d. Gaussian with $0$ mean and unit variance. Then for any small number $\delta > 0$, with probability $1 - \delta$, we have*

$$\left|\frac{1}{r}\sum_{i=1}^{r}\left[\mathbf{z}_i^\top \mathbf{A}\mathbf{z}_i\right] - \text{tr}\left(\mathbf{A}\right)\right| \leq \frac{4\|\mathbf{A}\|_2}{r}\log\frac{2}{\delta} + \frac{2\|\mathbf{A}\|_F}{\sqrt{r}}\log^{1/2}\frac{2}{\delta}.$$

Imposing a small probability of failure $\delta$ means the $\log\frac{2}{\delta}$ term in the upper bound is large, which implies that neither term in the upper bound is dominating for all the $r$ values. Depending on which term is dominant, the range of $r$ can be divided into two regimes, the small $r$ regime and the large $r$ regime. In the small $r$ regime, the first term dominates, and the error decays as $1/r$. In the large $r$ regime, the second term dominates and the error decays as $1/\sqrt{r}$. The phase transition happens when $r$ is about $4\|\mathbf{A}\|_2^2/\|\mathbf{A}\|_F^2 \log(2/\delta) \equiv \frac{4}{\rho}\log(2/\delta)$, where $\rho \equiv \|\mathbf{A}\|_F^2/\|\mathbf{A}\|_2^2$ is known as the effective rank, which reflects the rate of decay of the singular values of $\mathbf{A}$. We see that as $r$ increases, the larger the effective rank is, the earlier the phase transition occurs, after which the decay rate of the error will slow down. Before discussing details of the proposed algorithm, let us first extend the above randomized trace estimator to multi-channel convolutions.

**Multi-channel case** In general, convolutional layers involve several input and output channels. In that case, the output of the $m^{\text{th}}$ channel can be written as

$$\mathbf{Y}^m = \sum_{n=1}^{C_{\text{in}}}\mathbf{W}^{n,m}\mathbf{X}^n, \quad \text{where} \quad \mathbf{W}^{n,m} = \sum_{i=1}^{n_w} w_i^{n,m}\mathbf{T}_{k(i)} \tag{5}$$

for $n = 1, \ldots, C_{\text{in}}$, $m = 1, \ldots, C_{\text{out}}$ with $C_{\text{in}}$, $C_{\text{out}}$ the number of input and output channels and $w_i^{n,m}$ the $i^{\text{th}}$ weight between the $n^{\text{th}}$ input and $m^{\text{th}}$ output channel. In this multi-channel case, the gradients consist of the single channel gradient for each input/output channel pair, i.e., $\delta w_i^{n,m} = \text{tr}(\mathbf{X}^n(\delta\mathbf{Y}^m)^\top \mathbf{T}_{-k(i)})$.

While randomized trace estimation can in principle be applied to each input/output channel pair independently, we propose to treat all channels simultaneously to further improve computational performance and memory use. Let the outer product of the $(n, m)^{\text{th}}$ input/output channel be $\mathbf{A}^{n,m}$, i.e., $\mathbf{A}^{n,m} = (\mathbf{X}^n (\delta \mathbf{Y}^m)^\top \mathbf{T}_{-k(i)})^\top$, computing $\delta w_i^{n,m}$ means estimating $\text{tr}(\mathbf{A}^{n,m})$. To save memory, instead of probing each $\mathbf{A}^{n,m}$, we probe the stacked matrix

$$\mathbf{A} = \begin{pmatrix} \mathbf{A}^{1,1} & \cdots & \mathbf{A}^{1,C_{\text{in}}} \\ \vdots & & \vdots \\ \mathbf{A}^{C_{\text{out}},1} & \cdots & \mathbf{A}^{C_{\text{out}},C_{\text{in}}} \end{pmatrix}$$

by $r$ length $C_{\text{in}} \times N$ probing vectors stored in $\mathbf{Z} \in \mathbb{R}^{NC_{\text{in}} \times r}$, and estimate each $\text{tr}(\mathbf{A}^{m,n})$ via the following estimators

$$G^{m,n}(\mathbf{A}) := \frac{1}{r} \sum_{j=1}^{r} \mathbf{z}_{n,j}^\top \left( \mathbf{A} \mathbf{z}_j \right)_m, \tag{6}$$

where $(\cdot)_m$ extracts the $m^{\text{th}}$ block from the input vector. That is to say, we simply stack the input and residual, yielding matrices of size $(N \times C_{\text{in}}) \times B$ and $(N \times C_{\text{out}}) \times B$ whose outer product $\mathbf{X} \delta \mathbf{Y}^\top$ (i.e., $\mathbf{A}^\top$ of the $\mathbf{A}$ in equation 6) is no longer necessarily square. To estimate the trace of each $N \times N$ sub-block, in equation 6, we ($i$) probe the full outer product from the right with $r$ probing vectors $\mathbf{z}_j$ of length $(N \times C_{\text{in}})$; ($ii$) reshape the resulting matrix into a tensor of size $(N, C_{\text{out}}, B)$ while the probing matrix is shaped into a tensor of size $(N, C_{\text{in}}, B)$ (i.e., separate each block of $\mathbf{A} z_j$), and ($iii$) probe each individual block again from the left. This leads to the desired gradient collected in a $C_{\text{in}} \times C_{\text{out}}$ matrix. We refer to Figure 1, which illustrates this multi-channel randomized trace estimation. After ($i$), we only need to save $\overline{\mathbf{X}} = \mathbf{Z}^\top \mathbf{X}$ in memory rather than $\mathbf{X}$ that leads to a memory reduction by a factor of $\frac{NC_{\text{in}}}{r}$.

Unfortunately, the improved memory use and computational performance boost of the above multi-channel probing reduces the accuracy of the randomized trace estimation because of crosstalk amongst the channels. Since this crosstalk is random, the induced error can be reduced by increasing the number of probing vectors $r$, but this will go at the expense of more memory use and increased computation. To avoid this unwanted overhead, we introduce a new type of random probing vectors that minimizes the crosstalk by again imposing $\mathbb{E}(\mathbf{z}\mathbf{z}^\top) = \mathbf{I}$ but now on the multi-channel probing vectors that consist of multiple blocks corresponding to the number of input/output channels.

Explicitly, we draw each $\mathbf{z}_{n,j}$, the $n^{\text{th}}$ block of the $j^{\text{th}}$ probing vector, according to

$$\mathbf{z}_{n,j} \sim \begin{cases} \mathcal{N}(\mathbf{0}, \mathbf{I}_N) & \text{with probability } p_n \\ 0 & \text{with probability } 1 - p_n \end{cases}. \tag{7}$$

For different values of $(n, j)$, the $\mathbf{z}_{n,j}$'s are drawn independently with a predefined probability $p_n$ of generating a nonzero block. Compared to conventional (Gaussian) probing vectors (see Figure 1b top left), these multi-channel probing vectors contain sparse non-zero blocks (see Figure 1b top right), which reduces the crosstalk (juxtapose with second row of Figure 1b). It can be shown that crosstalk becomes less when $p_n \downarrow 0$ and $r \to \infty$.

Given probing vectors drawn from equation 7, we have to modify the scaling factor of the multi-channel randomized trace estimator equation 6 to ensure it is unbiased,

$$\widetilde{G}^{n,m}(\mathbf{A}) := \frac{1}{\text{nnz}(\mathbf{Z}_n)} \sum_{j=1}^{r} \mathbf{z}_{n,j}^\top \left( \mathbf{A} \mathbf{z}_j \right)_m, \tag{8}$$

where $\text{nnz}(\mathbf{Z}_n)$ is the number of non-zero columns in block $n$. We prove the following convergence result for this estimator (the proof can be found in Appendix A.2).

**Theorem 1** (Succinct version). *Let $p = \min\limits_{n} p_n$, $r$ be the number of probing vectors. For any small number $\delta > 0$, with probability at least $1 - \delta - 3C_{in}e^{-rp^2/2}$, we have for any $n = 1, \ldots, C_{in}$ and $m = 1, \ldots, C_{out}$,*

$$\left| \widetilde{G}^{n,m}(\mathbf{A}) - \text{tr}(\mathbf{A}^{n,m}) \right| \le c \cdot \frac{\frac{1}{\sqrt{p_n}} \|\mathbf{A}^{n,m}\|_F + \sum\limits_{j=1, j\neq n}^{C_{in}} \sqrt{\frac{p_j}{p_n}} \|\mathbf{A}^{n,j}\|_F}{\sqrt{r}} \log^{1/2} \frac{C_{out}C_{in}}{\delta},$$

*where $c$ is an absolute constant and $C_{in}$ and $C_{out}$ are the numbers of input and output channels.*

Theorem 1 provides convergence guarantee for our special multi-channel simultaneous probing procedure. Similar to Proposition 1, Theorem 1 in its original form (supplementary material) also has a two-phase behaviour. So the discussion under Proposition 1 applies here. For simplification of presentation, we only presented the bound for the large $r$ regime in this succinct version. Still, we can see that the error bound for estimating $\mathrm{tr}(\mathbf{A}^{n,m})$ not only depends on the norm of the current block $\mathbf{A}^{n,m}$, but also other blocks in that row, which is expected since we simultaneously probe the entire row instead of each block individually for memory efficiency. Admittedly, due to technical difficulties, we cannot theoretically show that decreasing the sampling probability $p$ decreases the error. Nevertheless, we observe better performance in the numerical experiments.

# 4 Stochastic optimization with multi-channel randomized trace estimation

Given the expressions for approximate gradient calculations and bounds on their error, we now present the XConv algorithm and demonstrate its use as a drop-in replacement in existing convolutional architectures.

## 4.1 Low-memory stochastic backpropagation

The key point of the randomized trace estimator in Equation equation 8 is that it allows for on-the-fly compression of the state variables during the forward pass. For a single convolutional layer $\mathbf{Y} = \mathbf{conv}(\mathbf{X}, \mathbf{w})$ with input $\mathbf{X}$ and convolution weights $\mathbf{w}$, our approximation involves three simple steps, namely **(1)** probing of the state variable $\overline{\mathbf{X}} = \mathbf{Z}^\top \mathbf{X}$, **(2)** matrix-free formation of the outer product $\mathbf{L} = \overline{\mathbf{X}}\delta \mathbf{Y}^\top$, and **(3)** approximation of the gradient via $\delta w_i = \frac{1}{r}\mathrm{tr}(\mathbf{LT}_{-k(i)}\mathbf{Z})$, $i = 1, \ldots, n_w$. These three steps lead to substantial memory reductions even for a relatively small image of size $32 \times 32$ ($N = 1024$ pixels) and $C_{\mathrm{in}} = 16$. In that case, our approach leads to a memory reduction by a factor of $\frac{NC_{\mathrm{in}}}{r} = 2^{14-\gamma}$ for $r = 2^\gamma$. For $\gamma = 7$ this gives a roughly $100\times$ reduction in the stored activation *of that single layer*; the network-level saving is smaller and set by the share of non-convolutional activations (Section 5.3). Because the probing vectors are generated on the fly, we only need to allocate memory for $\overline{\mathbf{X}}$ during the forward pass as long as we also store the state $s$ of the random generator. During backpropagation, we initialize the state, generate the probing vectors, followed by applying a shift and product by $\mathbf{L}$. These steps are summarized in Algorithm 1. This simple algorithm provides a highly compressed estimate of the true gradient with respect to its weights.

---

**Algorithm 1** Low-memory approximate gradient convolutional layer. The random seed $s$ and random probing matrix $\mathbf{Z}$ are independently redrawn for each layer and training iteration.

---

**Forward pass:**
1. Forward convolution $\mathbf{Y} = \mathbf{conv}(\mathbf{X}, \mathbf{w})$
2. Draw a new random seed $s$ and probing matrix $\mathbf{Z}[s]$
3. Compute and save $\overline{\mathbf{X}} = \mathbf{Z}^\top[s]\mathbf{X} \in \mathbb{R}^{r \times B}$
4. Store $\overline{\mathbf{X}}, s$

**Backward pass:**
1. Load random seed $s$ and probed forward $\overline{\mathbf{X}}$
2. Redraw probing matrix $\mathbf{Z}[s]$ from $s$
3. Compute backward probe $\mathbf{L} = \overline{\mathbf{X}}\delta \mathbf{Y}^\top$
4. Compute gradient $\delta w_i = \frac{1}{r}\mathrm{tr}(\mathbf{LT}_{-k(i)}\mathbf{Z}[s])$

---

## 4.2 Ease of Integration

Figure 2 illustrates the ease of integrating XConv: it implements Algorithm 1 internally but fully encapsulates the complexity within the layer abstraction. We also provide an adaptive version that selectively disables XConv in deep layers with small spatial extent, where activation storage is inexpensive.

(a) Directly instantiating XConv layers.

```python
class Net(nn.Module):
    def __init__(self):
        super(Net, self).__init__()
        self.conv1 = Xconv2D(
            1, 32, 3, 32, 1, padding=1
        )

        self.conv2 = Xconv2D(
            32, 64, 3, 32, 1, padding=1
        )
        self.dropout1 = nn.Dropout(0.25)
        self.dropout2 = nn.Dropout(0.5)
        self.fc2 = nn.Linear(128, 10)
```

(b) Converting an existing model in-place.

```python
from pyxconv.utils import convert_net

model = nn.Sequential(
    nn.Conv2d(3, 64, 3, padding=1),
    nn.BatchNorm2d(64),
    nn.ReLU()
)

convert_net(
    model,
    ps=256,
    xmode="independent"
)
```

Figure 2: XConv can be integrated either by directly instantiating XConv layers (left) or by converting existing convolutional models using a single API call (right).

### 4.3 Practical Guidance for using XConv

XConv exposes two knobs that jointly set the memory–accuracy operating point: the batch size $B$ and the number of probing vectors $r$. For a single convolutional layer the only stored tensor is the compressed activation $\overline{\mathbf{X}} = \mathbf{Z}^\top \mathbf{X} \in \mathbb{R}^{r \times B}$, whose size scales with the product $rB$; relative to storing the full input, this is a memory reduction by the factor $NC_{\text{in}}/r$. At the level of a single layer, $r$ and $B$ are therefore interchangeable memory knobs—i.e., dividing $r$ by a constant frees exactly as much activation memory as dividing $B$ by the same constant.

The two knobs are not equivalent at the network level. XConv compresses only convolutional activations, whereas the batch size scales the memory of every layer, convolutional or not. Reducing $r$ therefore yields diminishing returns once the non-convolutional activations (normalizations, skip connections, and pointwise operations) come to dominate the budget, while reducing $B$ continues to lower the memory of the entire network. Increasing $r$, on the other hand, improves gradient fidelity: as we show in Section 5, the average gradient error decreases with $r$ toward the exact-convolution floor (Figures 4 and 5).

These observations yield a simple rule of thumb. To meet a fixed memory budget, use the largest $r$ the budget allows and recover the remaining memory by lowering the batch size rather than by shrinking $r$—i.e., a large $r$ keeps the probing error small, and the batch size is the more effective memory lever because it acts on the whole network. The cost of a larger $r$ is compute: it raises the per-iteration cost of the gradient estimate, as quantified by the wall-clock benchmarks in Section 5 (Figure 12), so under a fixed memory budget the pair $(r, B)$ trades gradient accuracy against compute, and the practitioner can choose where to sit on this tradeoff.

Finally, the additional noise introduced by probing is unbiased (Theorem 1). Lowering the batch size raises the variance of the stochastic gradient, but because this noise is zero-mean its effect on the optimization trajectory can be averaged out over iterations—i.e., reducing the learning rate and training for correspondingly more steps suppresses it, exactly as for ordinary minibatch SGD. The volumetric finetuning experiment of Section 5.5.5 is trained under such a recipe—a reduced learning rate and more steps than the pretrained model's default, applied identically to both the exact and the probed runs—under which even the noisiest setting ($r = 4$) attains the exact-gradient accuracy.

## 5 Experiments

We now systematically analyze the consequences of replacing standard convolutional layers with XConv on gradient fidelity, memory consumption, and computational overhead. We first perform a layer-by-layer analysis of gradient deviation, then introduce Average Gradient Error (AGE) as a global diagnostic metric to quantify aggregate gradient fidelity across entire models. We then relate these findings to peak memory

usage and wall-clock benchmarks, enabling a comprehensive evaluation of the accuracy–memory–compute trade-offs induced by XConv.

## 5.1 Minibatch versus randomized trace estimation errors

Simply stated, stochastic optimization involves gradients that contain random errors known as gradient noise. As long as this noise is not too large and independent for different gradient calculations, algorithms such as stochastic gradient descent, where gradients are computed for randomly drawn minibatches, converge under certain conditions. In addition, the presence of gradient noise helps the algorithm to avoid bad local minima, which arguably leads to better generalization of the trained network (Neelakantan et al., 2015; Huang et al., 2020). Therefore, as long as the batch size is not too large, one can expect the trained network to perform well.

We argue that the same applies to stochastic optimization with gradients approximated by (multi-channel) randomized trace estimation as long as the errors behave similarly. In a setting where memory comes at a premium this means that we can expect training to be successful for gradient noise with similar variability. To this end, we conduct an experiment where the variability of $5 \times 5 \times C_{\text{in}} \times C_{\text{out}}$ convolution weights is calculated for the true gradient for different randomly drawn minibatches of size $B = 128$. We do this for a randomly initialized image classification network designed for the CIFAR-10 dataset (for network details, see Table 9 in Appendix A.5).

For comparison, approximate gradients are also calculated for randomized trace estimates obtained by probing independently ("Indep." in blue), multi-channel ("Multi" in orange), and multi-channel with orthogonalization ("Multi-Ortho" in green). The batch sizes are for a fixed number of probing vectors of $r = 2048$ selected such that the total memory use is the same as for the true gradient calculations. From the plots in Figure 3a, we observe that as expected the independent probing is close to the true gradient followed by the more memory efficient multi-channel probing with and without orthogonalization. While all approximate gradients are within the 99% confidence interval, the orthogonalization has a big effect when the gradients are small (see conv3).

To better understand the interplay between batch size $B \in \{64, 128, 256, 1024\}$ and probing vectors $r \in \{64, 256, 512\}$, we estimate the standard deviation from 40 randomly drawn minibatches. As expected, the standard deviations increase for smaller batch size and fewer probing vectors. However, since XConv's smaller memory footprint allows for larger batch sizes, the variability can be controlled within a given memory budget. Figure 3b shows that the standard deviation reduces with increasing batch size, mirroring the behaviour of stochastic gradient descent. This confirms that the approximation noise introduced by XConv does not dominate mini-batch noise, reaffirming that exact gradient computations are unnecessary for stable training.

## 5.2 Average Gradient Error

Comparing approximate and exact gradients in deep networks is nontrivial: gradients vary across layers, scales, and architectures, and per-layer discrepancies do not directly reflect the cumulative effect on optimization. To enable architecture-level comparison, we introduce a global diagnostic metric—Average Gradient Error (AGE)—that quantifies the aggregate deviation between gradients estimated by standard convolution and randomized trace estimation across an entire model. AGE measures the magnitude of gradient noise induced by approximation and stochasticity, aggregated across mini-batches, and is intended as a diagnostic of gradient fidelity rather than a predictor of generalization.

Let $\mathcal{D} = \{(x_i, y_i)\}_{i=1}^{K}$ denote a dataset of $K$ samples, $\ell(f_\theta(x_i), y_i)$ a per-example loss, and $L(\theta) = \frac{1}{K} \sum_{i=1}^{K} \ell(f_\theta(x_i), y_i)$ the empirical risk with full gradient $\mathbf{g}(\theta) = \nabla_\theta L(\theta)$. We partition $\mathcal{D}$ into $M = \lfloor K/B \rfloor$ mini-batches $\{\mathcal{B}_b\}_{b=1}^{M}$ of size $B$, each yielding a mini-batch gradient $\mathbf{g}^{(b)}(\theta) = \frac{1}{B} \sum_{i \in \mathcal{B}_b} \nabla_\theta \ell(f_\theta(x_i), y_i)$ so that $\mathbf{g}(\theta) = \frac{1}{M} \sum_{b=1}^{M} \mathbf{g}^{(b)}(\theta)$. The Average Gradient Error is then defined as

$$\text{AGE}(\theta) = \frac{1}{M} \sum_{b=1}^{M} \left\| \mathbf{g}(\theta) - \mathbf{g}^{(b)}(\theta) \right\|_2^2. \tag{9}$$

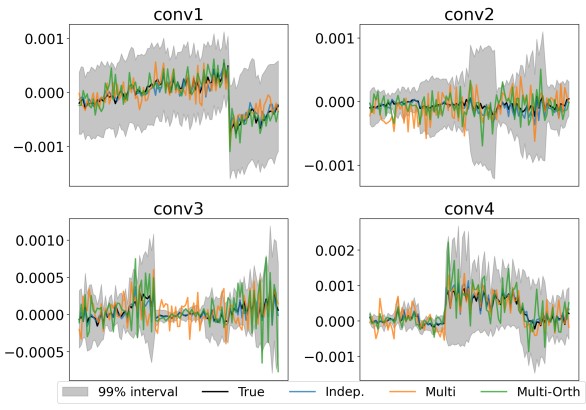

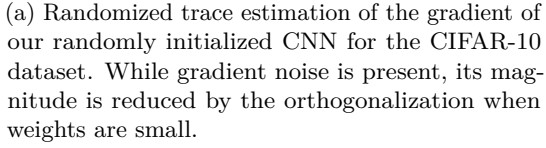

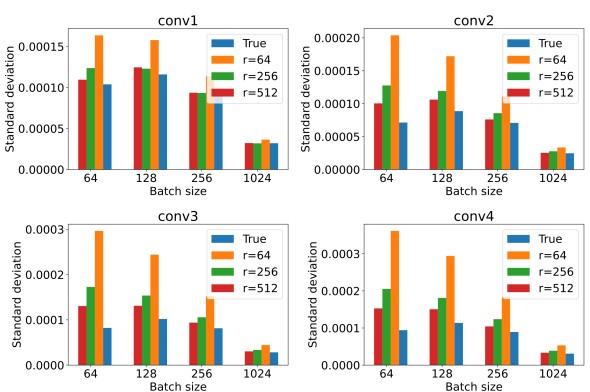

(a) Randomized trace estimation of the gradient of our randomly initialized CNN for the CIFAR-10 dataset. While gradient noise is present, its magnitude is reduced by the orthogonalization when weights are small.

(b) Standard deviation of the gradients w.r.t. the weights for each of the four convolutional layers in the neural network. The standard deviation is computed over 40 mini-batches randomly drawn from the CIFAR-10 dataset.

Figure 3: Gradient analysis on CIFAR-10. (a) Per-weight gradient estimates for four convolutional layers under different probing strategies. (b) Standard deviation of gradients across 40 mini-batches for varying batch sizes and probing vectors $r$.

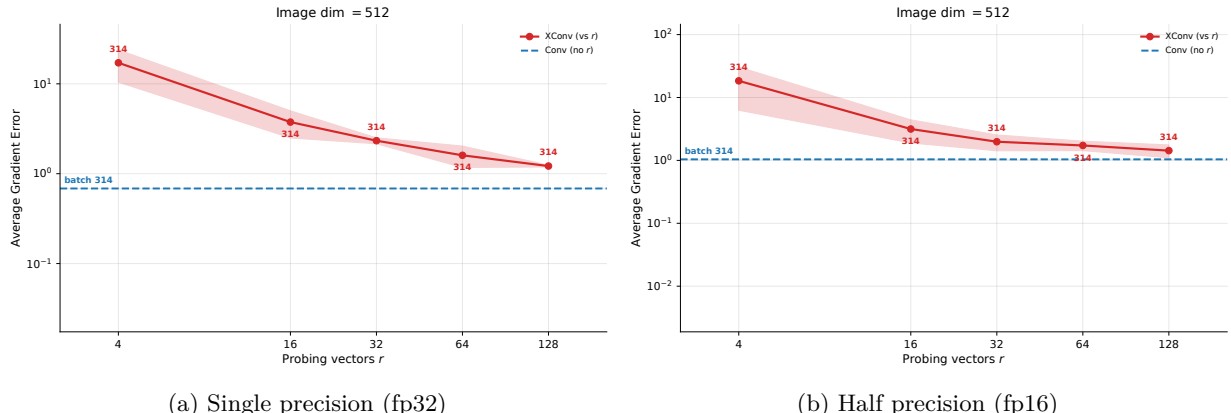

(a) Single precision (fp32)

(b) Half precision (fp16)

Figure 4: Average gradient error vs. the number of probing vectors $r$ for SqueezeNet at image dimension $N = 512$, in (a) single precision and (b) half precision, over 10 runs. The red curve is XConv and the blue dashed line is the standard convolution floor, which is independent of $r$. The annotated batch size at each point is the largest that fits the 80 GB memory budget. XConv's AGE decreases with $r$ toward the convolution floor; at $r = 128$ the gap is small. At half precision the XConv gradient error is negligible and coincides with the convolution floor.

For the randomized trace estimation, the mini-batch gradient given by $\mathbf{g}^{(b)}$ contains the stochastic mini-batch and approximation noise. For both the standard convolution and randomized trace estimation cases, AGE is measured relative to the same reference full-gradient $\mathbf{g}(\theta)$. We use AGE to compare gradient fidelity across models and approximation strategies, isolating the contribution of randomized trace estimation from stochastic mini-batch noise.

### 5.2.1 SqueezeNet

We begin our analysis with SqueezeNet (Iandola et al., 2016), a lightweight convolutional architecture designed to achieve competitive accuracy with significantly fewer parameters. SqueezeNet constitutes a challenging test

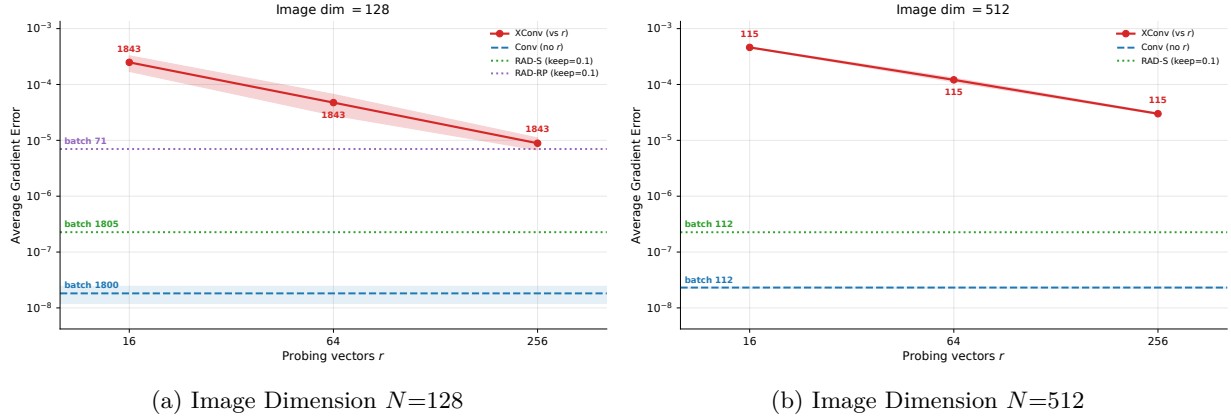

(a) Image Dimension $N$=128  (b) Image Dimension $N$=512

Figure 5: Average gradient error vs. the number of probing vectors $r$ for U-Net in single precision, at image dimensions (a) $N = 128$ and (b) $N = 512$, over 10 runs. The red curve is XConv, the blue dashed line the standard convolution floor, and the green dotted line the RAD-S reference at keep fraction 0.1. The annotated batch size at each point is the largest that fits the 80 GB memory budget. XConv increases AGE relative to standard convolution, but the gap narrows steadily as $r$ increases, indicating that the approximation noise can be controlled by adjusting the number of probing vectors. We omit the half-precision panels for U-Net because at fp16 the gradient error of every method collapses below the rounding floor, making the comparison uninformative.

case for XConv: its reduced spatial aggregation limits natural averaging effects that can otherwise mitigate stochastic gradient noise, potentially amplifying the impact of gradient approximation.

We evaluate AGE across probing vectors $r \in \{4, 16, 32, 64, 128\}$ and image dimensions $N \in \{128, 256, 512\}$. Results at $N = 512$ are reported in Figure 4, with the remaining image dimensions provided in Appendix A.3. Each point is annotated with the largest batch size that fits the 80 GB memory budget, which XConv attains by storing only the probed activations rather than the full input.

Despite the compact architecture and limited spatial smoothing, we observe a consistent reduction in AGE as the number of probing vectors $r$ increases, with XConv approaching the standard convolution floor. At $r = 128$ the gap is small, indicating that even in parameter-efficient architectures the gradient deviation introduced by XConv diminishes systematically with increased probing capacity.

### 5.2.2 U-Net

We continue our analysis on U-Net (Ronneberger et al., 2015), a convolutional encoder–decoder architecture with skip-connections that has become a core building block in score-based diffusion models. Its deep hierarchical structure and large intermediate activations make it a particularly challenging and practically relevant setting for evaluating XConv.

Figure 5 reports the AGE against the number of probing vectors $r \in \{16, 64, 256\}$ at image dimensions $N = 128$ and $N = 512$, with the intermediate resolution provided in Appendix A.3. Across both resolutions, XConv increases AGE relative to exact convolutional gradients; however, the gap narrows systematically as $r$ grows toward the standard convolution floor.

In contrast to lightweight architectures such as SqueezeNet, U-Net's extensive skip connections and large activations lead to higher gradient approximation error, though the error remains bounded and decreases with $r$. This behavior suggests that XConv can maintain adequate gradient fidelity even in deep, memory-intensive encoder–decoder architectures.

**Comparison with randomized automatic differentiation.** The U-Net is also the natural setting for a direct comparison with randomized automatic differentiation (RAD; Oktay et al. 2021), the approximate-gradient baseline closest to our method, in both its random-projection (RAD-RP) and sparse (RAD-S) variants.

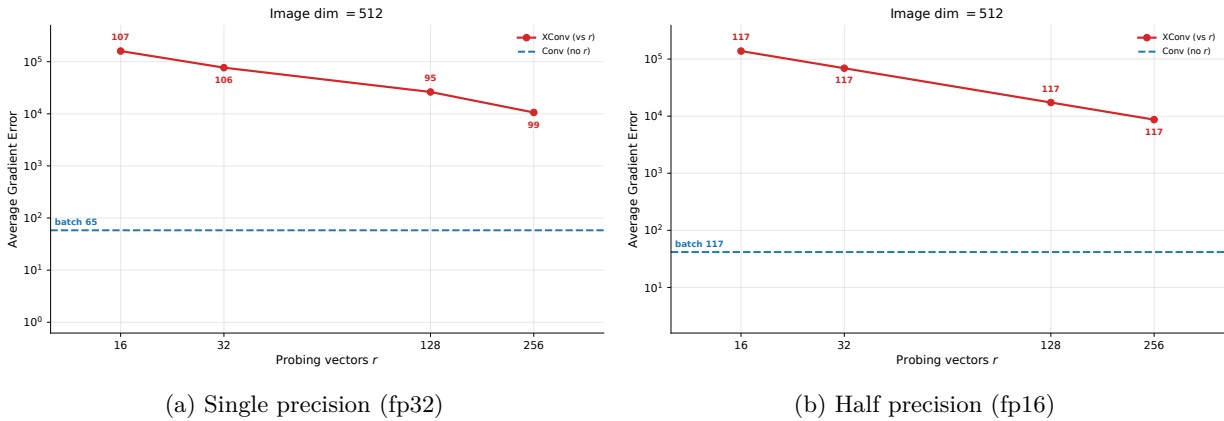

(a) Single precision (fp32)        (b) Half precision (fp16)

Figure 6: Average gradient error vs. the number of probing vectors $r$ for VanillaNet at image dimension $N = 512$, in (a) single precision and (b) half precision, over 10 runs. The red curve is adaptive XConv and the blue dashed line the standard convolution floor. The annotated batch size at each point is the largest that fits the 80 GB memory budget. In single precision the AGE remains above the convolution floor because VanillaNet's wide layers ($C_{\text{in}} \times C_{\text{out}} > 10^6$) inflate the estimator variance, consistent with Theorem 1; the adaptive selection of approximated layers also makes the trend in $r$ non-monotone. At half precision the XConv gradient error becomes negligible and coincides with the convolution floor.

We include RAD as a reference in the U-Net average-gradient-error curves (Figure 5) and peak-memory curves (Figure 9), with the batch size in every case taken as the largest that fits the memory budget. Two observations follow. First, on peak memory XConv is the most efficient of the compared methods—i.e., it tracks the lowest peak-memory profile and admits the largest batch size before reaching the budget, whereas RAD-RP reconstructs full-size activations in the backward pass, so its memory grows steeply with batch size; it admits only a small fraction of XConv's batch and becomes infeasible altogether at the highest resolution ($N = 512$). Second, on gradient fidelity the comparison is mixed—i.e., RAD-S attains a lower average gradient error than XConv at a comparable batch size, while XConv attains a lower error than RAD-RP. XConv and RAD-S therefore occupy complementary points on the memory–fidelity frontier—i.e., RAD-S is the more accurate estimator at matched memory, whereas XConv spans the full memory–fidelity range, scales to high resolution, and, unlike RAD, requires no intervention in the computational graph, acting as a drop-in layer replacement that preserves standard backpropagation.

### 5.2.3 VanillaNet

We complete our analysis on VanillaNet (Chen et al., 2023a), a minimalistic architecture with shallow feature maps and small activation footprints. Since aggressive gradient approximation is neither necessary nor uniformly beneficial in such lightweight networks, we employ an adaptive variant of XConv that selectively applies approximation to layers with the largest activation footprints.

We evaluate AGE across probing vectors $r \in \{16, 32, 128, 256\}$ and image dimensions $N \in \{64, 128, 256, 512\}$. Results at $N = 512$ are shown in Figure 6 with the remaining image dimensions provided in Appendix A.3.

As compared to U-Net and SqueezeNet, the single-precision AGE settles at a higher level above the convolution floor, which is consistent with the scaling behaviour provided by Theorem 1. In particular, the AGE depends on the product of the input and output channel dimensions. VanillaNet contains substantially wider convolutional layers than U-Net and SqueezeNet, with several layers for which $C_{\text{in}} \times C_{\text{out}} > 10^6$ (Table 6), whereas the other architectures remain significantly smaller. Since Theorem 1 predicts the estimator variance scales with channel dimensionality, VanillaNet naturally exhibits larger AGE under a fixed number of probing vectors.

Unlike the previous settings, AGE does not exhibit a strictly monotonic dependence on $r$. This behavior is expected: increasing the number of probing vectors reduces estimator variance only in layers where XConv is active, while layers computed with exact gradients contribute no approximation error. This highlights a key

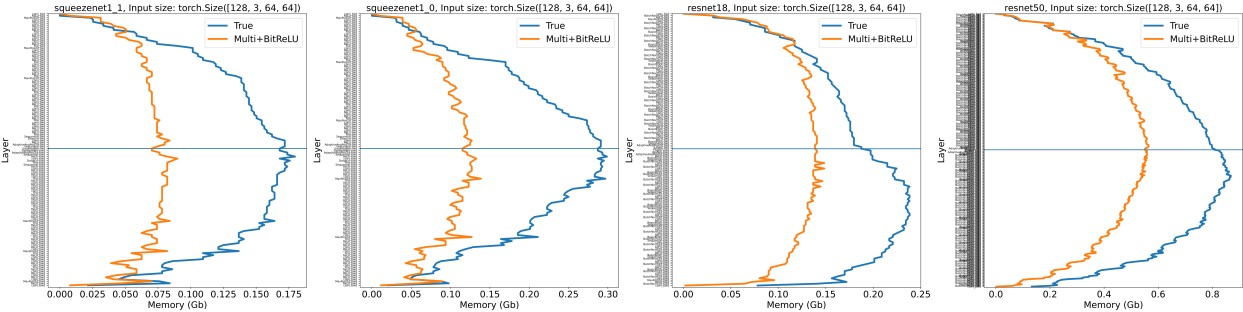

Figure 7: Layer-by-layer memory usage for a single gradient step. Standard convolution (blue) and XConv (orange) are compared for SqueezeNet 1.1, SqueezeNet 1.0, ResNet-18, and ResNet-50 (left to right) at a fixed input size. Memory savings reach 2× or more when convolutional layers dominate and are smaller when non-convolutional layers do, depending on the ratio of convolutional to non-convolutional layers.

design principle: gradient approximation need not be applied uniformly. In lightweight networks, selectively approximating only the most memory-intensive layers suffices, while preserving exact gradients elsewhere stabilizes optimization. This stands in contrast to deeper architectures, where increasing $r$ consistently improves gradient fidelity across the network.

Across all three architectures, the gradient error induced by XConv becomes very small at half precision—panel (b) of Figures 4 and 6 show that XConv's AGE coincides with the convolution floor in fp16, even for the wide layers of VanillaNet that dominate the single-precision error. The probing error is therefore small relative to the rounding error already incurred by half-precision training, which makes XConv particularly favorable for high-dimensional and high-resolution training where reduced precision is standard practice.

Taken together, these observations delineate the regimes in which XConv is least accurate. The single-precision approximation error is largest for architectures with very wide convolutional layers—i.e., VanillaNet, whose large channel products dominate the variance predicted by Theorem 1— and for generative and dense-prediction objectives, which are more sensitive to gradient noise than classification and therefore call for a larger number of probing vectors, with the marginal fidelity gain diminishing once $r$ is large. In every case the error is controlled by increasing $r$ rather than being intrinsic, and at half precision it falls to the fp16 rounding floor on all three tested architectures.

## 5.3 Overall effective memory savings

Gradient fidelity alone does not determine practical utility—XConv must also translate into meaningful end-to-end memory savings. We compare peak memory consumption between otherwise identical models differing only in whether standard convolutional layers are replaced with XConv.

Approximate gradient calculations with multi-channel randomized trace estimation can lead to significant memory savings within convolutional layers. Because these layers operate in conjunction with other network layers such as `ReLU` and batchnorms, the overall effective memory savings depend on the ratio of pure convolutional and other layers and on the interaction between them. This is especially important for layers such as `ReLU`, which rely on the next layer to store the state variable during backpropagation. Unfortunately, that approach no longer works because our low-memory convolutional layer does not store the state variable. However, this situation can be remedied easily by only keeping track of the signs (Oktay et al., 2021).

To assess the effective memory savings of multi-channel trace estimation, we include in Figure 7 layer-by-layer comparisons of memory usage for different versions of the popular SqueezeNet (Iandola et al., 2016) and ResNet (He et al., 2016). The memory use for the conventional implementation is plotted in blue and our implementation in orange. The results indicate that memory savings by a factor of two or more are achievable,

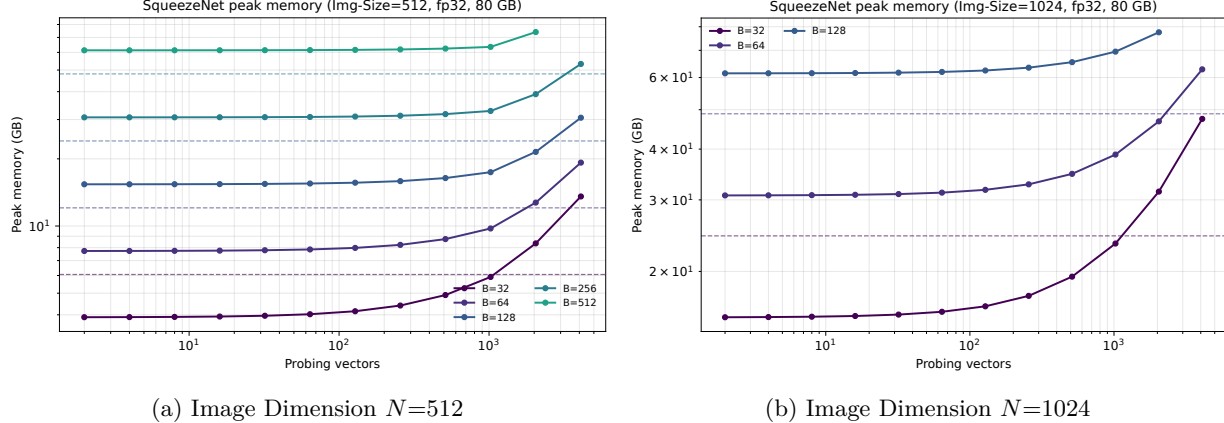

(a) Image Dimension $N$=512

(b) Image Dimension $N$=1024

Figure 8: Peak memory curves for SqueezeNet in single precision, as a function of the number of probing vectors, at image dimensions $N \in \{512, 1024\}$ under an 80 GB memory budget. Each solid curve is a fixed batch size $B$ and the dashed line of the matching color is the corresponding standard convolution. Peak memory grows only mildly with the number of probing vectors and stays well below the standard convolution baseline over a broad range, so XConv supports larger batch sizes within the same budget.

which can allow for larger batch sizes or increases in the width/depth of the network. As expected, the savings depend on the ratio of CNN versus other layers.

We extend this memory analysis to entire architectures: SqueezeNet, U-Net and VanillaNet. Peak memory is the limiting factor for feasible batch size and image resolution during training and therefore serves as the primary metric of interest.

Figure 8 reports the peak memory consumption of SqueezeNet across varying batch sizes and probing vectors at a fixed image resolution of $N = 512$ and $N = 1024$, under an 80 GB memory budget. The corresponding results for lower image resolutions and for half precision are provided in Figure 32 in the appendix. In these settings, XConv enables training regimes—either higher batch sizes or higher spatial resolutions—that would otherwise be infeasible under standard convolution due to memory constraints.

Results for U-Net at image resolutions $N \in \{128, 256\}$ are shown in Figure 9. The corresponding half-precision curves and the $N = 512$ result can be found in Figure 34. Despite U-Net's extensive skip connections and deep encoder–decoder structure, XConv reduces peak memory consumption across the tested probing vectors and batch sizes.

Finally, Figure 10 presents results for VanillaNet. Owing to its shallow architecture and heterogeneous layer-wise memory profile, we employ an adaptive variant of XConv that selectively applies approximation to layers. As a result, peak memory does not vary monotonically with the number of probing vectors. Nevertheless, across a broad range of probing budgets, XConv remains a memory-efficient alternative, confirming that selective approximation can yield meaningful savings.

### 5.4 Wall-clock benchmarks

Ideally, reducing the memory footprint during training should not come at the expense of significant computational overhead. To ensure this is indeed the case, we implemented the multi-channel randomized trace estimation optimized for CPUs in Julia (Bezanson et al., 2017) and for GPUs in PyTorch (Paszke et al., 2019). Implementation details and benchmarks are included in Appendix A.4.

Our benchmarking experiments show competitive performance on both CPUs, against NNLib (Innes et al., 2018; Innes, 2018), and GPUs, against optimized CUDA implementations. On CPUs, we observe speedups of up to 10× over the standard *im2col* (Chellapilla et al., 2006) implementation for large images and large batch sizes, provided the number of probing vectors remains relatively small. On GPUs, we remain competitive

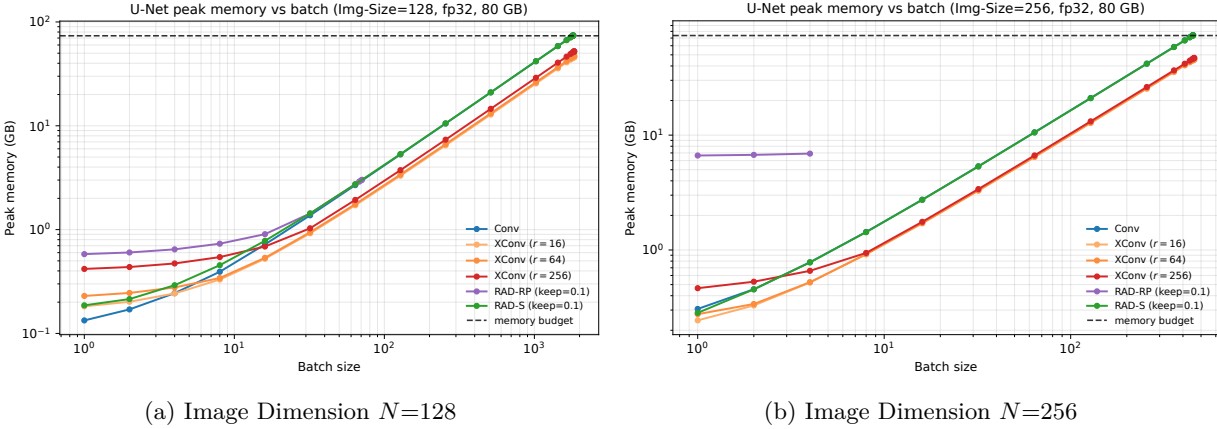

(a) Image Dimension $N$=128

(b) Image Dimension $N$=256

Figure 9: Peak memory curves for U-Net in single precision, as a function of batch size, at image dimensions $N \in \{128, 256\}$. We compare standard convolution, XConv at $r \in \{16, 64, 256\}$, and the RAD-RP and RAD-S baselines at keep fraction 0.1; the horizontal dashed line is the 80 GB memory budget. Despite U-Net's extensive skip connections and deep encoder–decoder structure, XConv tracks the lowest memory profile and admits the largest batch size before reaching the budget.

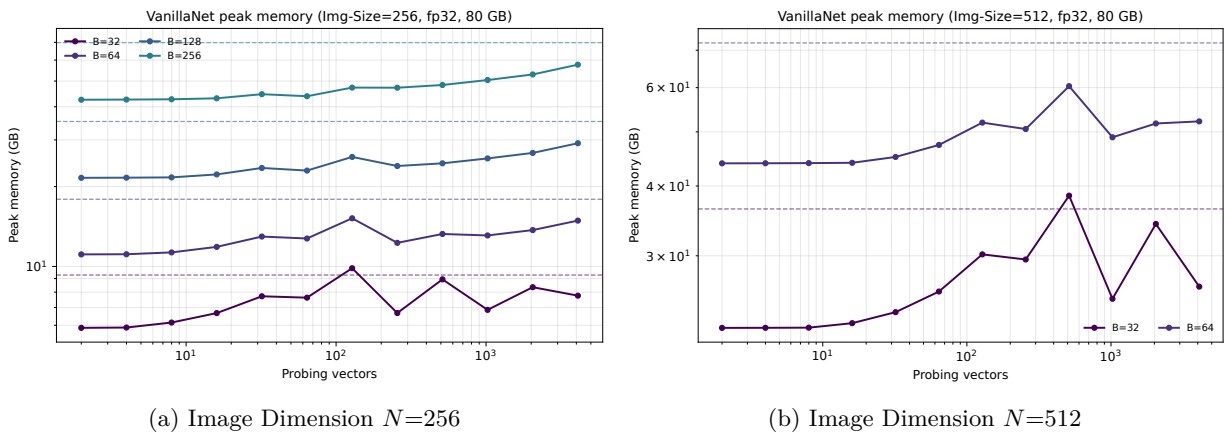

(a) Image Dimension $N$=256

(b) Image Dimension $N$=512

Figure 10: Peak memory curves for VanillaNet in single precision, as a function of the number of probing vectors, at image dimensions $N \in \{256, 512\}$ under an 80 GB memory budget. Each solid curve is a fixed batch size $B$ and the dashed line of the matching color is the corresponding standard convolution. Owing to the adaptive application of randomized trace estimation, peak memory does not exhibit a strictly monotonic dependence on the number of probing vectors $r$; nevertheless, across a broad range of probing budgets XConv remains below the standard convolution baseline.

with CuDNN kernels (Chetlur et al., 2014) and occasionally outperform them, though there is room for further optimization. In all cases, there is a slight decrease in computational throughput when the number of channels increases. Overall, approximate gradient calculations with multi-channel randomized trace estimation substitute expensive convolutions between the input and output channels with a relatively simple combination of matrix-free actions of the outer product on random probing vectors on the right and dense linear matrix operations on the left (cf. equation 8 and Algorithm 1).

The wall-clock benchmark results along with the hardware description can be found in Figures 11 and 12.

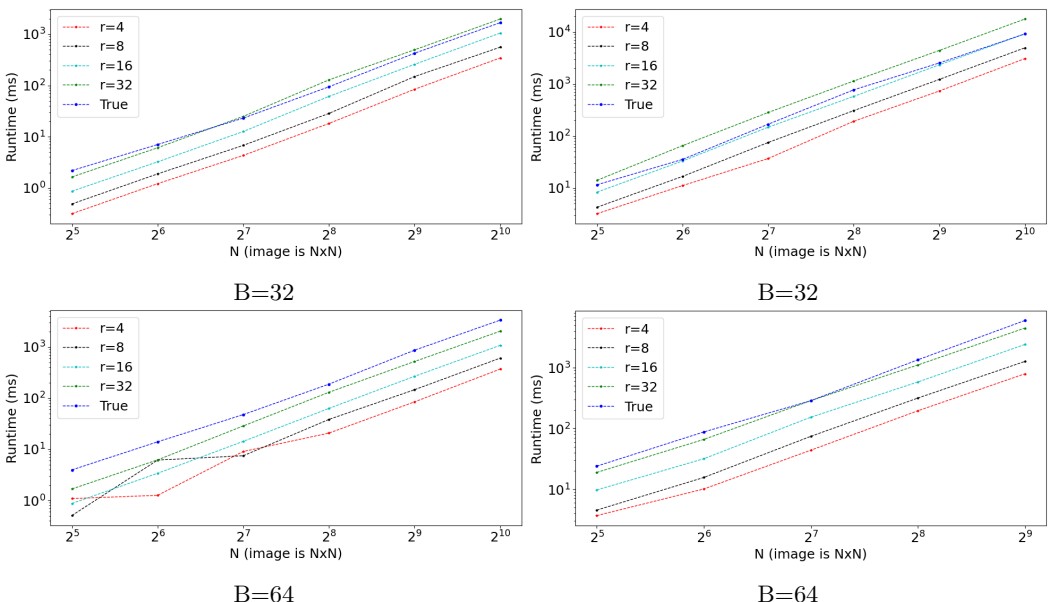

Figure 11: CPU benchmark on an *Intel(R) Xeon(R) CPU E3-1270 v6 @ 3.80GHz* node. The left column contains the runtimes for 4 channels and the right column for 32 channels. For large images and batch sizes, our implementation achieves speedups of up to $10\times$. Additional CPU benchmarks for smaller batch sizes can be found in Figure 29.

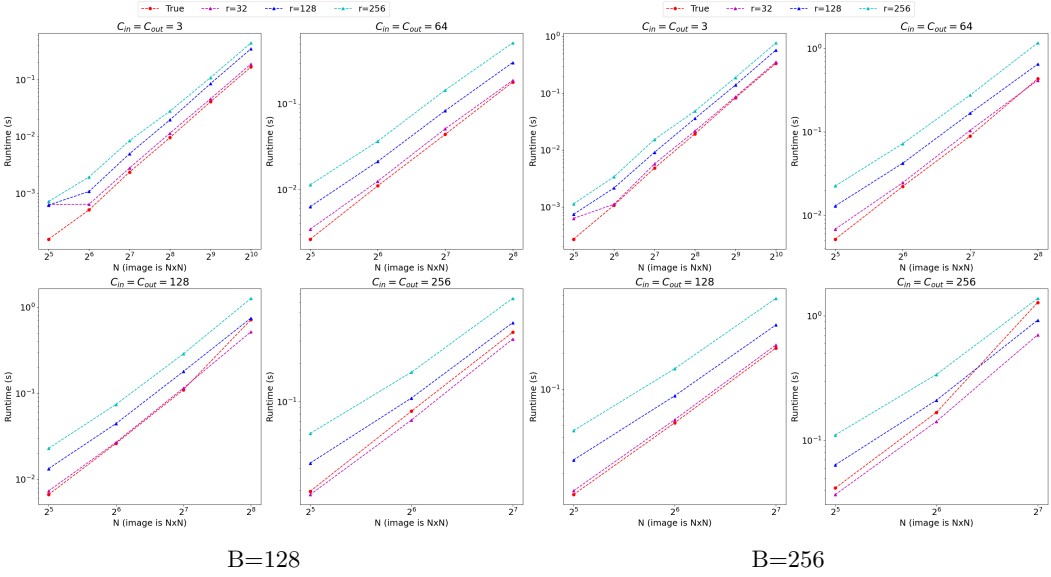

Figure 12: GPU benchmark on a *RTX 2000 Ada Generation* for a single gradient for varying batch sizes $B$, image sizes $N$ and number of channels $C_{\mathrm{in}} = C_{\mathrm{out}}$. At larger scale—higher channel dimensions and image sizes—we match or exceed state-of-the-art CuDNN kernels; at smaller scale CuDNN remains faster. Additional GPU benchmarks for smaller batch sizes can be found in Figure 30.

Table 1: Test accuracy for varying batch sizes $B$ and number of probing vectors $r$ on the MNIST dataset.

|           | $B = 64$ | $B = 128$ | $B = 256$ |
|-----------|----------|-----------|-----------|
| True      | 0.9905   | 0.9898    | 0.9901    |
| $r = 2$   | 0.9625   | 0.9692    | 0.9745    |
| $r = 16$  | 0.9753   | 0.9803    | 0.9823    |
| $r = 64$  | 0.9777   | 0.9723    | 0.9782    |
| $r = 256$ | 0.9718   | 0.9706    | 0.9791    |

## 5.5 Quantitative performance of XConv

Having quantified XConv's memory savings and gradient fidelity, we now evaluate whether these approximations meaningfully affect end-to-end model performance. We study classification, generative modeling, super-resolution, inpainting, and segmentation to determine whether the impact is consistent across fundamentally different learning objectives.

### 5.5.1 Classification

**MNIST dataset**  We design a simple neural network and evaluate classification performance on the MNIST dataset, varying the batch size $B$ and the number of probing vectors $r$. Implementations in both Julia and Python are evaluated.

We use the network architectures (detailed in Table 7 and 8 of Appendix A.5 for Julia and PyTorch, with training parameters listed in Appendix A.6) for varying batch sizes $B \in \{64, 128, 256\}$ and number of probing vectors $r \in \{2, 16, 64, 256\}$. The network test accuracies for the Julia implementation, where the default convolutional layer implementation is replaced by **XConv.jl**, are listed in Table 1 for the default implementation and for our implementation where gradients of the convolutional layers are replaced by our approximations. The results show that our low-memory implementation remains competitive even for a small number of probing vectors, yielding a memory saving of about 2.5×. We note that accuracy does not increase monotonically with $r$; the small fluctuations across probing vector settings are within the range of training stochasticity and do not indicate systematic degradation.

We obtained the results listed in Table 1 with the ADAM (Kingma & Ba, 2015) optimization algorithm. To further test robustness, we also train with stochastic line searches (SLS, (Vaswani et al., 2019)), which set learning rate parameters automatically at the cost of an extra gradient calculation. The full accuracy-vs.-epoch curves (Figure 35 in Appendix A.9) confirm that the induced randomness does not adversely affect the line searches: XConv achieves competitive accuracy for $r \geq 16$ across all batch sizes $B \in \{64, \dots, 2048\}$, yielding a 2.5× memory reduction.

**CIFAR-10 dataset**  We now compare standard stochastic gradient descent with approximate gradient methods based on multi-channel randomized trace estimation on the CIFAR-10 dataset.

To ensure a fair comparison, we fix the memory usage between the regular gradient, and the approximate gradients obtained by probing independently ("Indep." in green with $r = 32$), multi-channel ("Multi." in red with $r = 256$), and multi-channel with orthogonalization ("Multi-Ortho" in orange with $r = 256$). The batch size for the approximate gradient examples is increased from $B = 128$ to $B = 256$ to reflect the smaller memory footprint. Results for the training/testing loss and accuracy are included in Figure 13. The following observations can be made from these plots. First, there is a clear gap between the training/testing loss for the true and approximate gradients. This gap is also present in the training/testing accuracy, albeit relatively small. However, doubling the batch size effectively halves the training runtime.

**Facies classification benchmark**  Building on the image-classification benchmarks above, we next evaluate XConv on a more demanding dense-prediction task: the seismic Facies Classification Benchmark (Alaudah et al., 2019), which assigns a geological-facies label to every pixel of a seismic section. The benchmark exhibits

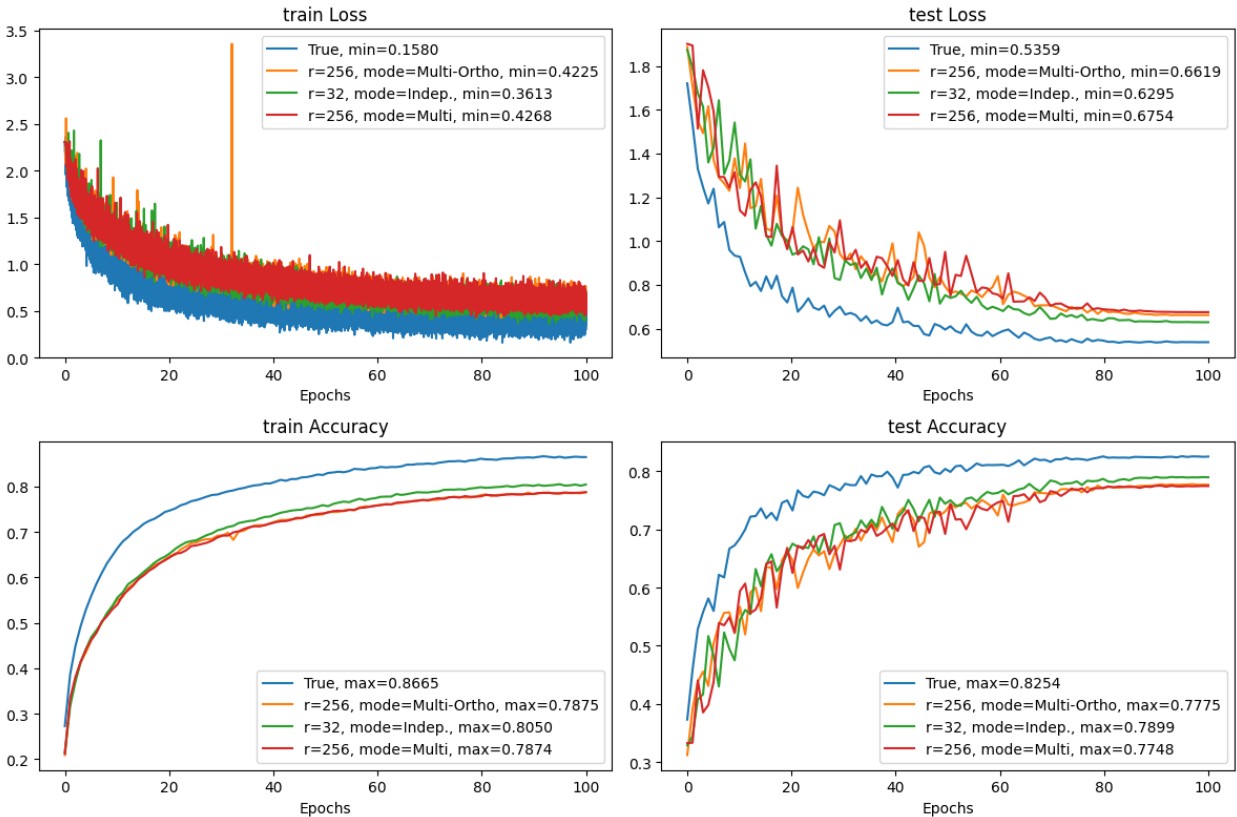

Figure 13: CIFAR-10 training with equivalent memory budget comparing standard convolution ($B$=128) to XConv with independent probing ($r$=32, green), multi-channel ($r$=256, red), and multi-channel orthogonalized probing ($r$=256, orange) at $B$=256. Top row: training and test loss. Bottom row: training and test accuracy after 100 epochs.

strong spatial structure, severe class imbalance, and large input dimensions, making it a challenging setting for convolutional architectures.

Our objective is to determine whether the memory savings provided by XConv can be obtained without substantially degrading dense-prediction performance. Following the protocol used throughout the paper, we fix the memory budget and vary the probing rank $r$. Figure 14 reports the average gradient error as a function of $r$, which decreases as the probing rank increases; based on this tradeoff we select $r = 128$ for the downstream evaluation.

We report Pixel Accuracy (PA), Mean Class Accuracy (MCA), Frequency-Weighted IoU, and Mean IoU. Table 2 compares standard convolution and XConv: at $r = 128$, XConv attains a Mean IoU of 0.473 against 0.538 for the exact-gradient baseline, with the other metrics following the same trend. The residual degradation is consistent with the heightened sensitivity of dense-prediction tasks to gradient approximation discussed in our failure-regime analysis.

Table 2: Facies Classification Benchmark: comparison between standard convolution and XConv ($r$=128) across Pixel Accuracy (PA), Mean Class Accuracy (MCA), Frequency-Weighted IoU (FWIoU), and Mean IoU (mIoU). Higher values indicate better segmentation performance.

|  | PA | MCA | FWIoU | mIoU |
|---|---|---|---|---|
| Convolution | 0.826 | 0.613 | 0.700 | 0.538 |
| XConv ($r$=128) | 0.802 | 0.556 | 0.662 | 0.473 |

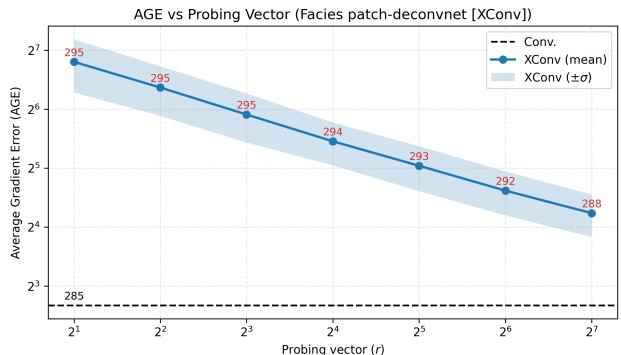

Figure 14: Facies classification: average gradient error (AGE) versus the number of probing vectors $r$. Consistent with the theoretical analysis, the AGE decreases monotonically as the number of probing vectors $r$ increases, approaching the convolutional baseline. The maximum permissible batch size remains largely unchanged, because convolution layers account for only a small fraction (25%) of the total memory footprint.

### 5.5.2 Generative modeling

While supervised classification provides a controlled setting to assess discriminative performance, generative modeling via diffusion places greater emphasis on optimization dynamics and is therefore more sensitive to gradient approximation. In this section, we evaluate whether the approximate gradients induced by XConv alter the generative performance of U-Net-based diffusion models.

**MNIST dataset** We consider a U-Net-based diffusion model trained on the MNIST dataset and evaluate the performance under varying numbers of probing vectors $r \in \{32, 64, 128, 256\}$. All experiments use a fixed batch size of 768, a learning rate of $10^{-3}$, and are trained for 200 epochs.

Although the average gradient error (AGE) introduced by XConv can be controlled through the number of probing vectors, it is unclear whether these approximation errors will propagate to downstream generative performance. We therefore investigate whether the increase in AGE observed for a small number of probing vectors results in measurable degradation in sample quality.

Training and validation loss curves (Appendix A.10, Figure 36) show that XConv exhibits optimization dynamics similar to those of the exact convolution baseline across all tested probing ranks. Despite the use of approximate gradients, the training trajectories remain stable and closely follow those of standard convolution.

To quantitatively evaluate generation quality, we report Fréchet Inception Distance (FID) in Figure 16. Consistent with the AGE analysis, increasing the probing rank improves generative performance and progressively reduces the gap to the convolutional baseline. At $r = 256$, XConv achieves FID scores close to those obtained with exact convolution, indicating that the diffusion model remains robust to the gradient approximation introduced by XConv when sufficient probing vectors are used.

Interestingly, these results hold even though the average gradient error of the diffusion U-Net rises by up to an order of magnitude relative to exact convolution (Figure 37), suggesting that U-Net-based diffusion models can tolerate moderate levels of gradient approximation error, particularly when the number of probing vectors is sufficiently large.

**CIFAR-10 Dataset** While MNIST provides an initial validation of XConv on generative modeling, its simplicity makes it difficult to assess the impact of gradient approximation on more realistic generative modeling tasks. We therefore evaluate XConv on CIFAR-10 (Krizhevsky et al., 2009), which contains 60000 RGB images of size $32 \times 32$ distributed across 10 semantic classes.

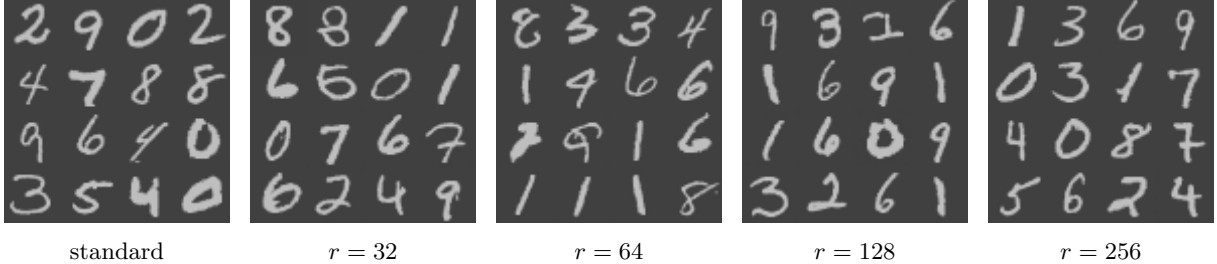

| standard | $r = 32$ | $r = 64$ | $r = 128$ | $r = 256$ |

Figure 15: Generated samples from the standard network and XConv. Quantitative evaluation using Fréchet Inception Distance (FID) in Figure 16 shows that increasing the number of probing vectors progressively recovers the performance of the convolutional baseline, with $r = 256$ achieving FID scores close to those of exact convolution.

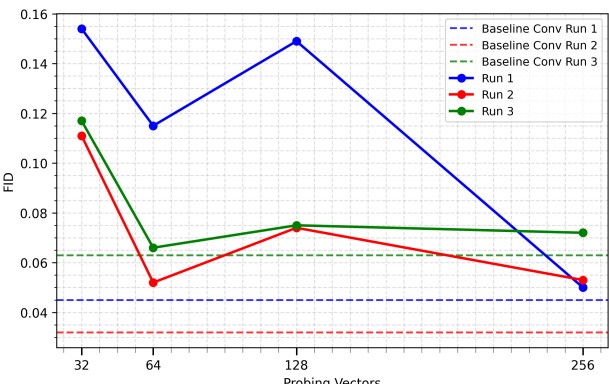

Figure 16: Fréchet Inception Distance (FID; lower is better) across three runs for the U-Net diffusion model on MNIST. Dashed lines denote FID scores of the standard convolution baseline. At $r = 256$, XConv achieves FID comparable to standard convolution.

Following the protocol used throughout the paper, we fix the memory budget and vary the number of probing vectors ($r$). Figure 17 reports the resulting average gradient error (AGE), which decreases as the number of probing vectors ($r$) increases. Based on this tradeoff, we select $r = 128$ for subsequent experiments.

We evaluate XConv-UNet using both qualitative samples and FID results. Figure 18 presents representative samples generated by the standard convolutional and XConv-based diffusion models. Consistent with the AGE analysis, the XConv-based model produces images of comparable visual quality. The corresponding FID scores (38.208 for convolution versus 40.175 for XConv at $r = 128$) suggest that the approximation error introduced by XConv has only a limited impact on generative performance on CIFAR-10.

**Seismic Dataset**  Unlike the previous image-based settings, the seismic dataset offers a distinct and higher-dimensional benchmark for generative modeling. The Parihaka survey (Veritas, 2005; WesternGeco., 2012) is a three-dimensional subsurface volume imaging the seafloor and the underlying rock strata; we train a diffusion model to generate $128 \times 128$ seismic sections extracted from this volume, an input dimensionality well beyond that of MNIST or CIFAR-10. As in the preceding generative experiments, all convolutional layers of the U-Net are replaced by XConv.

Both the standard-convolution and XConv diffusion models generate seismic sections whose layered geological texture is visually consistent with the recorded data (Figure 19). The gradient-fidelity behavior mirrors the other architectures—i.e., the average gradient error decreases steadily with the number of probing vectors toward the exact-convolution floor (Figure 20), while XConv simultaneously admits a larger training batch than standard convolution under the same memory budget. These results indicate that the memory–fidelity tradeoff established on natural images carries over to higher-dimensional scientific-imaging data.

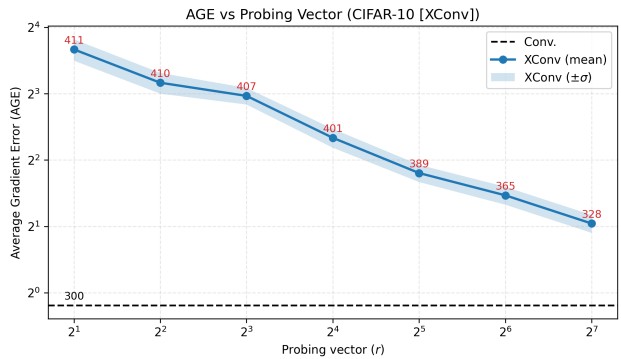

Figure 17: CIFAR-10: average gradient error (AGE) vs. the number of probing vectors $r$. Under a fixed memory budget, the gradient fidelity of XConv increases with an increasing number of probing vectors.

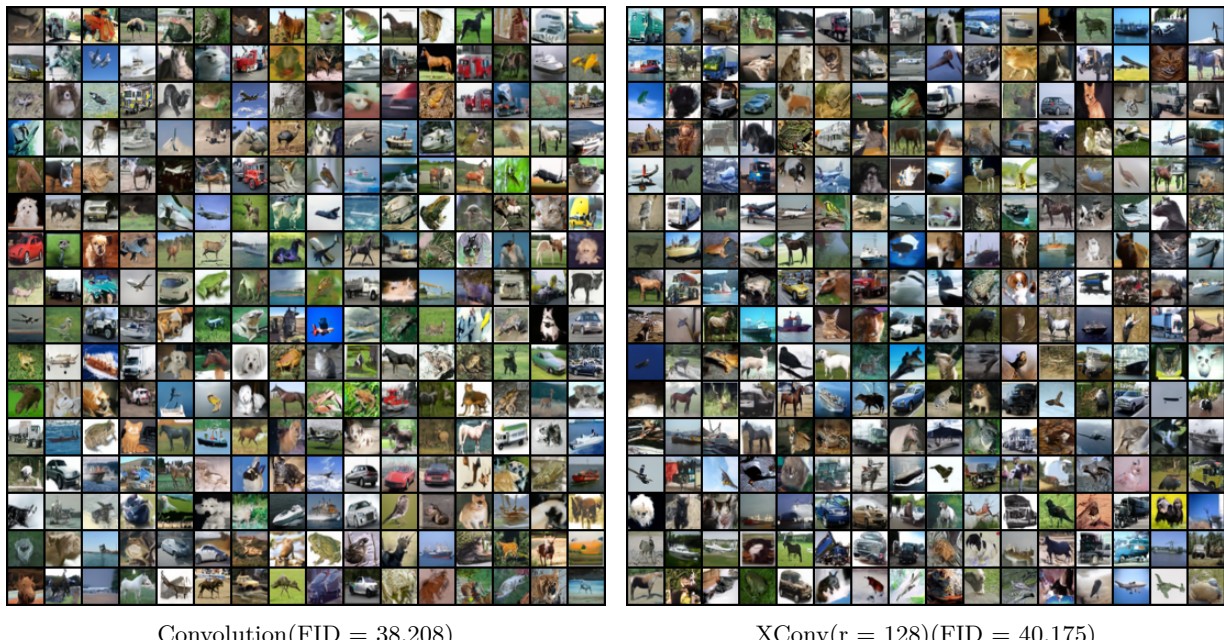

Convolution(FID = 38.208)  XConv(r = 128)(FID = 40.175)

Figure 18: Generated CIFAR-10 samples from the standard convolutional network and XConv ($r = 128$). The images generated by XConv-UNet attain an FID of 40.175, modestly above that of the standard convolutional network (FID = 38.208).

These experiments suggest that XConv-based diffusion models can accommodate moderate gradient approximation while largely preserving convergence and sample quality, provided the number of probing vectors is sufficiently large.

### 5.5.3 Super-resolution and inpainting

To evaluate performance on inverse problems, we adopt the Deep Image Prior (DIP) framework (Ulyanov et al., 2018), which leverages the inductive bias of convolutional networks to recover structure from corrupted observations without external training data. Since DIP derives its effectiveness from the structure of the convolutional network itself, it provides a stringent test of whether XConv preserves the optimization behavior of standard convolutions.

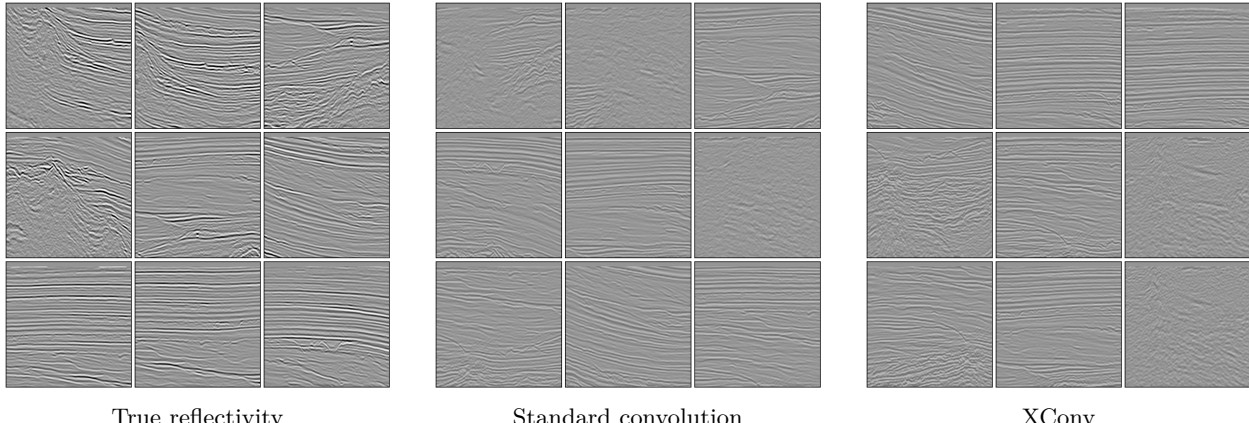

True reflectivity    Standard convolution    XConv

Figure 19: Recorded $128 \times 128$ Parihaka seismic sections (true reflectivity, left) and sections generated by the standard-convolution (center) and XConv (right) diffusion models. Both models reproduce the layered geological texture of the recorded data.

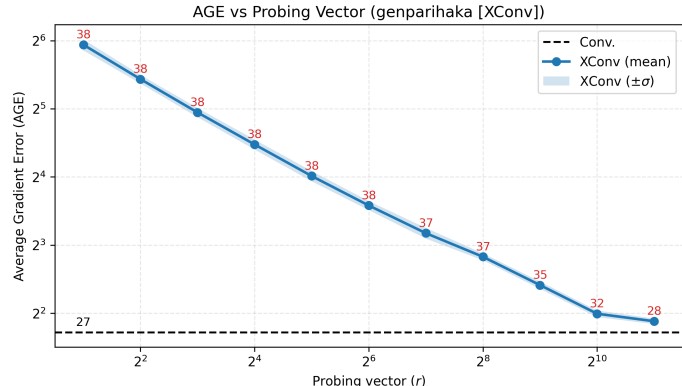

Figure 20: Average gradient error versus the number of probing vectors $r$ for the seismic diffusion model. XConv (mean $\pm \sigma$) approaches the exact-convolution floor (dashed) as $r$ increases; the annotated values are the largest batch size that fits the memory budget, which is larger for XConv than for standard convolution. The maximum permissible batch size remains nearly unchanged across different numbers of probing vectors because convolutional layers constitute only 22.5% of the overall network.

Our goal is to determine whether the memory savings offered by XConv can be achieved without substantially degrading reconstruction quality in DIP-based inverse problems. We consider both super-resolution and inpainting under fixed memory budgets.

Unlike classification or segmentation, DIP relies entirely on the optimization trajectory of a randomly initialized network. Consequently, the stochastic gradient approximation introduced by XConv may alter the implicit regularization that enables successful image reconstruction. It is therefore unclear whether the reconstruction quality achieved by standard convolutions can be maintained under approximate gradient computation.

Figure 31a in the appendix illustrates the tradeoff between reconstruction quality and peak memory consumption for super-resolution. Increasing the number of probing vectors reduces gradient approximation error and improves reconstruction fidelity, while also increasing memory consumption. Based on this tradeoff, we select $r = 256$ for super-resolution experiments. Quantitative PSNR results in Table 3a and qualitative examples in Figure 21 show that XConv achieves reconstruction quality close to that of standard convolution.

| (a) Super-resolution PSNR (dB). | | | | | (b) Inpainting PSNR (dB). | | | |
|---|---|---|---|---|---|---|---|---|
| Image | Conv | XConv($r = 256$) | Bicubic | Bilinear | Method | Kate | Vase | Memory(MiB) ↓ |
| Bird | 29.895 | 28.951 | 28.848 | 27.354 | Conv | 38.090 | 28.356 | 3351.5 |
| Head | 29.464 | 29.495 | 28.436 | 28.471 | XConv ($r = 512$) | 38.921 | 28.788 | 2719.2 (-18.9%) |

Table 3: Quantitative Deep Image Prior results. (a) Super-resolution performance measured using PSNR (dB). (b) Inpainting results on the Kate and Vase images. XConv ($r = 512$) reduces peak memory consumption by $\approx 19\%$ relative to standard convolution while maintaining competitive reconstruction quality. These results demonstrate that XConv can provide meaningful memory savings in image restoration tasks while maintaining competitive or better PSNR.

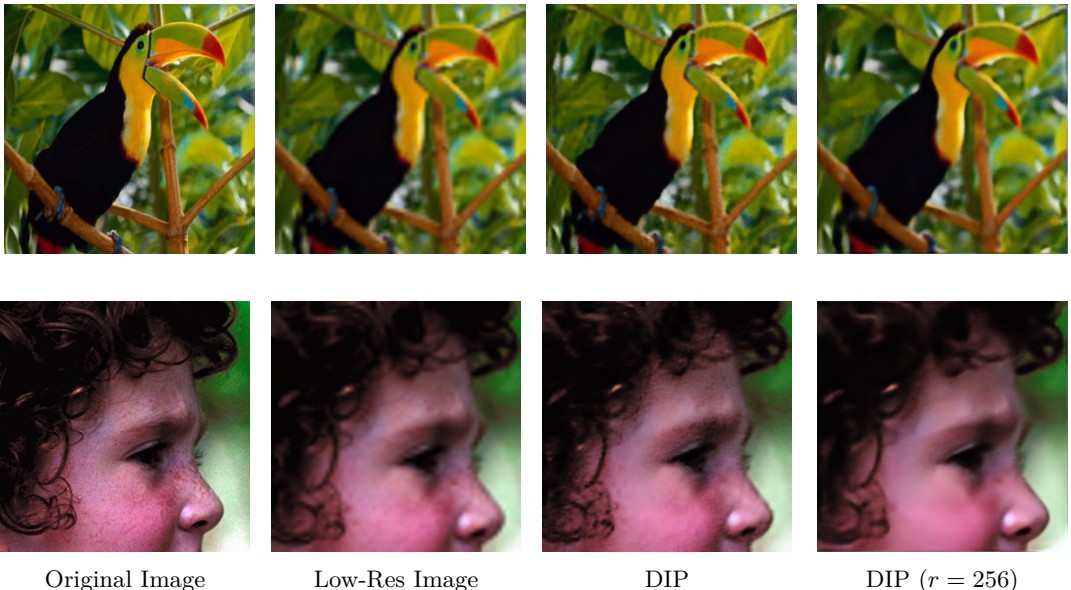

|            |            |     |            |
|:----------:|:----------:|:---:|:----------:|
| Original Image | Low-Res Image | DIP | DIP ($r = 256$) |

Figure 21: Qualitative super-resolution results comparing Conv and XConv. For both *head* and *bird*, XConv (r = 256) reconstructed images preserve the dominant edges, textures, and fine-scale structures recovered by the convolutional baseline.

For inpainting, Figure 31b in the appendix shows that XConv remains substantially more memory-efficient than standard convolution across a broad range of probing vectors. We therefore select $r = 512$, the largest number of probing vectors that remains within the convolutional memory budget. The qualitative inpainting results in Figure 22 and the quantitative results in Table 3b demonstrate that XConv maintains competitive reconstruction quality while reducing peak memory consumption by approximately 19%.

Across both inverse problems, larger numbers of probing vectors consistently improve reconstruction fidelity, mirroring the reduction in average gradient error observed throughout the paper. These results suggest that XConv preserves the optimization behavior required for DIP-based image restoration while providing meaningful memory savings relative to exact convolution.

### 5.5.4 Segmentation

Beyond image-level prediction and generative modeling, segmentation requires accurate dense prediction at every pixel and is therefore substantially more sensitive to gradient perturbations. To assess XConv in this setting, we consider gland segmentation on the GlaS dataset (Sirinukunwattana et al., 2017), which contains 165 histological images at a resolution of $256 \times 256$.

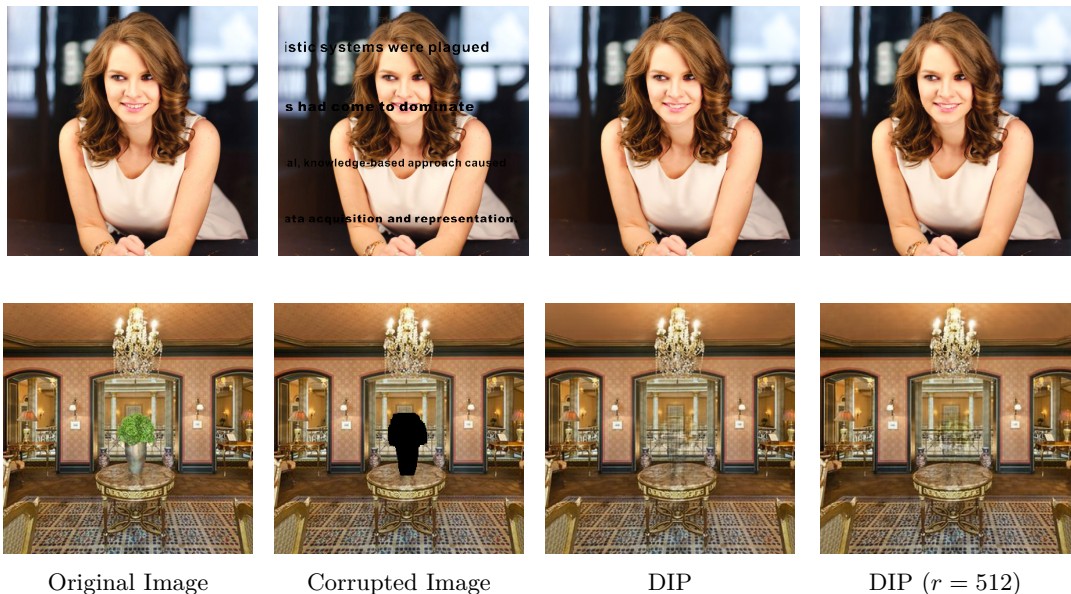

| Original Image | Corrupted Image | DIP | DIP ($r = 512$) |

Figure 22: Qualitative inpainting results comparing Conv and XConv. XConv ($r = 512$) reconstructs the missing regions while preserving the dominant image structure recovered by standard convolution.

Our objective is to determine whether the memory savings offered by XConv can be achieved without substantially degrading segmentation performance. We adopt TriConvUNeXt (Ma et al., 2024a), a lightweight segmentation architecture combining dilated and deformable convolutions, and replace all standard convolution layers with XConv while keeping all hyperparameters fixed.

As discussed throughout the paper, reducing the probing rank $r$ decreases memory consumption but increases the average gradient error (AGE). While the resulting approximation error can be quantified analytically, it is unclear whether these gradient perturbations will propagate to downstream dense-prediction performance. Figure 23a reports AGE as a function of the number of probing vectors $r$. Consistent with the theoretical analysis, increasing $r$ reduces both the magnitude and variance of the gradient approximation error. The maximum permissible batch size remains unchanged across varying probing vectors, as the convolutional layers constitute 23.63% of the network and therefore contribute a relatively small fraction of the total memory consumption. Figure 23b shows the corresponding peak memory consumption. While larger probing ranks improve gradient fidelity, they also increase memory requirements, resulting in a clear accuracy–memory tradeoff. Notably, the marginal reduction in AGE diminishes beyond $r = 1024$, whereas memory consumption continues to increase.

Based on this tradeoff, we select $r = 1024$ for subsequent segmentation experiments. Quantitative results in Table 4 show that XConv achieves a Dice score of 0.900 compared to 0.905 for standard convolution, while classification accuracy decreases only marginally from 0.904 to 0.898. These differences are below 1%, indicating that XConv largely preserves segmentation performance despite replacing exact convolution gradients with randomized trace-based approximations.

Figure 24 provides representative qualitative predictions. The segmentation masks produced by XConv closely match those obtained using standard convolution and accurately recover the dominant gland structures present in the ground-truth annotations.

The GlaS experiments demonstrate that XConv can be successfully applied to dense prediction tasks without substantial loss in segmentation accuracy. More importantly, the downstream performance trend mirrors the AGE analysis: larger numbers of probing vectors reduce gradient approximation error and recover performance closer to the convolutional baseline. These results provide empirical evidence that the gradient fidelity captured by AGE translates directly into improved segmentation performance while retaining the memory benefits of XConv.

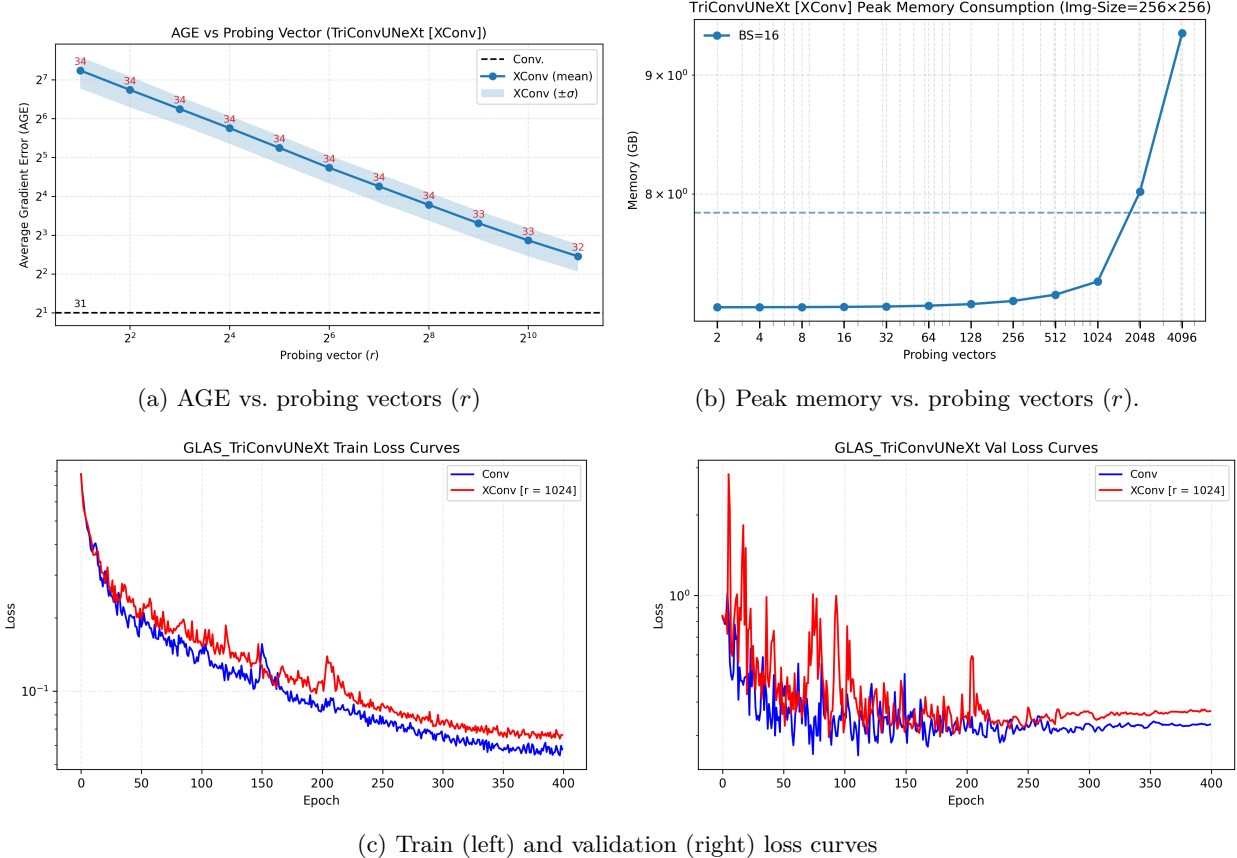

(a) AGE vs. probing vectors ($r$)

(b) Peak memory vs. probing vectors ($r$).

(c) Train (left) and validation (right) loss curves

Figure 23: AGE (top-left), peak memory consumption (top-right), and train-validation loss curves of the TriConvUNeXt model. The horizontal dashed line in both top graphs denotes standard convolution. Increasing the number of probing vectors $r$ reduces AGE while increasing memory consumption. The train and validation losses closely follow those of standard convolution, indicating that XConv preserves the optimization dynamics of the baseline model.

| Gradient | Dice | ACC |
|---|---|---|
| standard convolution | 0.905 | 0.904 |
| XConv ($r$=1024) | 0.900 | 0.898 |

Table 4: Quantitative segmentation results on the GlaS dataset with TriConvUNeXt. Dice measures the overlap between the ground-truth and predicted segmentation masks; higher values indicate better performance. Results comparing standard convolution to XConv with probing vectors ($r = 1024$). Despite approximate gradients, XConv achieves segmentation accuracy within 1% of Conv, consistent with the qualitative results in Figure 24.

These results suggest that, even for dense prediction tasks at high spatial resolution, XConv can maintain adequate optimization fidelity while reducing memory requirements.

### 5.5.5 Volumetric segmentation under finetuning

The experiments above train from scratch. A complementary and increasingly common regime is the finetuning of a pretrained model under a fixed memory budget—i.e., adapting a released checkpoint to a new dataset or on device, where activation memory rather than data throughput is the binding constraint. To evaluate XConv in this regime, we finetune the MONAI (Cardoso et al., 2022) `spleen_ct_segmentation`

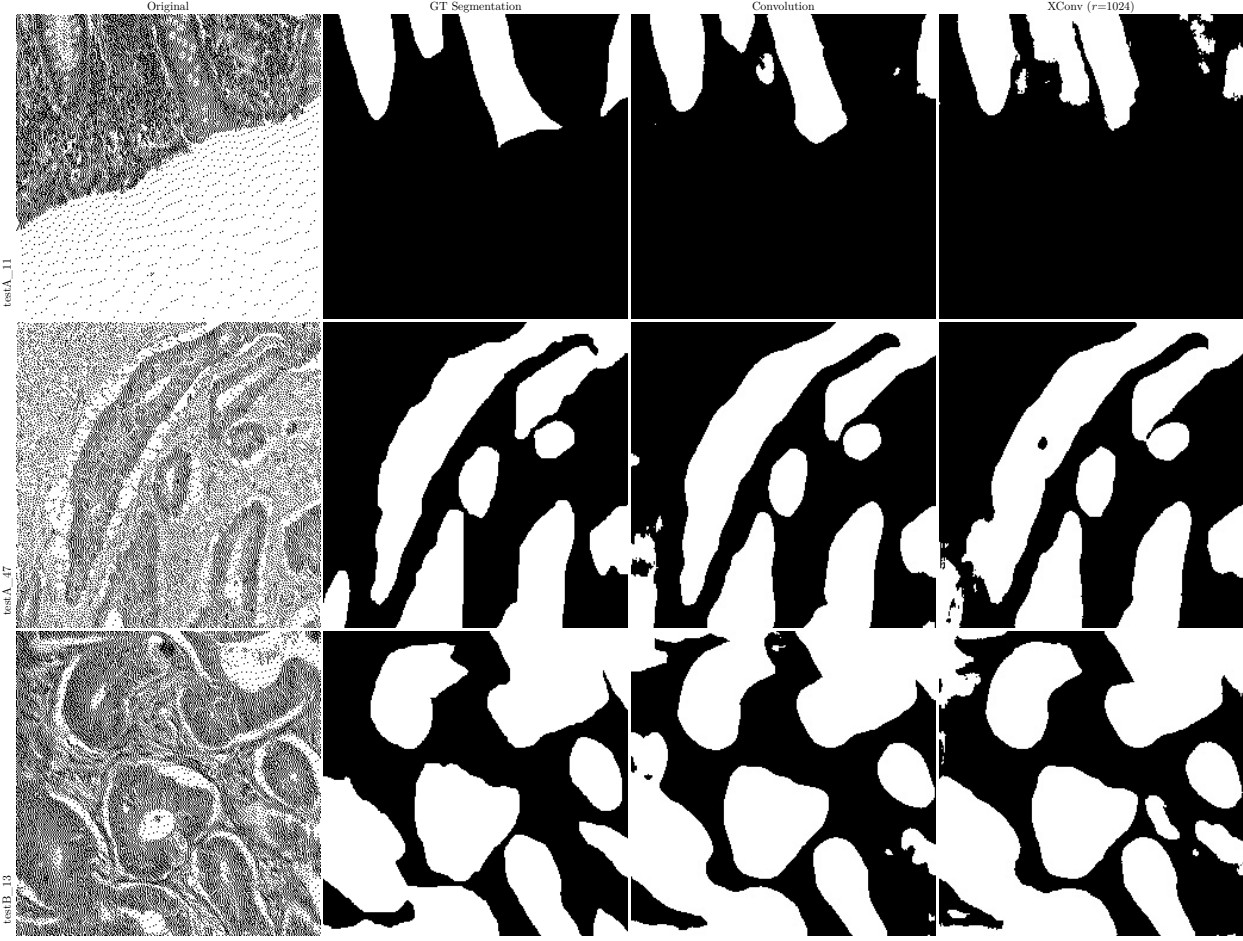

Figure 24: Qualitative segmentation results on the GlaS dataset comparing standard convolution and XConv. Each row corresponds to a different test image, while columns show the original image, ground-truth segmentation map, convolution, and XConv ($r$=1024) predictions. XConv ($r = 1024$) achieves a Dice score and classification accuracy within 1% of the standard convolution baseline, demonstrating that dense-prediction performance is largely preserved despite the use of approximate gradients.

model—a fully three-dimensional U-Net (a convolutional encoder–decoder with skip connections and feature widths 16–256 across five resolution levels, $3 \times 3 \times 3$ convolutions, PReLU activations, and three-dimensional batch normalization) that segments the spleen from abdominal CT volumes—on the Medical Segmentation Decathlon spleen task (Antonelli et al., 2022). Starting from the released pretrained weights, we continue training on $96^3$ voxel patches with the model's own recipe (Novograd optimizer, Dice–cross-entropy loss, and step learning-rate schedule), replacing every three-dimensional convolution with its XConv counterpart so that only the convolutional weight-gradient computation changes while every other component of the training pipeline is left unchanged.

Under an identical finetuning recipe—i.e., the same optimizer, learning rate, batch size, and number of training steps, so that only the gradient computation differs—even a small probing rank ($r$=4) finetunes the network to the same foreground Dice as the exact gradient (0.9622 versus 0.9620) while reducing the peak memory of the finetuning step by 17.5% (9775 versus 11854 MiB; Table 5). The magnitude of the reduction is governed by the share of the peak held by the convolutional activations—i.e., in this encoder–decoder the skip-connection working set, which XConv does not compress, sets a floor on the achievable saving, consistent with the layer-composition analysis of Section 5.3. The experiment nonetheless demonstrates that XConv supports memory-reduced finetuning of pretrained three-dimensional models without loss of accuracy. Beyond

| Gradient | Peak memory (MiB) | Foreground Dice |
|---|---|---|
| standard convolution | 11854 | 0.9620 |
| XConv ($r=4$) | 9775 ($-17.5\%$) | 0.9622 |

Table 5: Finetuning a pretrained volumetric U-Net (MONAI `spleen_ct_segmentation`, Medical Segmentation Decathlon spleen CT) under an identical recipe (batch size 64, 600 steps, learning rate $2 \times 10^{-4}$); only the gradient computation differs. At a small probing rank ($r=4$), the XConv gradient matches the exact-gradient foreground Dice while reducing the peak memory of the finetuning step by 17.5%.

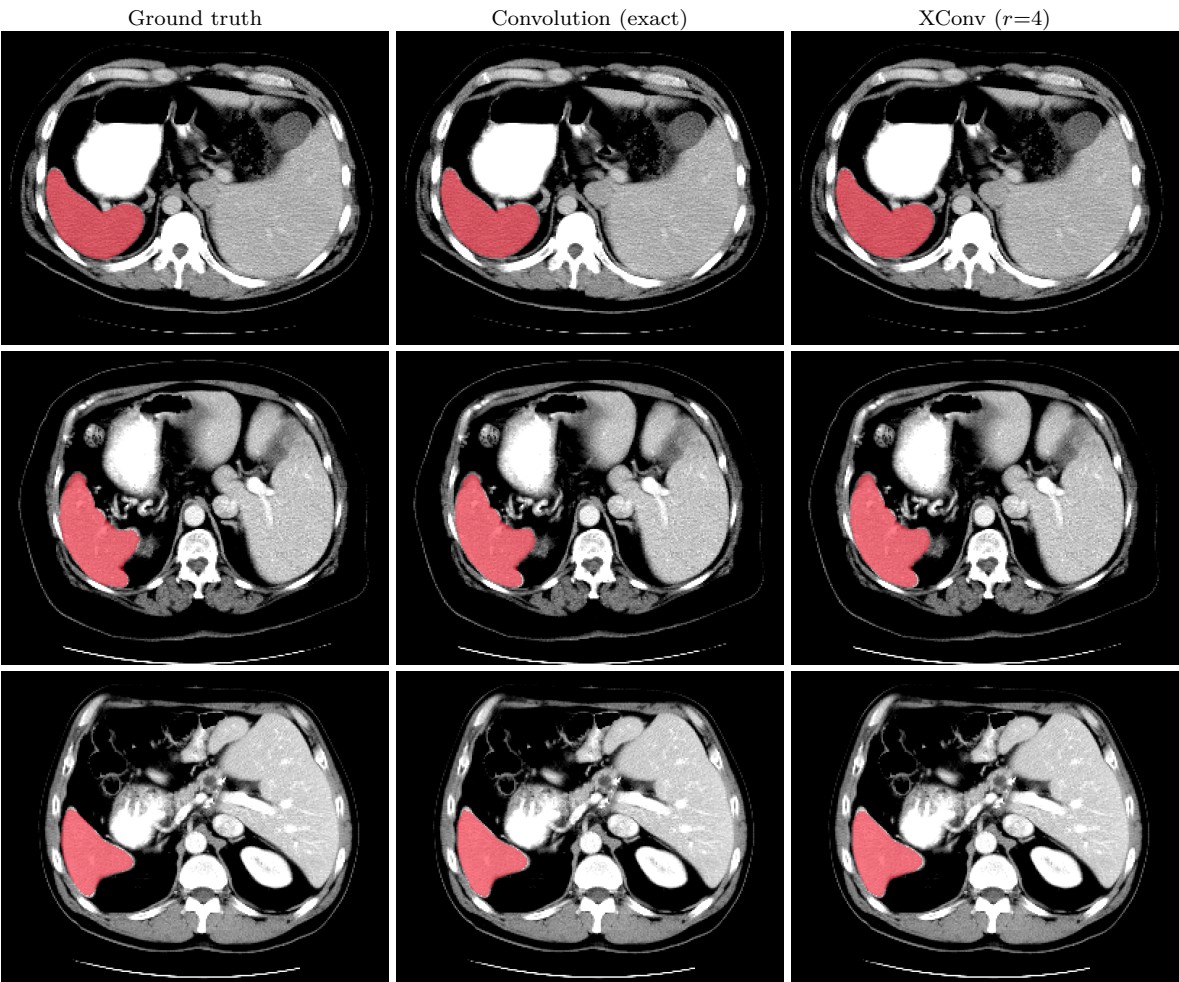

Figure 25: Qualitative spleen segmentation on three held-out CT volumes (one axial slice per volume): the ground-truth annotation (left), the exact-gradient standard-convolution prediction (center), and the XConv ($r=4$) prediction (right), where only the convolutional weight gradient differs between the two models. The probed-gradient segmentations are visually indistinguishable from the exact-gradient ones, with the per-volume foreground Dice of the two methods agreeing to within $\pm0.001$ on every held-out volume.

the aggregate Dice, the segmentations produced by the probed gradient are visually indistinguishable from those of the exact gradient on held-out volumes (Figure 25), with the per-volume foreground Dice of the two methods agreeing to within $\pm0.001$.

## 6 Related work

The memory pressure of backpropagation has motivated several lines of work. Checkpointing approaches (Griewank & Walther, 2000; Beaumont et al., 2019; Korthikanti et al., 2023) recompute activations during the backward pass, yielding exact gradients at the cost of additional computation. Invertible neural networks (Haber & Ruthotto, 2017; Jacobsen et al., 2018; Hascoet et al., 2023; Orozco et al., 2024) recover activations from outputs but impose architectural constraints that may limit expressiveness. Approximate arithmetic (Gupta et al., 2015; Xi et al., 2023) and memory-efficient optimizers (Zhao et al., 2024b) reduce memory through lower numerical precision or projected gradient updates, while local learning methods (Saikumar & Varghese, 2024) break inter-layer dependencies but require non-trivial architectural modifications. Direct feedback alignment (Nøkland, 2016; Han & Yoo, 2019; Frenkel et al., 2021) replaces backpropagated gradients with random projections. Randomized linear algebra techniques (Avron & Toledo, 2011; Martinsson & Tropp, 2020; Meyer et al., 2021) provide unbiased gradient approximations—a desirable property for stochastic optimization (Neelakantan et al., 2015). Among these, randomized automatic differentiation (RAD) (Oktay et al., 2021) is closest to our work but requires intervention in the computational graph. In contrast, XConv exploits the specific algebraic structure of convolutional layer gradients, enabling a simpler approach that acts as a drop-in replacement for 2D and 3D convolutional layers in existing frameworks.

## 7 Conclusion and future work

We introduced XConv, a memory-efficient convolutional layer that approximates gradients via multi-channel randomized trace estimation. By storing only compressed activations, XConv reduces the memory footprint while remaining computationally competitive with standard implementations. The approach comes with convergence guarantees and error bounds, and our experiments show that networks trained with XConv achieve performance close to exact-gradient methods on most tasks—with larger gaps on the most gradient-sensitive dense-prediction settings and very wide architectures, where a larger $r$ is needed—and accuracy that improves systematically as the number of probing vectors increases. The degree of memory savings and accuracy depends on the architecture and task, ranging from sub-$2\times$ when non-convolutional activations dominate to substantially larger savings for convolution-dominated, high-resolution workloads. These properties, combined with recent architectural innovations that reduce computational complexity (Howard et al., 2017; Liu et al., 2022; Chen et al., 2023a;b) and advances in specialized photonic hardware for randomized probing (Saade et al., 2016), open directions for scaling CNNs to higher-dimensional data such as video representation learning and other 3D applications. More broadly, the principle of approximating weight gradients via randomized trace estimation is not limited to convolutions—extending this approach to attention layers, where the memory footprint of stored activations is similarly prohibitive, is a promising direction for future work.

**Declaration of AI usage.** Claude (Anthropic) was used to improve the academic tone and technical clarity of the writing, correct grammatical errors and enhance readability, and ensure consistent formatting of bibliographic entries. All scientific content, including methodology, theoretical results, and experimental design, was produced entirely by the authors.

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

## A  Appendix

### A.1  Implementation and code availability

Our probing algorithm is implemented both in Julia, using `LinearAlgebra.BLAS` on CPU and `CUDA.CUBLAS` on GPU for the linear algebra computations, and in PyTorch using standard linear algebra utilities. The Julia interface is designed so that preexisting networks can be reused as we are overloading `rrule` (see ChainRulesCore.jl) to switch easily between the conventional true gradient (NNlib.jl) and ours. The PyTorch implementation defines a new layer that can be swapped for the conventional convolutional layer, `torch.nn.Conv2d` or `torch.nn.Conv3d`, in any network using the `convert_net` utility function. Both implementations support 2D and 3D convolutions, making XConv applicable to volumetric data such as medical imaging (demonstrated here) and, prospectively, video.

The software and examples will be made available under an MIT license upon publication.

## A.2 Proofs of Proposition 1 and Theorem 1

For a square matrix $\mathbf{A}$, let $G(\mathbf{A})$ be the trace estimator:

$$G(\mathbf{A}) = \frac{1}{r} \sum_{j=1}^{r} \mathbf{z}_j^\top \mathbf{A} \mathbf{z}_j$$

where $\mathbf{z}_j \sim \mathcal{N}(\mathbf{0}, \mathbf{I}_N)$ are i.i.d. Gaussian vectors. We now prove the proposition and theorem stated in Section 3.

### A.2.1 Proof of Proposition 1

We restate Proposition 1 here.

**Proposition 2.** *Let* $\mathbf{A} \in \mathbb{R}^{N \times N}$ *be a square matrix. Then for any small number* $\delta > 0$, *with probability* $1 - \delta$,

$$|G(\mathbf{A}) - \mathrm{tr}(\mathbf{A})| \leq \left( \frac{4\|\mathbf{A}\|_2}{r} \log \frac{2}{\delta} + \frac{2\|\mathbf{A}\|_F}{\sqrt{r}} \log^{1/2} \frac{2}{\delta} \right).$$

The proof uses the following result on trace estimation of symmetric matrices.

**Lemma 2** (Theorem 5 of (Cortinovis & Kressner, 2022)). *Let* $\mathbf{B} \in \mathbb{R}^{N \times N}$ *be symmetric. Then*

$$P(|G(\mathbf{B}) - \mathrm{tr}(\mathbf{B})| \geq \epsilon) \leq 2 \exp \left( -\frac{r\epsilon^2}{4\|\mathbf{B}\|_F^2 + 4\epsilon\|\mathbf{B}\|_2} \right)$$

*for all* $\epsilon > 0$.

*Proof of Proposition 2.* For a symmetric matrix $\mathbf{B}$, Lemma 2 immediately implies that for any small number $\delta > 0$, with probability $1 - \delta$,

$$|G(\mathbf{B}) - \mathrm{tr}(\mathbf{B})| \leq \frac{4\|\mathbf{B}\|_2}{r} \log \frac{2}{\delta} + \frac{2\|\mathbf{B}\|_F}{\sqrt{r}} \log^{1/2} \frac{2}{\delta}.$$

Now for our asymmetric $\mathbf{A}$, let $\mathbf{B} = \frac{\mathbf{A} + \mathbf{A}^\top}{2}$. Then $G(\mathbf{A}) = G(\mathbf{B})$, $\mathrm{tr}(\mathbf{A}) = \mathrm{tr}(\mathbf{B})$, $\|\mathbf{B}\|_2 \leq \|\mathbf{A}\|_2$, and $\|\mathbf{B}\|_F \leq \|\mathbf{A}\|_F$, then the proposition follows. $\qquad\square$

### A.2.2 Preparation lemmas for Theorem 1

**Lemma 3.** *Let* $\mathbf{A} \in \mathbb{R}^{N \times N}$ *be a square matrix,* $\mathbf{z}_j, \mathbf{x}_j \sim \mathcal{N}(\mathbf{0}, \mathbf{I}_N)$ *be random Gaussian vectors for* $j = 1, \ldots, r$, *and all the* $\mathbf{x}_j$ *and* $\mathbf{z}_j$ *are independent of each other. Then for any* $\delta > 0$, *with probability* $1 - \delta$,

$$\left| \frac{1}{r} \sum_{j=1}^{r} \mathbf{z}_j^\top \mathbf{A} \mathbf{x}_j \right| \leq c \left( \frac{\|\mathbf{A}\|_2}{r} \log \frac{2}{\delta} + \frac{\|\mathbf{A}\|_F}{\sqrt{r}} \log^{1/2} \frac{2}{\delta} \right)$$

*where* $c$ *is some absolute constant independent of* $r$.

*Proof of Lemma 3.* Set $T := \frac{1}{r} \sum_{j=1}^{r} \mathbf{z}_j^\top \mathbf{A} \mathbf{x}_j$. For each summand, we have

$$\mathbf{z}_j^\top \mathbf{A} \mathbf{x}_j = \mathbf{z}_j^\top \mathbf{U} \mathbf{S} \mathbf{V}^\top \mathbf{x}_j = \tilde{\mathbf{z}}_j^\top \mathbf{S} \tilde{\mathbf{x}}_j = \sum_t s_t \tilde{\mathbf{z}}_j[t] \tilde{\mathbf{x}}_j[t] \equiv \sum_t s_t f_{j,t}, \tag{10}$$

where the first equality used the singular value decomposition $\mathbf{A} = \mathbf{U} \mathbf{S} \mathbf{V}^\top$, in the second equality, we defined $\tilde{\mathbf{z}}_j = \mathbf{U}^\top \mathbf{z}_j$ and $\tilde{\mathbf{x}}_j = \mathbf{V}^\top \mathbf{x}_j$, which are still Gaussian. In the third equality, we used $s_t$ to denote the $t^{\text{th}}$

diagonal entry of $\mathbf{S}$ and $\tilde{\mathbf{z}}_j[t]$ and $\tilde{\mathbf{x}}_j[t]$ to denote the $t^{\text{th}}$ entry of $\tilde{\mathbf{z}}_j$ and $\tilde{\mathbf{x}}_j$, respectively. In the last equality, we defined $f_{j,t} := \tilde{\mathbf{z}}_j[t]\tilde{\mathbf{x}}_j[t]$. Since $\mathbf{z}_j$ and $\mathbf{x}_j$ are i.i.d., so are $f_{j,t}$. And since $f_{j,t}$ are products of independent sub-Gaussian random variables, they obey the sub-exponential distribution, i.e.,

$$\|f_{j,t}\|_{\psi_1} \leq \|\tilde{\mathbf{z}}_j[t]\|_{\psi_2}\|\tilde{\mathbf{x}}_j[t]\|_{\psi_2} = c^2,$$

where $\|\cdot\|_{\psi_1}$ denotes the sub-exponential norm and $\|\cdot\|_{\psi_2}$ the sub-Gaussian norm. We also used the property that there is a constant $c$, such that for any $\sigma$, a Gaussian variable $a \sim \mathcal{N}(0, \sigma^2)$ has a sub-Gaussian norm $\|a\|_{\psi_2} \leq c\sigma$, and this property is applied on $\tilde{\mathbf{z}}_j[t]$ and $\tilde{\mathbf{x}}_j[t]$ who are both $\mathcal{N}(0,1)$ variables due to the rotation invariance of Gaussian vectors.

Applying the Bernstein inequality (Vershynin, 2018) to $T = \frac{1}{r}\sum_{j,t} s_t f_{j,t}$, we obtain

$$P(|T| \geq \tilde{t}) \leq e^{-c' \min\{\frac{r\tilde{t}^2}{4\|\mathbf{A}\|_F^2}, 4\frac{r\tilde{t}}{\|\mathbf{A}\|_2}\}},$$

where $c'$ is some absolute constant. Letting $\delta$ be the right-hand-side probability, the above implies

$$P\left(|T| \geq c\left(\frac{\|\mathbf{A}\|_2}{r}\log\frac{2}{\delta} + \frac{\|\mathbf{A}\|_F}{\sqrt{r}}\log^{1/2}\frac{2}{\delta}\right)\right) \leq \delta,$$

with some constant $c$. Then the lemma is proved. $\qquad\square$

**Lemma 4.** *Let $\mathbf{A} \in \mathbb{R}^{N \times N}$ be a square matrix, $\mathbf{z}_j, \mathbf{x}_j \sim \mathcal{N}(\mathbf{0}, \mathbf{I}_N)$ be random Gaussian vectors for $j = 1, \ldots, r$, and all the $\mathbf{x}_j$'s and $\mathbf{z}_j$'s are independent of each other. Let $\mathbf{y}_j$ be the random vector that equals $\mathbf{x}_j$ with probability $p$ and equals $0$ with probability $1 - p$. Then for any $\delta > 0$ with probability over $1 - \delta - 2e^{-rp^2/2}$,*

$$\left|\frac{1}{r}\sum_{j=1}^{r} \mathbf{z}_j^\top \mathbf{A}\mathbf{y}_j\right| \leq c\left(\frac{\|\mathbf{A}\|_2}{r}\log\frac{2}{\delta} + \frac{\sqrt{p}\|\mathbf{A}\|_F}{\sqrt{r}}\log^{1/2}\frac{2}{\delta}\right),$$

*where $c$ is some absolute constant independent of $r$.*

*Proof.* The proof is very similar to that of Lemma 3. Set $T := \frac{1}{r}\sum_{j=1}^{r} \mathbf{z}_j^\top \mathbf{A}\mathbf{y}_j$. Let $g_j = 1_{\{\mathbf{y}_j \neq \mathbf{0}\}}$ be the indicator function of whether $\mathbf{y}_j$ is non-zero. Then clearly $\mathbf{y}_j = \mathbf{x}_j g_j$. For each summand, we have

$$\mathbf{z}_j^\top \mathbf{A}\mathbf{y}_j = \mathbf{z}_j^\top \mathbf{U}\mathbf{S}\mathbf{V}^\top \mathbf{x}_j g_j = \tilde{\mathbf{z}}_j^\top \mathbf{S}\tilde{\mathbf{x}}_j g_j = \sum_t s_t \tilde{\mathbf{z}}_j[t]\tilde{\mathbf{x}}_j[t]g_j \equiv g_j \sum_t s_t f_{j,t}, \qquad (11)$$

where in the first equality, we used the singular value decomposition, $\mathbf{A} = \mathbf{U}\mathbf{S}\mathbf{V}^\top$, and in the second equality, we defined $\tilde{\mathbf{z}}_j = \mathbf{U}^\top \mathbf{z}_j$ and $\tilde{\mathbf{x}}_j = \mathbf{V}^\top \mathbf{x}_j$. In the third equality, we used $s_t$ to denote the $t^{\text{th}}$ diagonal entry of $\mathbf{S}$ with $\tilde{\mathbf{z}}_j[t]$ and $\tilde{\mathbf{x}}_j[t]$ denoting the $t^{\text{th}}$ entry of $\tilde{\mathbf{z}}_j$ and $\tilde{\mathbf{x}}_j$, respectively. In the last equality, we defined $f_{j,t} := \tilde{\mathbf{z}}_j[t]\tilde{\mathbf{x}}_j[t]$. From Lemma 3, $f_{j,t}$ follow sub-exponential distributions, i.e., $\|f_{j,t}\|_{\psi_1} \leq c^2$.

Conditional on $g_j$, applying the Bernstein inequality to $\tilde{T} := \frac{1}{\sum_j 1_{\{g_j \neq 0\}}}\sum_{j,t} s_t f_{j,t}g_j$, we obtain

$$P(|\tilde{T}| \geq \tilde{t}) \leq e^{-c' \min\left\{\frac{\tilde{t}^2 \sum_i 1_{\{g_i \neq 0\}}}{4\|\mathbf{A}\|_F^2}, 4\frac{\tilde{t}\sum_j 1_{\{g_j \neq 0\}}}{\|\mathbf{A}\|_2}\right\}}.$$

Letting $\delta$ be the right-hand-side probability, the above implies

$$P\left(|\tilde{T}| \geq c\left(\frac{\|\mathbf{A}\|_2}{\sum_j 1_{\{g_j \neq 0\}}}\log\frac{2}{\delta} + \frac{\|\mathbf{A}\|_F}{\sqrt{\sum_j 1_{\{g_j \neq 0\}}}}\log^{1/2}\frac{2}{\delta}\right)\right) \leq \delta.$$

Since $\sum_j 1_{\{g_j \neq 0\}} \sim \mathbf{B}(r, p)$, we then have with probability at least $1 - 2e^{-rp^2/2}$, $3rp/2 \geq \sum_j 1_{\{g_j \neq 0\}} \geq rp/2$. Plugging this estimate into the above, we have

$$P\left(|\tilde{T}| \geq c\left(\frac{\|\mathbf{A}\|_2}{pr} \log \frac{2}{\delta} + \frac{\|\mathbf{A}\|_F}{\sqrt{pr}} \log^{1/2} \frac{2}{\delta}\right)\right) \leq \delta.$$

Then, with this bound of $|\tilde{T}|$, we have

$$|T| = \left|\frac{1}{r}\sum_{j=1}^{r} \mathbf{z}_j^\top \mathbf{A}\mathbf{y}_j\right| = \left|\frac{\sum_j 1_{\{g_j \neq 0\}}}{r}\tilde{T}\right| \leq c\left(\frac{\|\mathbf{A}\|_2}{r}\log\frac{2}{\delta} + \frac{\sqrt{p}\|\mathbf{A}\|_F}{\sqrt{r}}\log^{1/2}\frac{2}{\delta}\right),$$

which is the statement of this lemma. $\qquad\square$

### A.2.3 Multi-channel result: Proof of Theorem 1

**Index convention.** In this proof, $G^{m,n}$ denotes the estimator for $\mathrm{tr}(\mathbf{A}^{m,n})$, with the first superscript indexing output channels and the second indexing input channels. In the main text (Theorem 1, succinct version), the superscript order is reversed: $\widetilde{G}^{n,m}$ with the first index for input channels and the second for output channels. The two conventions are equivalent up to relabeling.

For simplicity, we assume the number of input and output channels are the same and both equal to $C$. Let

$$\mathbf{A} = \begin{pmatrix} \mathbf{A}^{1,1} & \cdots & \mathbf{A}^{1,C} \\ \vdots & & \vdots \\ \mathbf{A}^{C,1} & \cdots & \mathbf{A}^{C,C} \end{pmatrix}$$

the goal is to estimate $\mathrm{tr}(\mathbf{A}^{m,n})$, for $m, n = 1, \ldots, C$.

Let $\mathbf{Z} \in \mathbb{R}^{NC \times r}$ be the "orthogonalized" matrix of $r$ probing vectors. We further denote by $\mathbf{z}_{n,\cdot}$ the $n^{\text{th}}$ row block of $\mathbf{Z}$, which is the block containing the $((n-1)N+1)^{th}$ to the $(nN)^{th}$ rows of $\mathbf{Z}$. We also denote by $\mathbf{z}_j \in \mathbb{R}^{NC}$, $j = 1, \ldots, r$ the $j^{\text{th}}$ column of $\mathbf{Z}$ (i.e., the $j^{\text{th}}$ probing vector), and by $\mathbf{z}_{n,j}$ the $n^{\text{th}}$ block of $\mathbf{z}_j$. For any $n = 1, \ldots, C$, $j = 1, \ldots, r$, we define $\mathbf{z}_{n,j}$ as

$$\mathbf{z}_{n,j} \sim \begin{cases} \mathcal{N}(\mathbf{0}, \mathbf{I}_N), & \text{with probability } p_n \\ 0, & \text{with probability } 1 - p_n \end{cases}. \tag{12}$$

For different values of $(n, j)$, $\mathbf{z}_{n,j}$ are independent of each other. Here $p_n$ is a predefined probability for randomly generating each nonzero block.

With these probing vectors, we define the following estimator for $\mathrm{tr}(\mathbf{A}^{m,n})$

$$G^{m,n}(\mathbf{A}) := \frac{1}{\mathrm{nnz}(\mathbf{z}_{n,\cdot})}\sum_{j=1}^{r} \mathbf{z}_{n,j}^\top (\mathbf{A}\mathbf{z}_j)_m,$$

where $\mathrm{nnz}(\mathbf{z}_{n,\cdot})$ is the number of nonzeros columns of $\mathbf{z}_{n,\cdot}$, which is also a random variable.

**Theorem 1.** *Let $p = \min\limits_{n} p_n$, $r$ be the number of probing vectors. For any small number $\delta > 0$, with probability at least $1 - \delta - 3Ce^{-rp^2/2}$, we have for any $n, m = 1, \ldots, C$,*

$$|G^{m,n}(\mathbf{A}) - \mathrm{tr}(\mathbf{A}^{m,n})| \leq c\left(\frac{\sum_{k=1}^{C}\|\mathbf{A}^{m,k}\|_2}{p_n r}\log\frac{C^2}{\delta} + \frac{\frac{1}{\sqrt{p_n}}\|\mathbf{A}^{m,n}\|_F + \sum_{j=1,j\neq n}^{C}\sqrt{\frac{p_j}{p_n}}\|\mathbf{A}^{m,j}\|_F}{\sqrt{r}}\log^{1/2}\frac{C^2}{\delta}\right),$$

*where $c$ is an absolute constant and $C$ is the number of channels. For sufficiently large number of probing vectors, the above bound reduces to*

$$|G^{m,n}(\mathbf{A}) - \mathrm{tr}(\mathbf{A}^{m,n})| \le c \cdot \frac{\frac{1}{\sqrt{p_n}}\|\mathbf{A}^{m,n}\|_F + \sum_{j=1,j\neq n}^{C}\sqrt{\frac{p_j}{p_n}}\|\mathbf{A}^{m,j}\|_F}{\sqrt{r}} \log^{1/2}\frac{C^2}{\delta}.$$

*Proof.* First we show the estimator is unbiased. For simplicity of notation, let $g_{n,l} = 1_{\{\mathbf{z}_{n,l}\neq \mathbf{0}\}}$ be the random variable that indicates whether $\mathbf{z}_{n,l}$ is non-zero. By definition, conditional on $g_{n,l} = 1$, $\mathbf{z}_{n,l}$ is Gaussian, and this Gaussian distribution is independent of $g_{n,l}$. Also $\sum_l g_{n,l} \sim \mathbf{B}(r,p_n)$ is Binomial distribution with probability $p_n$. Then the estimator can be written as

$$G^{m,n}(\mathbf{A}) := \frac{1}{\sum_l g_{n,l}}\sum_{l=1}^{r}\mathbf{z}_{n,l}^{\top}(\mathbf{A}\mathbf{z}_l)_m g_{n,l}.$$

Taking the expectation, we have

$$\mathbb{E}(G^{m,n}(\mathbf{A})) = \mathbb{E}_g\left[\mathbb{E}_{\mathbf{Z}|g}\left(\frac{1}{\sum_l g_{n,l}}\sum_{l=1}^{r}\mathbf{z}_{n,l}^{\top}(\mathbf{A}\mathbf{z}_l)_m g_{n,l}\right)\right]$$

$$= \mathbb{E}_g\left[\frac{1}{\sum_l g_{n,l}}\mathbb{E}_{\mathbf{Z}|g}\left(\sum_{l=1}^{r}\sum_{k=1}^{C}\mathbf{z}_{n,l}^{\top}\mathbf{A}^{m,k}\mathbf{z}_{k,l}g_{n,l}\right)\right]$$

$$= \mathbb{E}_g\left[\frac{1}{\sum_l g_{n,l}}\sum_{l=1}^{r}\sum_{k=1}^{C}\mathbb{E}_{\mathbf{Z}|g}\left(\mathrm{tr}(\mathbf{A}^{m,k}\mathbf{z}_{k,l}\mathbf{z}_{n,l}^{\top})g_{n,l}\right)\right]$$

$$= \mathbb{E}_g\left[\frac{1}{\sum_l g_{n,l}}\sum_{l=1}^{r}\sum_{k=1}^{C}\mathrm{tr}(\mathbf{A}^{m,k}\mathbb{E}_{\mathbf{Z}|g}(\mathbf{z}_{k,l}\mathbf{z}_{n,l}^{\top}))g_{n,l}\right]$$

$$= \mathbb{E}_g\left[\frac{1}{\sum_l g_{n,l}}\sum_{l=1}^{r}\mathrm{tr}(\mathbf{A}^{m,n})g_{n,l}\right]$$

$$= \mathrm{tr}(\mathbf{A}^{m,n}),$$

where the second to last equality used $\mathbb{E}_{\mathbf{Z}|g}(\mathbf{z}_{n,l}\mathbf{z}_{n,l}^{\top}) = g_{n,l}\mathbf{I}_N$ and $\mathbb{E}_{\mathbf{Z}|g}(\mathbf{z}_{k,l}\mathbf{z}_{n,l}^{\top}) = 0$ for $k \neq n$. Then we estimate the large deviation,

$$G^{m,n}(\mathbf{A}) - \mathrm{tr}(\mathbf{A}^{m,n}) = \frac{1}{\sum_l g_{n,l}}\sum_{l=1}^{r}\sum_{k=1}^{C}\mathbf{z}_{n,l}^{\top}\mathbf{A}^{m,k}\mathbf{z}_{k,l}g_{n,l} - \mathrm{tr}(\mathbf{A}^{m,n})$$

$$= \frac{1}{\sum_l g_{n,l}}\sum_{l=1}^{r}\mathbf{z}_{n,l}^{\top}\mathbf{A}^{m,n}\mathbf{z}_{n,l}g_{n,l} - \mathrm{tr}(\mathbf{A}^{m,n}) + \sum_{k=1,k\neq n}^{C}\left(\frac{1}{\sum_l g_{n,l}}\sum_{l=1}^{r}\mathbf{z}_{n,l}^{\top}\mathbf{A}^{m,k}\mathbf{z}_{k,l}g_{n,l}\right).$$

The first term is bounded by Proposition 2, and the second term is bounded by Lemma 4. Explicitly, $\sum_l g_{n,l}$ is the number of probing vectors we are using in probing $G^{m,n}(\mathbf{A})$, so conditional on $g_{n,l}$, Proposition 2 yields an upper bound on the first term in the above right hand side, with probability $1 - \delta'$

$$\left|\frac{1}{\sum_l g_{n,l}}\sum_{l=1}^{r}\mathbf{z}_{n,l}^{\top}\mathbf{A}^{m,n}\mathbf{z}_{n,l}g_{n,l} - \mathrm{tr}(\mathbf{A}^{m,n})\right| \le \frac{4\|\mathbf{A}^{m,n}\|_2}{\sum_l g_{n,l}}\log\frac{2}{\delta'} + \frac{2\|\mathbf{A}^{m,n}\|_F}{\sqrt{\sum_l g_{n,l}}}\log^{1/2}\frac{2}{\delta'}. \tag{13}$$

By Lemma 4, the bound on the second term is, with probability $1 - \delta' - 2Ce^{-rp^2/2}$,

$$\left|\sum_{k=1,k\neq n}^{C}\left(\frac{1}{\sum_l g_{n,l}}\sum_{l=1}^{r}\mathbf{z}_{n,l}^{\top}\mathbf{A}^{m,k}\mathbf{z}_{k,l}g_{n,l}\right)\right| \le \sum_{k=1,k\neq n}^{C}\left|\frac{1}{\sum_l g_{n,l}}\sum_{l=1}^{r}\mathbf{z}_{n,l}^{\top}\mathbf{A}^{m,k}\mathbf{z}_{k,l}g_{n,l}\right|$$

$$\le c\sum_{k=1,k\neq n}^{C}\left(\frac{\|\mathbf{A}^{m,k}\|_2}{\sum_l g_{n,l}}\log\frac{2}{\delta'} + \frac{\sqrt{p_k}\|\mathbf{A}^{m,k}\|_F}{\sqrt{\sum_l g_{n,l}}}\log^{1/2}\frac{2}{\delta'}\right). \tag{14}$$

Since $\sum_l g_{n,l} \sim \mathbf{B}(r, p_n)$, so with probability at least $1 - Ce^{-rp_n^2/2}$, we have $\sum_l g_{n,l} > p_n r/2$. Combining this, equation 13, and equation 14 gives

$$|G^{m,n}(\mathbf{A}) - \text{tr}(\mathbf{A}^{m,n})| \leq c \left( \frac{\sum_{k=1}^{C} \|\mathbf{A}^{m,k}\|_2}{p_n r} \log \frac{2}{\delta'} + \frac{\frac{1}{\sqrt{p_n}} \|\mathbf{A}^{m,n}\|_F + \sum_{k=1,k\neq n}^{C} \sqrt{\frac{p_k}{p_n}} \|\mathbf{A}^{m,k}\|_F}{\sqrt{r}} \log^{1/2} \frac{2}{\delta'} \right).$$

For sufficiently large $r$, the second term is dominant and the bound reduces to

$$|G^{m,n}(\mathbf{A}) - \text{tr}(\mathbf{A}^{m,n})| \leq c' \frac{\frac{1}{\sqrt{p_n}} \|\mathbf{A}^{m,n}\|_F + \sum_{k=1,k\neq n}^{C} \sqrt{\frac{p_k}{p_n}} \|\mathbf{A}^{m,k}\|_F}{\sqrt{r}} \log^{1/2} \frac{1}{\delta'}.$$

So far the result is for a given pair of $m, n$. By the union bound of probability, the probability of failure for any $m, n$ is $\delta = \delta' C^2$. Then with probability at least $1 - \delta - 3Ce^{-rp^2/2}$, we have

$$|G^{m,n}(\mathbf{A}) - \text{tr}(\mathbf{A}^{m,n})| \leq c \cdot \frac{\frac{1}{\sqrt{p_n}} \|\mathbf{A}^{m,n}\|_F + \sum_{k=1,k\neq n}^{C} \sqrt{\frac{p_k}{p_n}} \|\mathbf{A}^{m,k}\|_F}{\sqrt{r}} \log^{1/2} \frac{C^2}{\delta}.$$

$\square$

### A.3 Average Gradient Error

We report the full set of average gradient error curves versus the number of probing vectors $r$ for all three models, in both single (fp32) and half (fp16) precision, under the 80 GB memory budget. SqueezeNet is shown in Figure 26, VanillaNet in Figure 27, and U-Net in Figure 28. The half-precision panels are omitted for U-Net because at fp16 the gradient error of every method collapses below the rounding floor, which makes the comparison uninformative.

Table 6: Convolutional channel widths in VanillaNet. The per-layer product $C_{\text{in}} \times C_{\text{out}}$ grows to $1.7 \times 10^7$ in the deepest stages, far exceeding the corresponding products in SqueezeNet and the U-Net; by Theorem 1 this larger channel product drives VanillaNet's higher single-precision average gradient error under a fixed probing budget.

| Layer group | $C_{\text{in}}$ | $C_{\text{out}}$ | $C_{\text{in}} \times C_{\text{out}}$ |
|---|---|---|---|
| stem | 512 | 512 | $2.6 \times 10^5$ |
| stages 0–1 | 512 | 512–1024 | $2.6$–$5.2 \times 10^5$ |
| stage 2 | 1024 | 1024–2048 | $1.0$–$2.1 \times 10^6$ |
| stages 3–5 | 2048 | 2048 | $4.2 \times 10^6$ |
| stage 6 | 2048 | 4096 | $8.4 \times 10^6$ |
| stage 7 | 4096 | 4096 | $1.7 \times 10^7$ |

### A.4 Computational Overhead

### A.5 Model Architectures

We describe here the network architectures used in our experiments. The architectures used in the MNIST experiments are standard architectures inspired by existing networks for this dataset. The CIFAR-10 architecture is intentionally chosen to be mostly convolutional and obtained from (Oktay et al., 2021).

### A.6 Training parameters

We now detail the training hyperparameters for the results presented in Section 5.

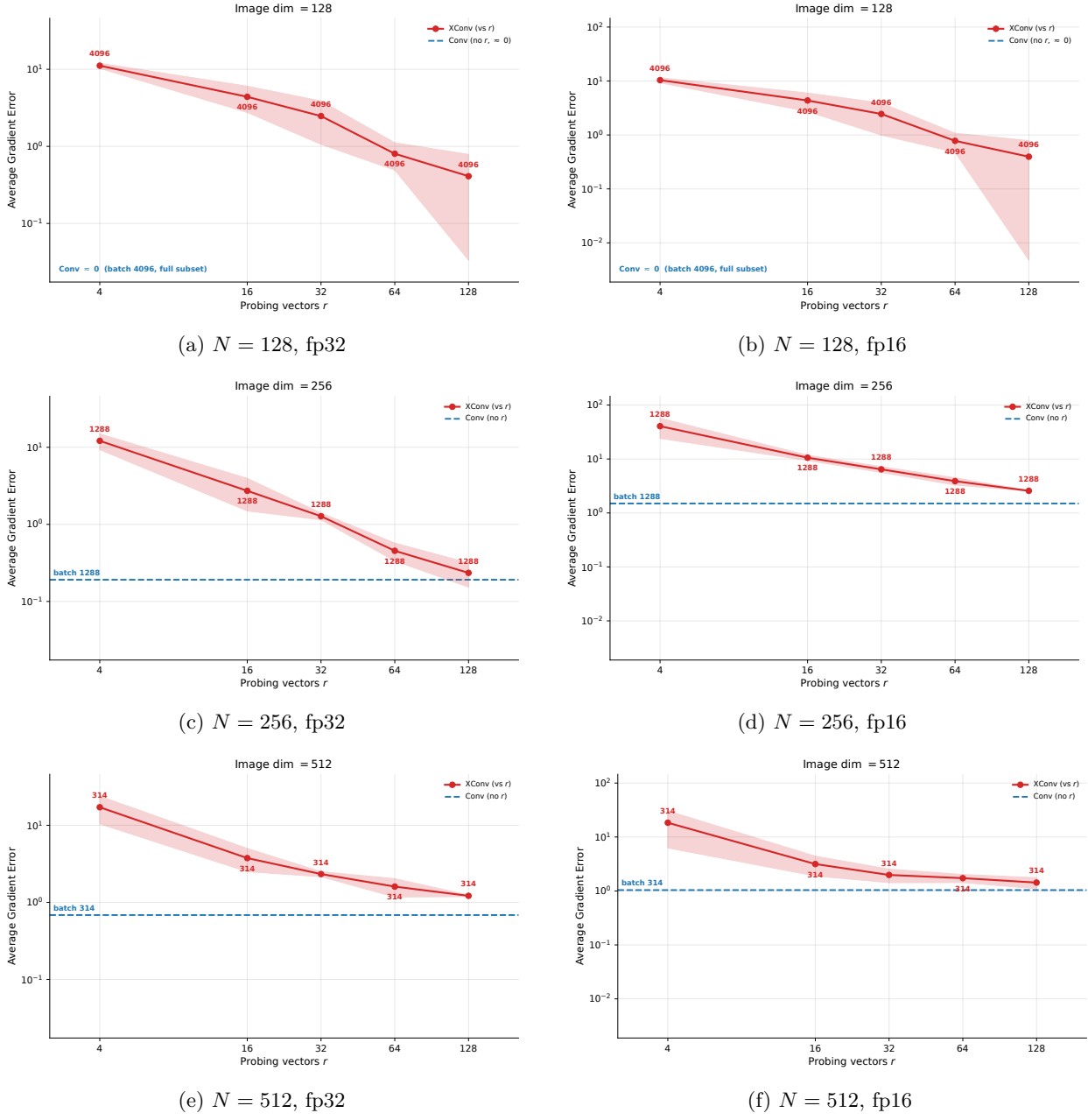

Figure 26: Average gradient error versus the number of probing vectors $r$ for SqueezeNet at image dimensions $N \in \{128, 256, 512\}$, in single precision (left) and half precision (right), under the 80 GB memory budget. The red curve is XConv and the blue dashed line the standard convolution floor. The gap closes with $r$ in single precision, and at half precision the XConv error coincides with the convolution floor.

### A.6.1 MNIST with Julia

- NVIDIA Quadro P1000 GPU
- 20 epochs
- Adam with initial learning rate of 0.003
- MNIST dataset for varying batch size $B$ and number of probing vectors $r$
- Julia implementation

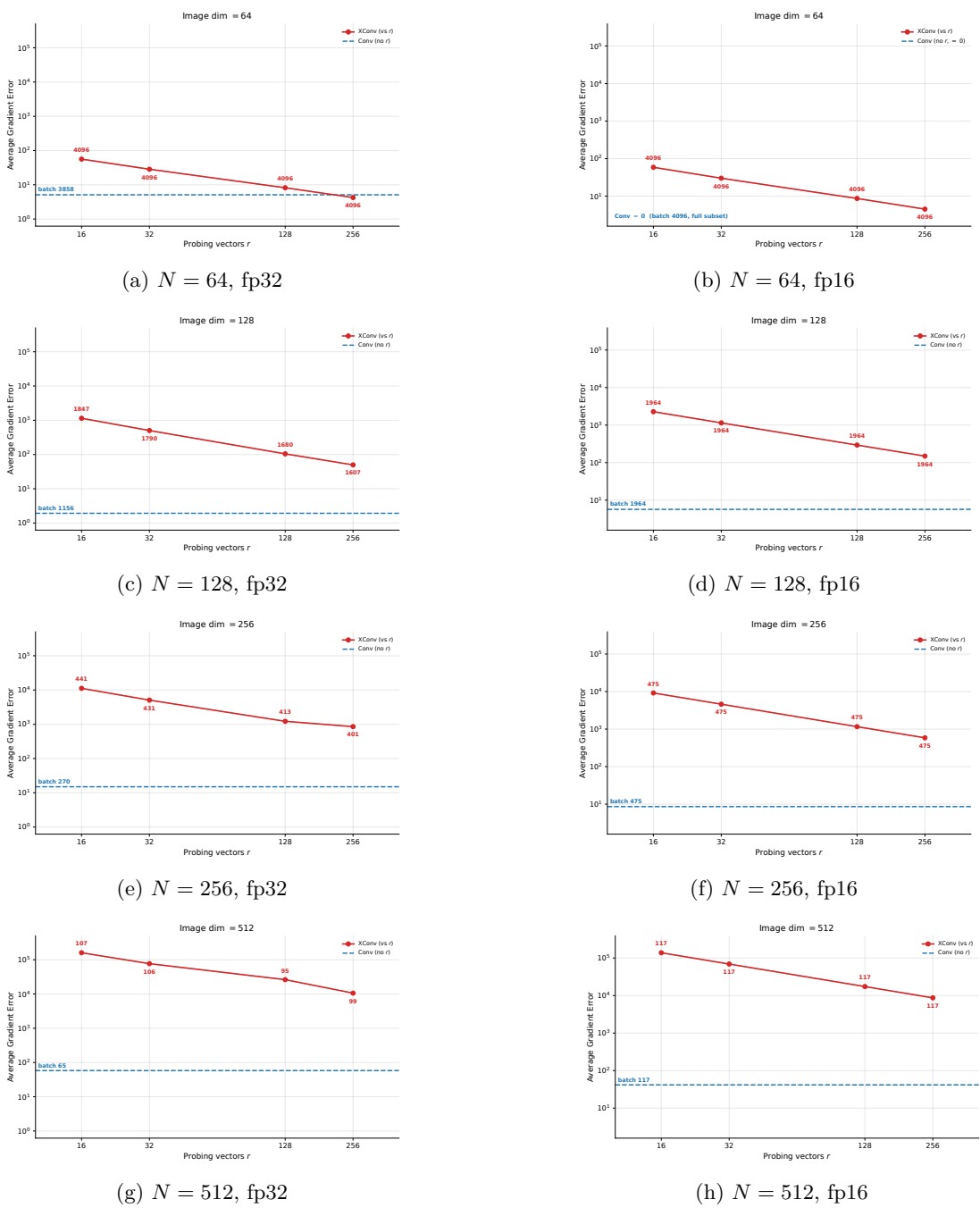

Figure 27: Average gradient error versus the number of probing vectors $r$ for VanillaNet at image dimensions $N \in \{64, 128, 256, 512\}$, in single precision (left) and half precision (right), under the 80 GB memory budget. The red curve is adaptive XConv and the blue dashed line the standard convolution floor. The single-precision error sits above the floor, consistent with VanillaNet's wide layers and Theorem 1, and is non-monotone in $r$ owing to the adaptive layer selection; at half precision the XConv error coincides with the convolution floor.

### A.6.2   MNIST with PyTorch

- NVIDIA Tesla K80 (Azure NC24 4xK80, one K80 per case) GPU
- 50 epochs

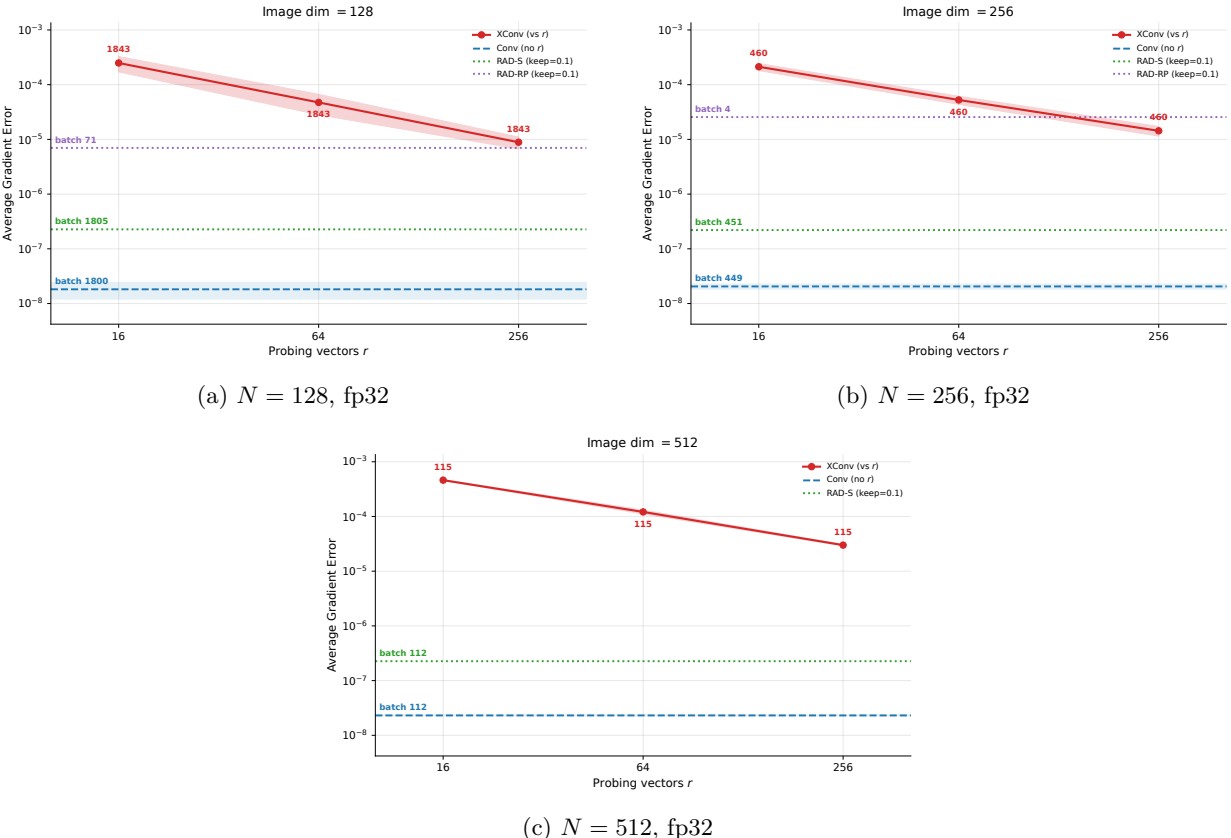

(a) $N = 128$, fp32

(b) $N = 256$, fp32

(c) $N = 512$, fp32

Figure 28: Average gradient error versus the number of probing vectors $r$ for U-Net at image dimensions $N \in \{128, 256, 512\}$ in single precision, under the 80 GB memory budget. The red curve is XConv, the blue dashed line the standard convolution floor, and the green dotted line the RAD-S reference at keep fraction 0.1. The half-precision panels are omitted because at fp16 the gradient error of every method collapses below the rounding floor.

Table 7: MNIST network and sizes for training with our Julia implementation for a batch size $B$.

| Layer | kernel size | Input size ($C_{\text{in}} \times N_x \times N_y$) | Output size ($C_{\text{out}} \times N_x \times N_y$) |
|---|---|---|---|
| Conv2d | (3, 3) | B×1×28×28 | B×16×28×28 |
| ReLU | – | B×16×28×28 | B×16×28×28 |
| MaxPool | (2, 2) | B×16×28×28 | B×16×14×14 |
| Conv2d | (3, 3) | B×16×14×14 | B×32×14×14 |
| ReLU | – | B×32×14×14 | B×32×14×14 |
| MaxPool | (2, 2) | B×32×14×14 | B×32×7×7 |
| Conv2d | (3, 3) | B×32×7×7 | B×32×7×7 |
| ReLU | – | B×32×7×7 | B×32×7×7 |
| MaxPool | (2, 2) | B×32×7×7 | B×32×3×3 |
| Flatten | – | B×32×3×3 | B×288 |
| Dense | – | B×288 | B×10 |

- Stochastic Line Search (SLS (Vaswani et al., 2019)) with initial learning rate of 1.0 and default SLS parameters
- MNIST dataset for varying batch size and number of probing vectors
- PyTorch implementation

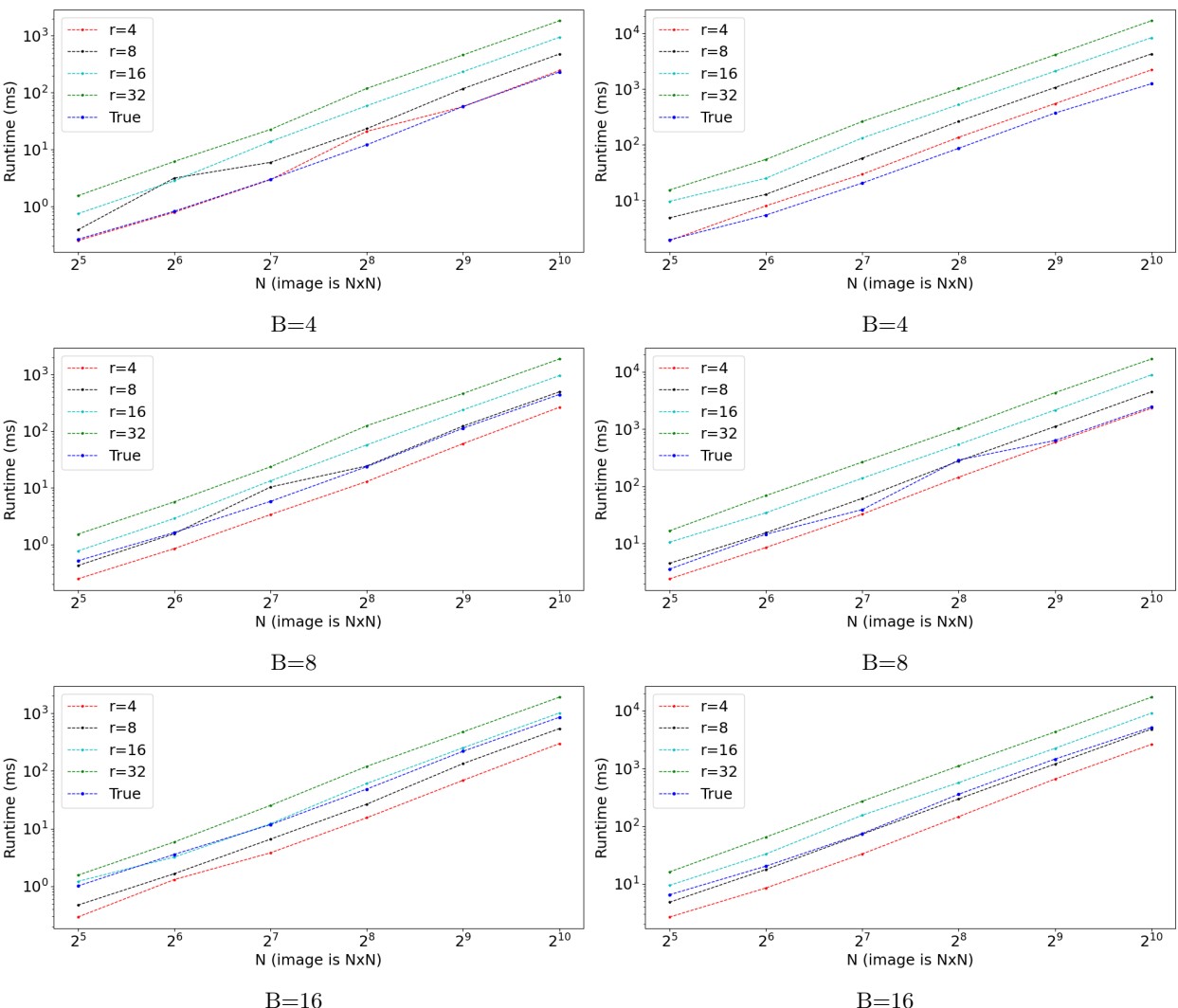

Figure 29: CPU benchmark on an *Intel(R) Xeon(R) CPU E3-1270 v6 @ 3.80GHz* node. The left column contains the runtimes for 4 channels and the right column for 32 channels. For batch sizes: B = 4 to 16.

### A.6.3 CIFAR-10 with PyTorch

- NVIDIA Tesla K80 (Azure NC6) GPU
- 100 epochs
- Stochastic Gradient Descent optimizer
- Initial learning rate of 0.001 with cosine annealing scheduler. When using probing, the learning rate is scaled by a factor of 1.5.
- CIFAR-10 dataset
- PyTorch implementation

### A.7 DIP Super-Resolution and Inpainting

### A.8 Peak Memory Curves

We present the full set of peak memory curves for all three models across image resolutions ($N$) and both precisions, under the 80 GB memory budget. SqueezeNet is shown in Figure 32, VanillaNet in Figure 33, and

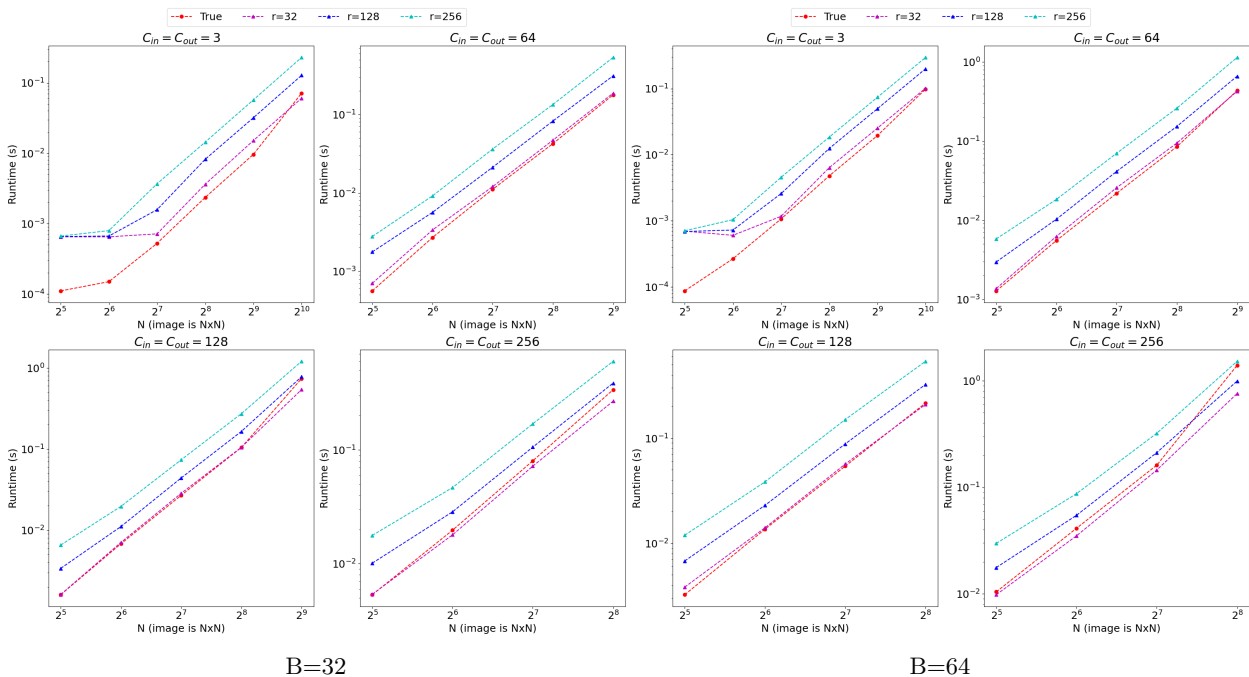

Figure 30: GPU benchmark on a *RTX 2000 Ada Generation* for a single gradient for smaller batch sizes $B = 32$ and $B = 64$, with varying image sizes $N$ and number of channels $C_\text{in} = C_\text{out}$. Results for larger batch sizes ($B = 128, 256$) are shown in Figure 12.

Table 8: MNIST network and sizes for training with PyTorch on the MNIST dataset for a batch size $B$.

| Layer | kernel size | Input size $(C_\text{in} \times N_x \times N_y)$ | Output size $(C_\text{out} \times N_x \times N_y)$ |
|---|---|---|---|
| Conv2d | (3, 3) | B×1×28×28 | B×32×28×28 |
| ReLU | – | B×32×28×28 | B×32×28×28 |
| Conv2d | (3, 3) | B×32×28×28 | B×64×28×28 |
| ReLU | – | B×64×28×28 | B×64×28×28 |
| MaxPool | (2, 2) | B×64×28×28 | B×64×14×14 |
| Dropout | – | B×64×14×14 | B×64×14×14 |
| Flatten | – | B×64×14×14 | B×12544 |
| Dense | – | B×12544 | B×128 |
| ReLU | – | B×128 | B×128 |
| Dropout | – | B×128 | B×128 |
| Dense | – | B×128 | B×10 |
| Log Softmax | – | B×10 | B×10 |

U-Net in Figure 34. For SqueezeNet and VanillaNet each solid curve is a fixed batch size and the dashed line of the matching color is the corresponding standard convolution; for U-Net each curve is a method (standard convolution, XConv at $r \in \{16, 64, 256\}$, RAD-RP and RAD-S at keep fraction 0.1) and the single dashed line is the memory budget.

## A.9 MNIST Classification with SLS

Figure 35 shows the full MNIST test accuracy curves when training with the Stochastic Line Search optimizer across a range of batch sizes and probing vectors.

Table 9: CIFAR-10 network and sizes for training with PyTorch on the CIFAR-10 dataset for a batch size $B$.

| Layer | kernel size | Input size ($C_{\text{in}} \times N_x \times N_y$) | Output size ($C_{\text{out}} \times N_x \times N_y$) |
|---|---|---|---|
| Conv2d | (5, 5) | B×3×32×32 | B×16×32×32 |
| ReLU | – | B×16×32×32 | B×16×32×32 |
| Conv2d | (5, 5) | B×16×32×32 | B×32×32×32 |
| ReLU | – | B×32×32×32 | B×32×32×32 |
| AvgPool | (2, 2) | B×32×32×32 | B×32×16×16 |
| Conv2d | (5, 5) | B×32×16×16 | B×32×16×16 |
| ReLU | – | B×32×16×16 | B×32×16×16 |
| Conv2d | (5, 5) | B×32×16×16 | B×32×16×16 |
| ReLU | – | B×32×16×16 | B×32×16×16 |
| AvgPool | (2, 2) | B×32×16×16 | B×32×8×8 |
| Flatten | – | B×32×8×8 | B×2048 |
| Dense | – | B×2048 | B×10 |
| Log Softmax | – | B×10 | B×10 |

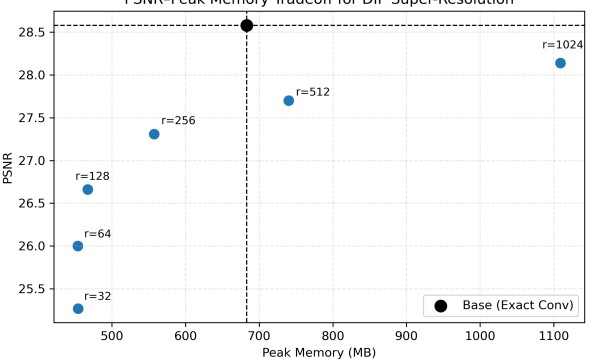

(a) Peak signal-to-noise ratio (PSNR) vs. peak memory tradeoff for DIP (super-resolution) on Set5.

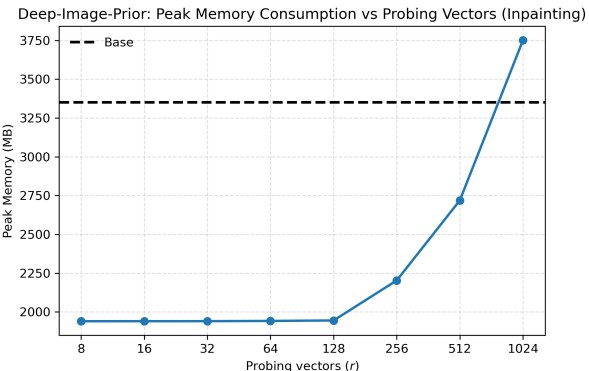

(b) DIP inpainting peak memory consumption as a function of the number of probing vectors $r$.

Figure 31: DIP peak memory results on super-resolution and inpainting. The horizontal dashed line indicates the memory consumption of exact convolution. PSNR measures the quality of the reconstructed signal relative to the original; higher values indicate better reconstruction. In both super-resolution (left) and inpainting (right), XConv attains memory savings.

## A.10  U-Net Generative Modeling

The training and validation U-Net loss curves for probing vectors $r \in \{32, 64, 128, 256\}$ are shown in Figure 36.

We further report the average gradient error of the U-Net diffusion model as a function of image dimension in Figure 37. As with the other architectures, the approximation error decreases with the number of probing vectors and remains within one order of magnitude of the exact gradient across resolutions.

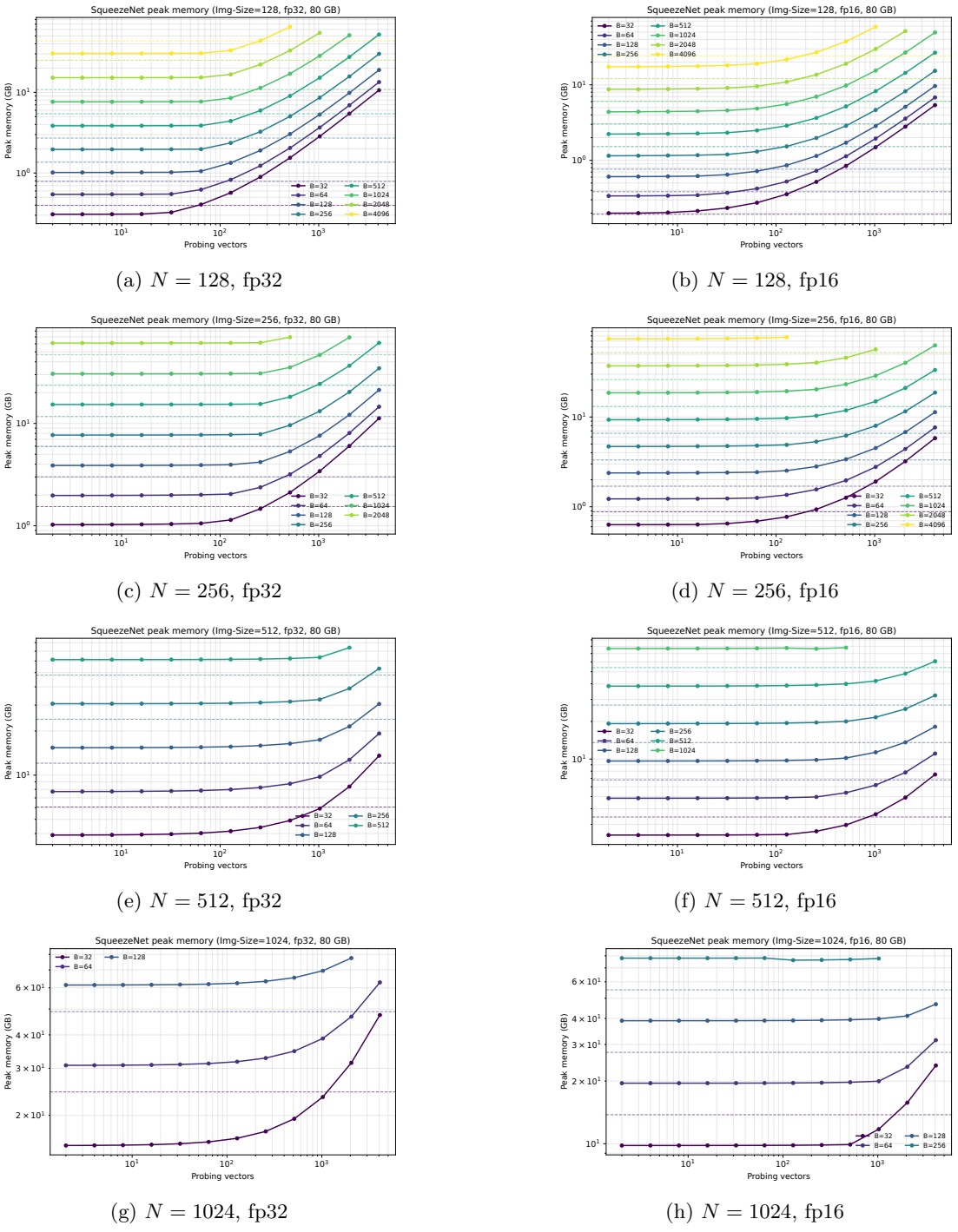

Figure 32: Peak memory curves for SqueezeNet at image dimensions $N \in \{128, 256, 512, 1024\}$, in single precision (left) and half precision (right), under the 80 GB memory budget.

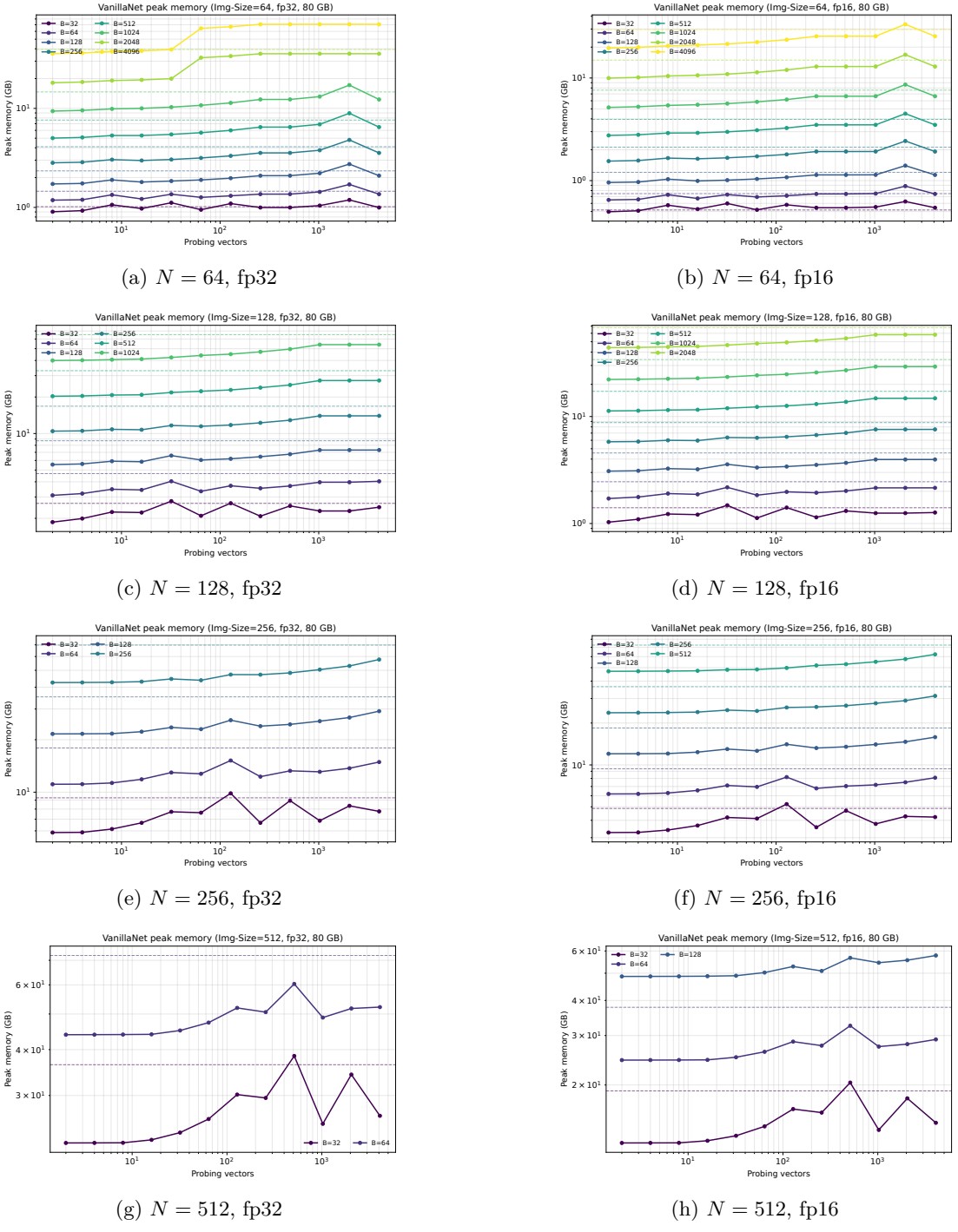

Figure 33: Peak memory curves for VanillaNet at image dimensions $N \in \{64, 128, 256, 512\}$, in single precision (left) and half precision (right), under the 80 GB memory budget. Owing to the adaptive application of randomized trace estimation, peak memory does not vary monotonically with the number of probing vectors.

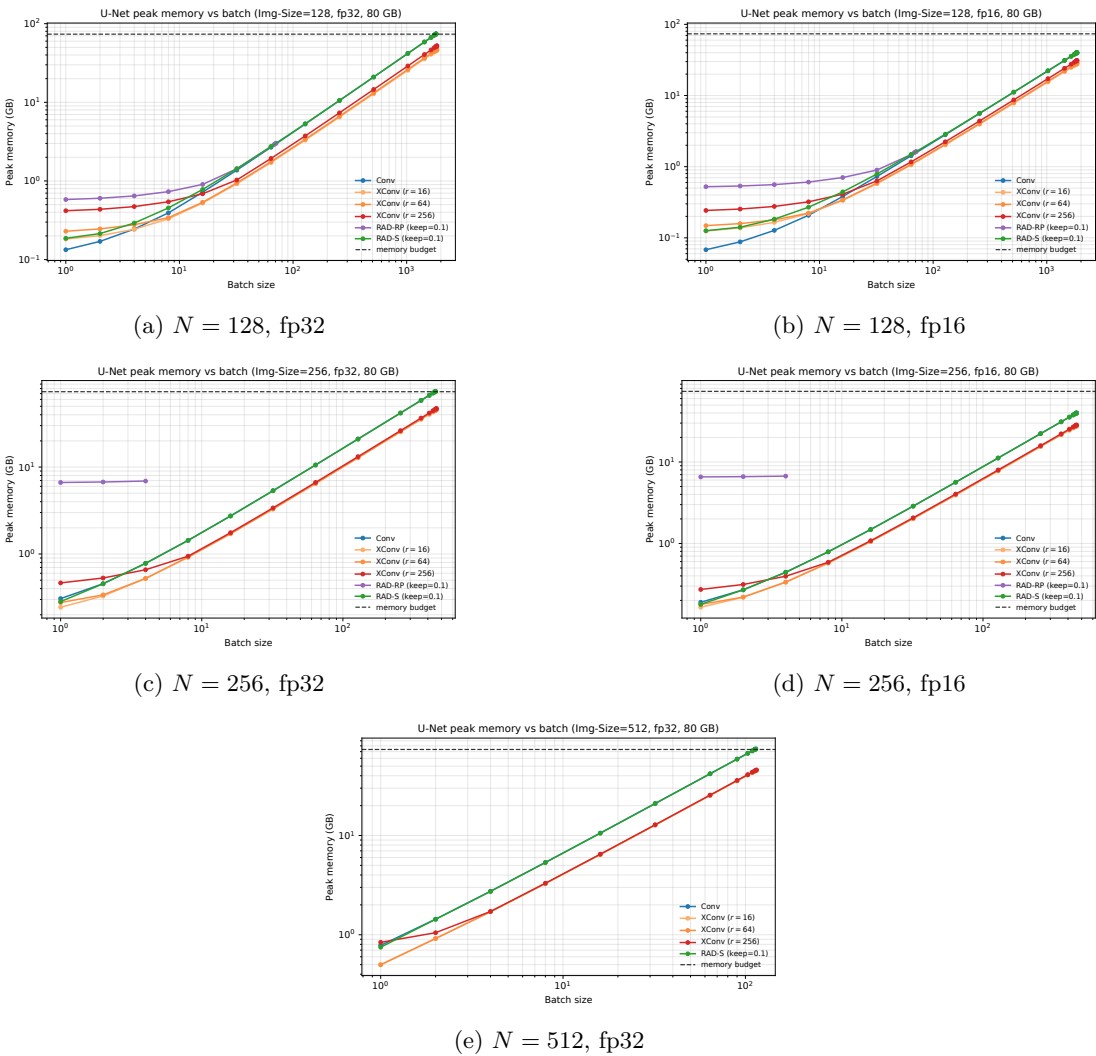

(a) $N = 128$, fp32

(b) $N = 128$, fp16

(c) $N = 256$, fp32

(d) $N = 256$, fp16

(e) $N = 512$, fp32

Figure 34: Peak memory curves for U-Net as a function of batch size, comparing standard convolution, XConv at $r \in \{16, 64, 256\}$, and the RAD-RP and RAD-S baselines at keep fraction 0.1; the dashed line is the 80 GB memory budget. Panels cover $N = 128$ and $N = 256$ in both precisions and $N = 512$ in single precision; half precision was not re-run at $N = 512$, and RAD-RP is infeasible at $N = 512$ and is therefore dropped from that panel.

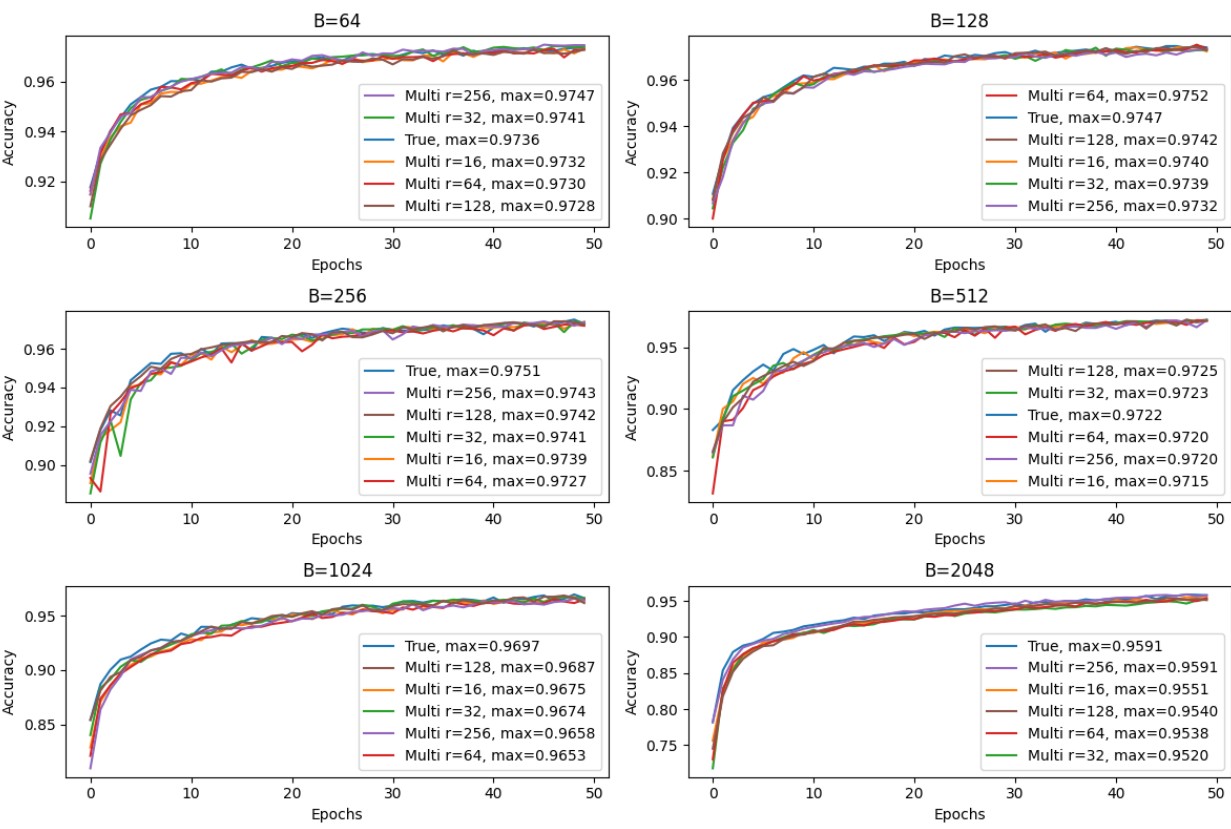

Figure 35: MNIST test accuracy vs. epoch for varying batch sizes $B \in \{64, \ldots, 2048\}$ and probing vectors $r \in \{16, \ldots, 256\}$, trained with the Stochastic Line Search algorithm (SLS, (Vaswani et al., 2019)). XConv converges to competitive accuracy for $r \geq 16$ across all batch sizes.

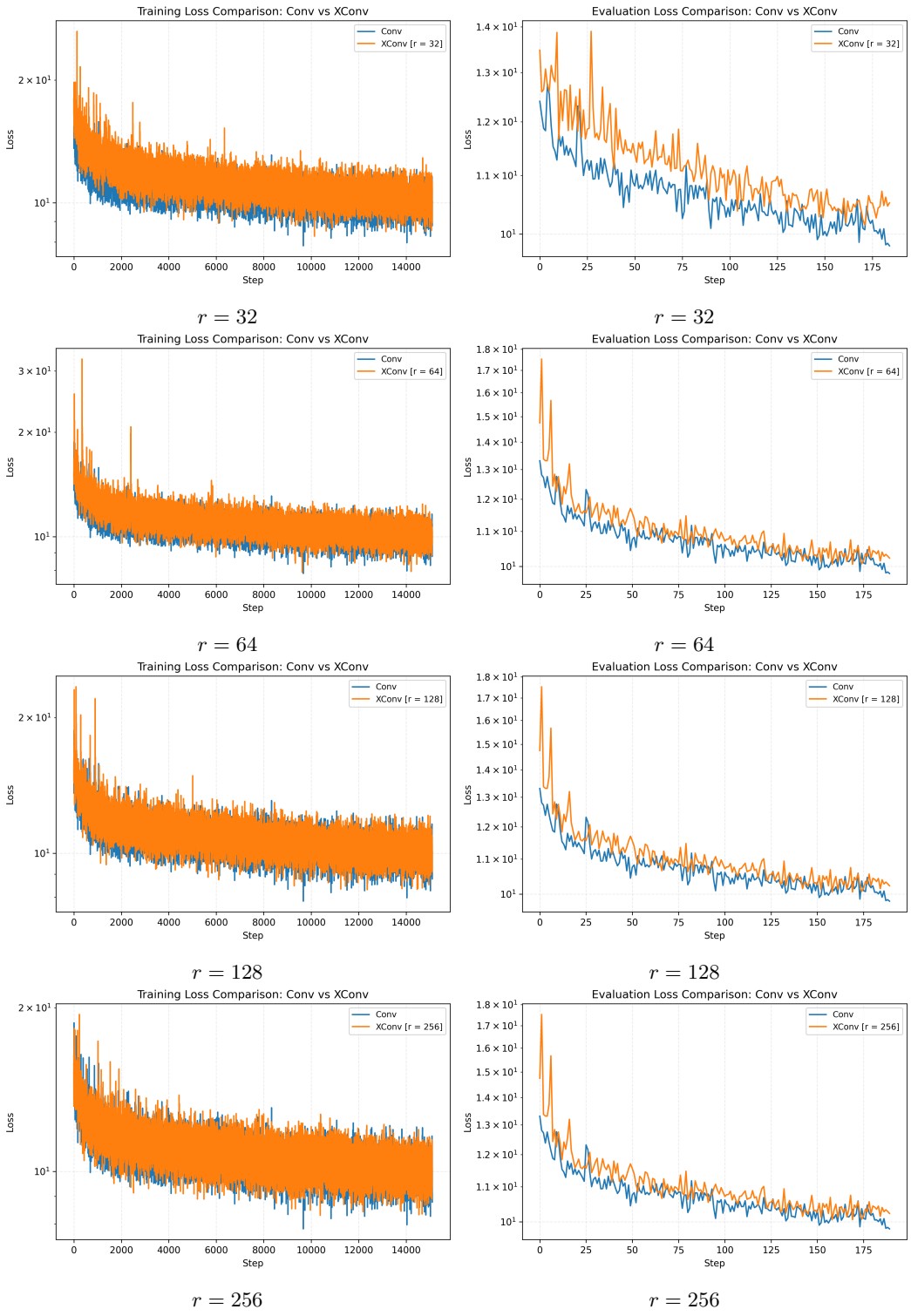

Figure 36: U-Net training (left) and validation (right) loss curves comparing standard convolution and XConv across probing vectors $r \in \{32, 64, 128, 256\}$. While XConv exhibits noisier optimization dynamics during training, its validation loss closely matches standard convolution and converges to a comparable scale as $r$ increases.

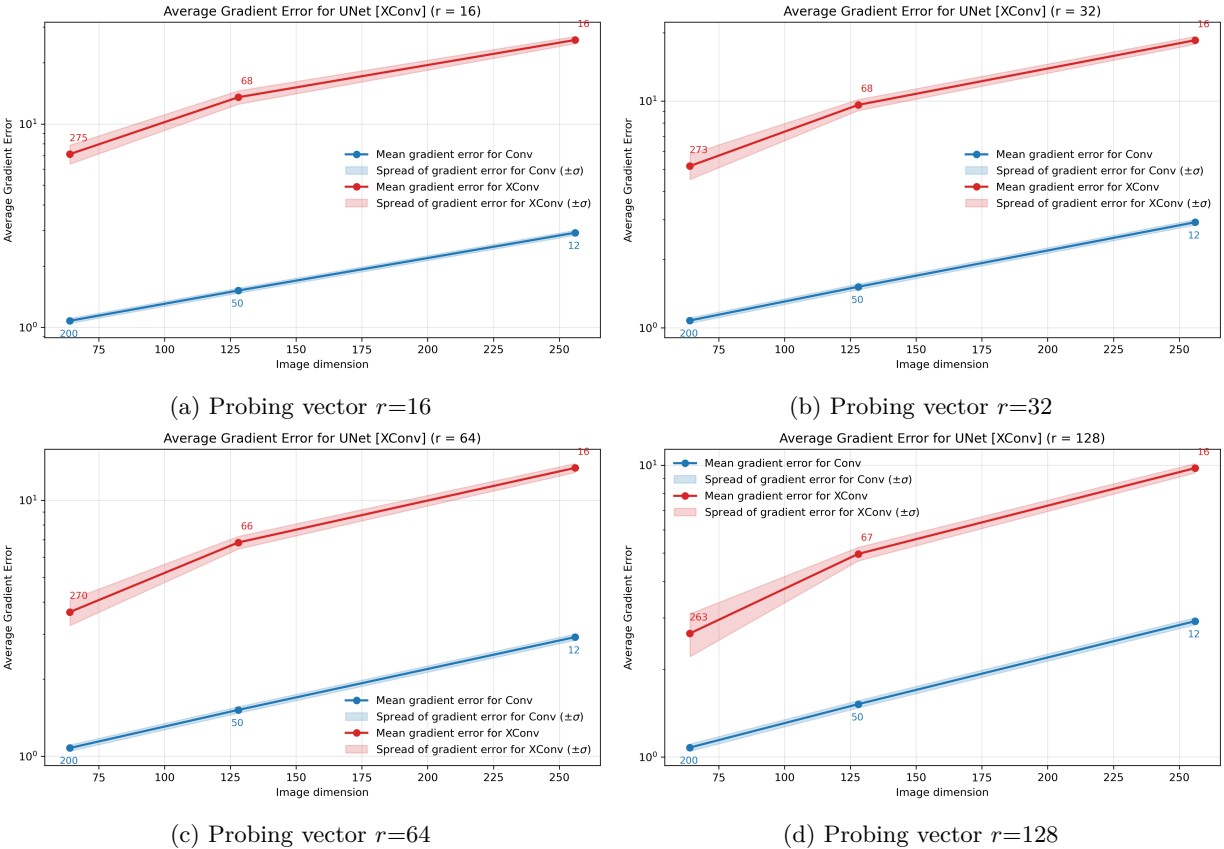

(a) Probing vector $r$=16

(b) Probing vector $r$=32

(c) Probing vector $r$=64

(d) Probing vector $r$=128

Figure 37: Average gradient error versus image dimension for the U-Net diffusion model used in generative modeling, for probing vectors $r \in \{16, 32, 64, 128\}$. The annotated values indicate the maximum batch size that fits in memory for each method (blue: standard convolution, red: XConv). The approximation error decreases with the number of probing vectors and remains within one order of magnitude of the exact gradient.

