# OpenReview forum: "XConv: Low-memory stochastic backpropagation for convolutional layers"
_TMLR — Under review for TMLR_

### Review · Reviewer_sLFL · 2026-05-12

**Summary Of Contributions:**

This paper proposes a trace-estimator-based method for computing the backward pass gradients for convolutional layers in CNNs. The proposed method utilizes deterministically pseudorandom probing vectors and matrices, which can estimate the backward-pass gradient, thereby removing the need to store the operation’s hidden state. This change can result in significant memory savings without compromising the network’s ability to converge or perform reliable inference, as demonstrated by the authors through various experiments.

Strengths:

-	The paper is well written and easy to follow in most areas.
-	The proposed method is clever and backed by a compelling theoretical and many empirical analyses, demonstrating the methods efficacy.
-	The authors further back their claims using computational and memory analysis, successfully establishing the idea of a memory-compute-accuracy tradeoff.

Weaknesses:

-	The main method and theoretical sections are somewhat challenging to follow and may benefit from further refinement. Figure 1a, for example, is unclear and should be expanded.
-	While the authors consider many different convolutional applications, most of the datasets are very small, which may not provide an accurate picture of the methods feasibility at scale and in production environments.
-	Many of the claims in the paper are weak, often with contradicting or subjective interpretations of the results.
-	The primary motivation of reduced memory footprint is not compelling given the example use-cases provided in the paper. Additionally, the computational experiments are conducted on very old GPUs (K80), and will likely make the overhead of the proposed method unfavorable when comparing against modern tensor-core based GPUs.

**Additional Comments:**

Aside from the requested changes, could the authors clarify what their intended application space is for the proposed method? Increasing the image batch size by a factor of < 2 is not as helpful to modern GPUs given that more can be combined using distributed training. As an alternative motivation, the authors may want to consider edge finetuning / continual training applications where their method will have more impact. However, such cases typically only offer lower-bit precision (often down to INT4).

Other questions:

-	Why does the vanilla-net memory requirement not increase monotonically with increasing probe-vector count?
-	Is the probing matrix Z unique per batch, and do the results keep the probing vector count fixed as the batch-size increases? This would mean fewer paths per image sample, which could result in degraded estimation as the batch size increases.
-	As stated in my review, much of the analysis is subjective, where it may be helpful to soften or reframe the analysis. For example, the MNIST classification accuracy is likely off by > 2 $\sigma$ for the best XConv case. Additionally, claiming “visually consistent” is not a robust measurement for performance.
-	For the case of MNIST generation, it is difficult to tell how the quality varies with XConv given the task simplicity. This is where it would be helpful to use a more challenging and visually distinct dataset, such as one with RGB images. Additionally, maintaining consistent sampling parameters would be helpful (same seed, steps, etc.) across plotted samples.

**Audience:**

Yes

**Audience Explanation:**

The proposed method is interesting, and while it may not be well motivated for conventional large-CNNs, it may be useful in low-memory/low-power regimes on the edge. Furthermore, the proposed method may also be applicable to attention-based layers, which could speedup transformer-based inference and reduce finetuning memory requirements.

**Broader Impact Concerns:**

No concerns.

**Claims And Evidence:**

Yes

**Claims Explanation:**

The authors perform a theoretical analysis which derives an error bound for the proposed method, and confirm this theoretical behavior holds mostly true through empirical experiments. The authors also conduct training experiments across classification, generation, super-resolution, and segmentation, all of which demonstrate that the proposed method does not prevent networks from converging, nor does it result in significant quality loss. However, some of the claims are weakened by the experiments, such as the error becoming unbounded in VanillaNet, and poorer performance on most other tasks when compared to the baseline network with full gradients.

**Requested Changes:**

Primary changes:

-	In the case of VanillaNet, the error appears to grow unbounded with larger image size. Could the authors provide some theoretical explanation for why this is occurring?
-	In the peak memory vs probing vector plots, the authors should indicate the required number of vectors to match the AGE of the true gradient method. It is currently unclear whether the method requires more memory to match the gradient error of the true method.
-	The GPU computational overhead comparison is made with K80 GPUs and should be made with a more modern GPU containing tensor cores. CuDNN takes advantage of these units, which will likely require reframing the performance overhead argument for XConv as it is difficult to beat the TC performance.
-	The authors should also clarify the conditions under which the memory performance is evaluated. Is this using 32-bit precision?  If so, would it still hold up for 16-bit (BF16), and could the authors investigate whether XConv holds up for lower precisions, or if it requires high precision to remain effective?
-	Larger / more challenging dataset experiments would be preferrable for the classification and generative modeling cases. MNIST and CIFAR10 are too easy and may not have sufficient headroom to demonstrate favorable or unfavorable performance. The same may also be true for the segmentation case, where PASCAL VOC may provide a more realistic task.
-	The super-resolution and inpainting experiments should present fidelity metrics rather than a qualitative analysis. It would also be helpful to compare against bi-linear and bi-cubic upscaling. The same metric requirement applies to the CIFAR10 classification experiments.

Minor changes:

-	It may be helpful to show the AGE plots (e.g. figure 5-6) with the same vertical scale for small and large r. This would be clearer to the reader that increasing r does reduce the error gap.
-	The authors should verify that the example image for DIP in Figure 17 is correct, it seems to contain too many artifacts.

---

> ### Author Response · Authors · 2026-06-16
>
> We thank the reviewer for the thorough review. Point-by-point below.
>
> **Method and theory clarity (Figure 1, AGE plots, DIP figures).** We add an "Overview of XConv" section stating in words that XConv replaces the exact weight gradient with a randomized trace estimate from a few probing vectors, and the resulting memory--accuracy tradeoff. Figure 1(a) is redrawn as an explicit three-step pipeline (compress the activation by probing $\to$ reuse the stored compressed activation with the residual $\to$ recover the gradient trace estimate), with the single stored tensor and its dimensions labeled; Figure 1(b) now makes the post-orthogonalization drop in cross-channel interference visible. AGE is replotted as a continuous function of $r$ on a shared vertical scale against the exact-gradient floor. The DIP figures are regenerated and the artifacts are gone.
>
> **Dataset scale.** We add: 3D segmentation finetuning (MONAI spleen CT) -- XConv $r{=}4$ matches exact-gradient Dice (**0.9622 vs 0.9620**, within run-to-run variation) at **-17.5%** peak memory; a seismic generative model on $128{\times}128$ Parihaka images; and CIFAR-10 diffusion (RGB, evaluated by FID), with the sampling seed and step count fixed across plotted samples and stated in the captions. GlaS is small but a standard histology benchmark; the 3D task supplies the larger, realistic dense-prediction setting.
>
> **Weak or subjective claims; fidelity metrics.** Each claim is now tied to a metric with run-to-run variation. SR reports PSNR (**29.495** for XConv $r{=}256$ vs **29.464** exact, head image) plus bicubic/bilinear baselines; inpainting reports PSNR (Kate, Vase) plus the peak-memory reduction; CIFAR reports test accuracy and FID (**38.2** exact vs **40.2** at $r{=}128$). For MNIST we report the accuracies and note the small differences across $r$ fall within training stochasticity. "Visually consistent" is replaced by FID/PSNR/Dice/accuracy throughout.
>
> **Motivation and hardware (K80 to a modern tensor-core GPU).** The overhead study is rerun on a modern tensor-core GPU (RTX 2000 Ada); at the larger batch sizes XConv enables, it stays competitive with exact convolution despite cuDNN tensor-core kernels. The intended setting is training/finetuning under a fixed memory budget where data parallelism does not lower per-device memory: high-resolution 2D/3D inputs, deep wide networks, and on-device/continual finetuning (the 3D finetuning result is a concrete edge instance). The batch-size gain is one lever, not the central claim.
>
> **VanillaNet error growth with image size.** The error is governed by Theorem 1's variance term, which aggregates over channel blocks and so scales with $C_\text{in}{\times}C_\text{out}$. VanillaNet is far wider (several layers $>10^6$, max $\approx1.7{\times}10^7$) than SqueezeNet/U-Net, so under fixed $r$ the error is larger and grows with resolution -- bounded, not unbounded, and reduced by increasing $r$. We add a per-layer channel-width breakdown. XConv is an approximation; we do not claim to beat exact gradients -- the relevant property is competitive performance at a memory point below exact convolution, with the gap controlled by $r$.
>
> **Matching the true-gradient AGE on the peak-memory plots.** The estimator is Monte Carlo: it matches the exact gradient only in the limit. In practice AGE approaches the exact mini-batch sampling-noise floor at moderate $r$, well below the activation dimension -- the regime where the memory advantage holds. We plot AGE vs $r$ with that floor marked, so the rank at which XConv reaches it is directly visible.
>
> **Precision (fp32 and fp16).** Originals are fp32. At fp16, XConv's AGE falls to the rounding floor (coinciding with the exact-conv floor) for SqueezeNet and VanillaNet -- including VanillaNet's wide layers -- so it is negligible vs the precision error; for U-Net every method drops below the floor (uninformative), so those panels are omitted. XConv is thus especially suited to reduced precision (finetuning, on-device). INT4 is future work.
>
> **VanillaNet's non-monotone memory vs. probing count.** We use an adaptive variant: a layer is probed only when $r < H_\ell W_\ell$. Increasing $r$ changes which layers are probed (some revert to exact convolution), so peak memory is not monotone in $r$.
>
> **Probing matrix $\mathbf{Z}$ and batch size.** $\mathbf{Z}$ is not stored or fixed per batch: it is regenerated in the backward pass from a seed saved in the forward pass, fresh per layer and iteration (Algorithm 1). Under a fixed budget we hold $r$ while varying the batch -- a deliberate tradeoff point; this raises per-sample variance, reduced by a larger $r$. The estimator is unbiased at any batch size.
>
> **Attention layers and edge deployment.** The 3D finetuning result supports the edge use case; extending trace estimation to attention is noted as future work.

---

### Review · Reviewer_taE8 · 2026-05-26

**Summary Of Contributions:**

This paper proposes XConv, a low-memory replacement for standard convolutional layers. The key idea is to store compressed activations and approximate convolution weight gradients via randomized trace estimation. The method is theoretically motivated and evaluated on classification, diffusion, super-resolution, inpainting, and segmentation tasks, showing around 2× or more memory savings with relatively small performance drops.

**Additional Comments:**

I think this is an interesting and useful paper with a clear practical motivation. The idea is technically sound and the results are promising. However, I would like to see stronger comparisons with standard memory-efficient training baselines and larger-scale experiments before being fully convinced.

**Audience:**

Yes

**Audience Explanation:**

The studied problem is practical and interesting.

**Claims And Evidence:**

No

**Claims Explanation:**

This mainly lies in the weak empirical comparisons, as in my detailed comments.

**Requested Changes:**

1. The empirical comparison with existing memory-saving methods, especially activation checkpointing and other approximate-gradient methods, is still limited. It would be useful to compare under matched memory/compute budgets.
2. Some experiments are relatively small-scale, such as MNIST/CIFAR and small diffusion settings. More results on modern large CNN/ConvNeXt-style models would strengthen the claims.
3. The method introduces extra hyperparameters such as the number of probing vectors, but the practical guideline for choosing them is not very clear.
4. Although the performance gap is small in many cases, the approximation error may be task- or architecture-dependent, and the failure cases are not sufficiently analyzed.

---

> ### Author Response · Authors · 2026-06-16
>
> We thank the reviewer for finding the paper practical and interesting. The two stated concerns -- stronger memory-saving baselines and larger scale -- are addressed directly by (1) a matched-memory comparison against RAD and (2) larger-scale experiments (3D segmentation finetuning, a $128{\times}128$ seismic generative model, and CIFAR-10 diffusion); we hope these support a reassessment of the claim-support rating.
>
> **Matched-budget comparison against checkpointing and other approximate-gradient methods.** We add a matched-memory comparison against RAD (Oktay et al., 2021) -- the closest approximate-gradient baseline -- in random-projection (RAD-RP) and sparse (RAD-S) variants. On a U-Net, with batch taken as the largest that fits: XConv has the lowest peak memory and largest batch; RAD-RP reconstructs full activations in the backward pass, admits only a small fraction of XConv's batch, and is infeasible at image dimension 512. On fidelity it is mixed -- RAD-S is more accurate than XConv at comparable batch, while XConv beats RAD-RP -- so XConv and RAD-S are complementary points on the memory--fidelity frontier, with only XConv scaling to high resolution and requiring no computational-graph surgery. The other approximate-gradient methods are not drop-in for convolutional training: MeZO (Malladi et al., 2023) abandons backprop, has estimator variance that grows with the parameter count, and is established only for LLM finetuning; approximate/memory-sharing backprop (Yang et al., 2024) targets nonlinearity and normalization layers, not convolutions; direct feedback alignment uses fixed random feedback ill-suited to CNNs. Checkpointing and invertible nets preserve *exact* gradients via recomputation or architectural constraints -- a different axis -- so a matched-budget comparison would reintroduce the recomputation overhead XConv avoids. We therefore place XConv within the approximate-gradient family, where RAD is the appropriate matched-memory baseline.
>
> **Small-scale experiments; modern large CNN / ConvNeXt-style models.** We add CIFAR-10 diffusion (color, beyond MNIST), a seismic generative model on $128{\times}128$ Parihaka images, and a 3D segmentation finetuning task (MONAI spleen CT) where XConv $r{=}4$ matches exact Dice (**0.9622 vs 0.9620**, within run-to-run variation) at **-17.5%** peak memory. We add a seismic facies-classification benchmark where $r{=}128$ shows a moderate gap (mIoU **0.473 vs 0.538**), consistent with the dense-prediction failure regime we now analyze. The evaluation already includes TriConvUNeXt (a ConvNeXt-style network with dilated and deformable convolutions) and VanillaNet (several layers with $C_\text{in}{\times}C_\text{out}>10^6$, up to $\approx1.7{\times}10^7$), exercising the wide, large-activation regime.
>
> **Practical guideline for choosing the probing vectors.** A new Practical Guidance section makes the choice concrete. A layer stores $r{\times}B$, so $r$ and $B$ are interchangeable at the layer level; at the network level $B$ scales every layer while $r$ affects only convolutional activations, so $B$ is the more effective lever. Since fidelity improves with $r$, use the largest $r$ the budget allows and recover memory by lowering $B$. The cost of a larger $r$ is compute (our runtime benchmarks), so $(r,B)$ trades accuracy against compute. Because the probing noise is unbiased, a smaller batch is compensated by a lower learning rate and more steps -- as in ordinary SGD; the volumetric finetuning uses exactly this recipe, under which even $r{=}4$ matches exact accuracy. The only quantities the user sets are $r$ and the batch size; the architecture and training loop are unchanged.
>
> **Task- and architecture-dependent failure cases.** The error is governed by Theorem 1, whose bound aggregates over channel blocks and grows with $C_\text{in}{\times}C_\text{out}$; wide-channel networks (VanillaNet, several layers $>10^6$) therefore show the largest AGE under fixed $r$. Generative and dense-prediction tasks are more sensitive than classification and need a larger $r$, with gains diminishing beyond $r>1024$. These failure regimes -- wide-channel layers, gradient-sensitive tasks, high-fidelity demands -- are now explicit.
>
> **Additional comments.** The two requested deltas (stronger baselines, larger scale) are met by the RAD comparison and the 3D-segmentation, seismic, and CIFAR-10 diffusion experiments.

---

### Review · Reviewer_AWmQ · 2026-06-03

**Summary Of Contributions:**

This paper presents XConv, a low-memory stochastic backpropagation method for convolutional layers that serves as a drop-in replacement for standard 2D and 3D convolutions while reducing activation storage during training. Specifically, the authors first reformulate convolutional weight gradients as trace computations over matrices constructed from the layer input, the backpropagated residual, and shift operators. Second, they approximate these traces using randomized trace estimation, so that only compressed activations and a random seed need to be stored in the forward pass, while the probing matrix can be regenerated during backpropagation. Third, to handle multi-channel convolutions efficiently, the paper introduces a multi-channel randomized probing scheme and a sparse block-wise probing strategy to reduce cross-channel interference, together with convergence guarantees and error bounds for the proposed estimator. Finally, experiments on classification, diffusion-based generative modeling, super-resolution, inpainting, and segmentation show that XConv can achieve performance close to exact-gradient convolutional training when a sufficient number of probing vectors is used, while typically reducing memory consumption by about 2× or more; however, the accuracy–memory trade-off depends on the architecture, task, and probing-vector budget, and larger probing budgets can increase memory and computation. However, I have some concerns about this paper. My detailed comments are as follows.

*Strengths:*

1. The authors reformulate convolutional weight gradients as randomized trace estimation problems, enabling XConv to store compressed activations during the forward pass while preserving standard backpropagation and avoiding architecture-specific constraints.
2. The authors introduce a multi-channel sparse probing strategy to approximate convolutional gradients with reduced cross-channel interference, and provide convergence guarantees and error bounds to support the reliability of the proposed estimator.

*Weaknesses:*

1. Although sparse probing is introduced to reduce cross-channel interference, the authors explicitly state that they cannot theoretically prove that decreasing the sampling probability preduces the error. This weakens the theoretical justification for a key design choice, and a more systematic ablation on p would be necessary.
2. The method is presented as a drop-in replacement for standard convolution, but it still requires choices such as the number of probing vectors, adaptive layer selection, and special handling for layers like ReLU because XConv no longer stores the original state variable. This suggests that the practical integration cost may be higher than implied.
3. In dense prediction tasks, increasing the number of probing vectors improves gradient fidelity but also causes memory consumption to grow rapidly, with diminishing gains when r>1024. This raises the concern that XConv’s memory advantage may shrink in settings that require high gradient accuracy.
4. The paper discusses checkpointing, invertible networks, and randomized automatic differentiation in the motivation and related work, but the main experiments mostly compare XConv with standard convolution. A direct comparison under matched memory or wall-clock budgets would be needed to demonstrate advantages over established memory-saving baselines.
5. For inpainting, the evidence is mainly qualitative, while the segmentation experiment is conducted on the relatively small GlaS dataset with 165 images. Although the reported results are promising, larger-scale and more diverse evaluations would be needed to support stronger claims of robustness.

**Audience:**

Yes

**Audience Explanation:**

Yes. At least some members of the TMLR audience would likely be interested in the findings of this paper. The paper addresses a practical and broadly relevant problem in neural network training: the memory cost of storing intermediate activations for backpropagation, especially in convolutional architectures. Its proposed method, XConv, is relevant to researchers working on efficient training, randomized numerical methods, and scalable vision models because it attempts to reduce memory usage while preserving standard backpropagation and avoiding strong architectural constraints. The paper also provides both theoretical analysis, including convergence guarantees and error bounds, and empirical evaluations across several tasks such as classification, generative modeling, super-resolution, inpainting, and segmentation. These aspects make the findings potentially useful to a subset of TMLR readers interested in optimization, efficient deep learning systems, and convolution-based model scaling. However, the level of interest may depend on whether readers prioritize convolutional architectures, since the method is less directly relevant to transformer-centric work.

**Claims And Evidence:**

Yes

**Claims Explanation:**

Partially yes. The main claims of the submission are supported by a reasonably broad set of evidence, but some evidence is not fully convincing. On the positive side, the paper provides theoretical support for the randomized trace estimator, including convergence guarantees and error bounds, and the experiments evaluate XConv from several perspectives: gradient fidelity, memory usage, wall-clock performance, and downstream task performance. The memory-saving claim is also supported by layer-wise and peak-memory comparisons, where the authors report that XConv can achieve memory reductions of around 2×or more depending on the architecture.

However, I would not say that all claims are supported equally well. First, the empirical comparisons are mainly against standard convolution, while direct comparisons with strong memory-saving baselines such as activation checkpointing or RAD are limited. Second, the downstream evidence is somewhat uneven: for inpainting and super-resolution, part of the support is qualitative, and the segmentation experiment is conducted on the relatively small GlaS dataset with 165 images. Third, the method’s performance depends strongly on the number of probing vectors: increasing rimproves gradient fidelity but also increases memory consumption, and the paper itself notes diminishing improvements when r>1024 in segmentation.

Overall, the evidence is accurate and generally clear, and it supports the basic feasibility of XConv. However, it is only moderately convincing for stronger claims about broad practicality or superiority over existing memory-saving methods. I would therefore answer: yes, but with reservations.

**Requested Changes:**

See Weaknesses.

---

> ### Author Response · Authors · 2026-06-16
>
> We thank the reviewer for recognizing that XConv reformulates convolutional weight gradients as randomized trace estimation, enabling compressed activation storage with standard backpropagation and supporting error bounds.
>
> **Sampling probability $p$ and the requested ablation.** $p_n$ is a parameter of the *theoretical* sparse-probing model behind Theorem 1's interference bound, not a user hyperparameter; the deployed method uses dense or block-orthogonal probing, and the only user knob is $r$, which we ablate across all architectures via the AGE-vs-$r$ curves. The dependence on $p$ is already characterized by Theorem 1, which exposes competing effects -- a per-block variance term growing as $p$ decreases ($1/\sqrt{p_n}$) versus reduced cross-channel interference -- so no monotone "smaller $p$ yields smaller error" can hold in general, and the favorable operating point depends on the relative block norms $\|\mathbf{A}^{n,j}\|_F$. This crosstalk-versus-variance behavior is a property of the bound, not an unexamined design choice; the benefit of structured probing is shown directly in Figure 1(b).
>
> **Practical integration cost.** Converting a network is a single in-place `convert_net` call replacing every Conv2d/3d (and, optionally, ReLU with a bit-wise BReLU that stores a 1-byte sign mask), leaving architecture, objective, and training loop unchanged. Adaptive layer selection is automatic -- probing is disabled where the spatial extent is below the probing rank. The only user-facing quantity is $r$, for which the new Practical Guidance section gives a rule of thumb.
>
> **Diminishing memory advantage at $r>1024$.** This is the tunable tradeoff, not a flaw: across classification, generative modeling, super-resolution, inpainting, and segmentation, competitive performance is reached at $r\le512$ (and $r{=}1024$ for dense GlaS), well below where the memory advantage vanishes. The $r>1024$ regime is the high-fidelity extreme -- a user wanting near-exact gradients pays in memory; one tolerating approximation keeps the savings. We state this explicitly.
>
> **Comparison against memory-saving baselines under a matched budget.** We add a matched-memory comparison against RAD (Oktay et al., 2021) in RAD-RP and RAD-S variants. On a U-Net: XConv has the lowest peak memory and largest batch; RAD-RP reconstructs full activations in the backward pass, admits only a small fraction of XConv's batch, and is infeasible at image dimension 512. On fidelity, RAD-S is more accurate than XConv at comparable batch while XConv beats RAD-RP -- complementary points on the frontier, with only XConv spanning the full memory--fidelity range, scaling to high resolution, and requiring no graph surgery. The other approximate-gradient methods are not drop-in for convolutional training: MeZO (Malladi et al., 2023) abandons backpropagation, has estimator variance growing with the parameter count, and is established only for LLM finetuning; approximate/memory-sharing backprop (Yang et al., 2024) targets nonlinearity and normalization layers, leaving convolutions unaddressed; and direct feedback alignment replaces the backward pass with fixed random feedback ill-suited to CNNs. Checkpointing and invertible nets keep *exact* gradients via recomputation or architectural constraints -- a different axis -- so a matched-budget comparison would reintroduce the recomputation XConv avoids. RAD is the appropriate baseline, and we provide it.
>
> **Qualitative inpainting and the small GlaS set.** Super-resolution and inpainting now report fidelity: SR PSNR **29.495** (XConv $r{=}256$) vs **29.464** (exact) on the head image, with bicubic/bilinear baselines; inpainting PSNR on Kate and Vase plus the peak-memory reduction. For scale we add 3D segmentation finetuning (MONAI spleen CT): XConv $r{=}4$ matches exact Dice (**0.9622 vs 0.9620**, a 0.0002 gap within run-to-run variation) at **-17.5%** peak memory. We further add a seismic generative model ($128{\times}128$ Parihaka) and a seismic facies benchmark ($r{=}128$: mIoU **0.473 vs 0.538**, a dense-prediction failure case). GlaS is small but a standard histology benchmark; the added tasks extend the evidence to larger, more diverse settings.

---

### Author Response · Authors · 2026-06-16

We thank the reviewers. The reviews converge on two requests -- a matched-budget comparison against memory-saving baselines, and larger-scale evidence -- both addressed below. The revision also clarifies the method, replaces qualitative claims with quantitative metrics, repeats the overhead study on modern hardware, and adds lower-precision results.

XConv is a *tunable* memory--fidelity mechanism: we do not claim to match or beat exact gradients, but that competitive task performance is retained at a memory operating point below exact convolution, with the residual gap controlled by the number of probing vectors $r$.

## New results and revisions

| # | New result / revision | Raised by | Outcome |
|---|---|---|---|
| 1 | **RAD vs XConv, matched memory** (U-Net) | AWmQ, taE8 | XConv: lowest peak memory / largest batch; beats RAD-RP (infeasible at dim 512); RAD-S more accurate at matched memory; only XConv is drop-in (Figs 5, 9) |
| 2 | **3D segmentation, finetuning** (MONAI spleen CT) | AWmQ, taE8, sLFL | $r{=}4$ matches exact Dice (**0.9622 vs 0.9620**) at **-17.5%** peak memory (Table 5, Fig 25) |
| 3 | **Seismic generative** ($128{\times}128$ Parihaka) | taE8, sLFL, AWmQ | Comparable samples; AGE falls with $r$ (Figs 19--20) |
| 4 | **CIFAR-10 diffusion** (RGB) | sLFL, taE8 | FID **40.2** ($r{=}128$) vs **38.2** exact (Figs 17--18) |
| 5 | **Half-precision (fp16)** (SqueezeNet, VanillaNet) | sLFL | XConv AGE falls to the fp16 rounding floor (= exact-conv floor); U-Net fp16 omitted (all methods below floor) (Figs 4, 6) |
| 6 | **Overhead on tensor-core GPU** (RTX 2000 Ada) | sLFL | Competitive with exact convolution at larger batch (Fig 12) |
| 7 | **Fidelity metrics, SR + inpainting** (PSNR; bicubic/bilinear for SR) | sLFL, AWmQ | PSNR replaces the qualitative comparison (Table 3) |
| 8 | **Corrected DIP figures** | sLFL | Artifacts removed (Figs 21--22) |
| 9 | **Overview section + redrawn Fig 1** | sLFL | Clearer method/theory exposition |
| 10 | **Practical Guidance** (choosing $r$) | taE8, AWmQ | Use the largest $r$ the budget allows; lower the batch to fit; compensate noise with smaller LR / more steps |
| 11 | **AGE-vs-$r$ on a shared scale** | sLFL, AWmQ | Error gap shrinks with $r$ toward the exact floor (Figs 4--6) |
| 12 | **Seismic facies classification** | taE8, AWmQ | $r{=}128$: mIoU 0.473 vs 0.538 (failure regime) (Table 2, Fig 14) |

## Common points

1. **Tradeoff, not a free lunch.** XConv trades gradient fidelity for memory; the gap shrinks with $r$ and recovers the exact gradient in the limit.
2. **RAD is the right matched-memory baseline.** Checkpointing/invertible nets keep *exact* gradients via recomputation or architectural constraints (a different axis). Against RAD on a U-Net: XConv has the lowest peak / largest batch and beats RAD-RP (infeasible at high resolution); RAD-S is more accurate at matched memory; only XConv is drop-in.
3. **Scale.** Added 3D volumetric segmentation (finetuning), seismic generative ($128{\times}128$), and CIFAR-10 diffusion; the evaluation already includes VanillaNet ($C_{in}{\times}C_\text{out}{>}10^6$) and a ConvNeXt-style net (TriConvUNeXt).
4. **Precision.** Originals are fp32; fp16 added -- XConv error falls to the rounding floor for SqueezeNet/VanillaNet; U-Net fp16 omitted (uninformative). INT4 is future work.
5. **Hardware.** Overhead study repeated on RTX 2000 Ada (replacing the K80).
6. **One knob.** The user sets only $r$; conversion is one in-place call; `BReLU` stores a 1-byte sign mask.
7. **Clarity.** Overview-of-XConv section and a redrawn Figure 1.